# ContextFlow: Context-Aware Flow Matching for Trajectory Inference from Spatial Omics Data

## Abstract

Inferring trajectories from longitudinal spatially-resolved omics data is fundamental to understanding the dynamics of structural and functional tissue changes in development, regeneration and repair, disease progression, and response to treatment. We propose ContextFlow, a novel context-aware flow matching framework that incorporates prior knowledge to guide the inference of structural tissue dynamics from spatially resolved omics data. Specifically, ContextFlow integrates local tissue organization and ligand-receptor communication patterns into a transition plausibility matrix that regularizes the optimal transport objective. By embedding these contextual constraints, ContextFlow generates trajectories that are not only statistically consistent but also biologically meaningful, making it a generalizable framework for modeling spatiotemporal dynamics from longitudinal, spatially resolved omics data. Evaluated on three datasets, ContextFlow consistently outperforms state-of-the-art flow matching methods across multiple quantitative and qualitative metrics of inference accuracy and biological coherence.

## 1 Introduction

Flow matching (Lipman et al., 2023) is an emerging paradigm that provides an efficient approach for learning the complex latent dynamics, or normalizing flows (Papamakarios et al., 2021), of a system of variables, while enabling parametric flexibility to model data distributions. Inferring the underlying dynamics from sparse and noisy observations is a central challenge in many domains (Gontis et al., 2010; Brunton et al., 2016; Pandarinath et al., 2018; Li et al., 2025), where continuous trajectories are rarely captured; instead, cross-sectional snapshots, collected at discrete time points, are typically available. In single-cell RNA sequencing (scRNA-seq), this challenge becomes especially critical as the destructive nature of profiling technologies yields only unpaired population-level snapshots over time. Uncovering temporal dynamics from such snapshot data is essential for understanding developmental processes, disease progression, treatment and perturbation responses (Wagner & Klein, 2020). Traditional approaches often rely on heuristics or computationally intensive likelihood-based generative models, which struggle with scalability and flexibility in high-dimensional single-cell data. Flow matching overcomes these challenges by directly learning continuous latent dynamics that are constrained to match observed population-level distributions at sampled time points.

The state and function of cells within a tissue are affected by interactions with neighboring cells, extracellular matrix components, and local signaling gradients (Rao et al., 2021). Recent advances in spatial omics technologies, particularly spatial transcriptomics (ST), allow gene expression profiling without tissue dissociation, thereby preserving spatial context and providing a complementary view of cellular organization. The dynamics of complex cellular processes is affected by the tissue microenvironment, where cells engage in reciprocal communication with their neighbors (Dimitrov et al., 2022; Tanevski et al., 2025). A growing body of work highlights the critical role of spatial cell–cell communication patterns in shaping cellular phenotypes (Armingol et al., 2021). In particular, location-specific communication circuits between distinct cell types dynamically interact to reprogram cellular states and influence tissue-level behavior (Mayer et al., 2023; Aguadé-Gorgorió et al., 2024; Zheng et al., 2025). These insights, made possible by the spatiotemporal resolution of transcriptomics data, pave the way for understanding the mechanisms by which cellular interactions drive tissue

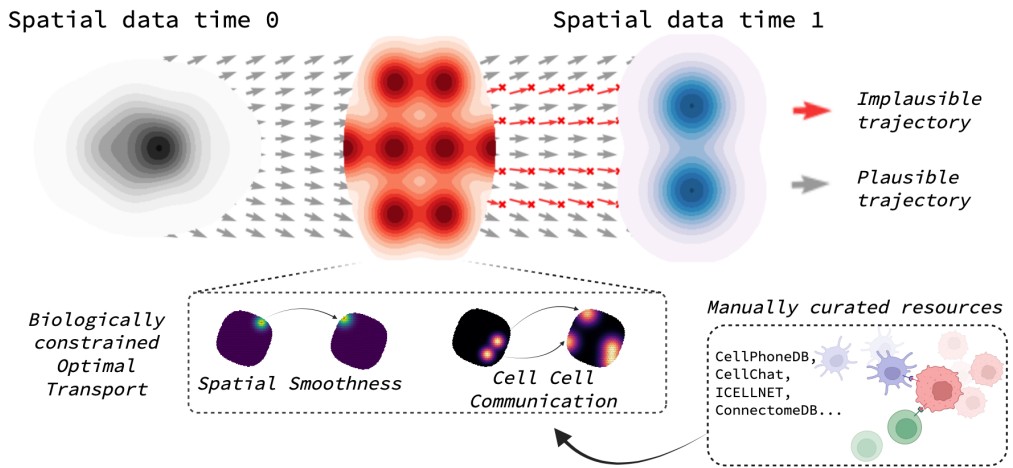

Figure 1: ContextFlow integrates local tissue organization and ligand-receptor communications to learn biologically meaningful trajectories from spatial omics data. Prior knowledge acts as a soft filter that discourages implausible transitions while preserving flexibility in trajectory inference.

organization and function in organogenesis (Chen et al., 2022), regeneration (Ben-Moshe et al., 2021; Wei et al., 2022), disease progression (Kukanja et al., 2024), and treatment response (Liu et al., 2024).

Optimal transport (OT) has become a foundational framework to align spatially resolved samples and infer putative developmental or temporal couplings (Zeira et al., 2022; Liu et al., 2023). As a result, state-of-the-art flow matching frameworks such as minibatch-OT flow matching (MOTFM) (Tong et al., 2024) use OT-derived couplings to define conditional paths to train velocity fields, thus overcoming the lack of generative capabilities in optimal transport. The OT formulation adopted in MOTFM, however, does not account for the contextual richness present in spatial transcriptomics and can result in trajectories that are statistically optimal yet biologically implausible (see Figure 4a in Appendix I.1 for an illustration). While recent studies have extended widely-used OT objectives (Halmos et al., 2025; Ceccarelli et al., 2025) for spatial transcriptomics, they primarily focus on pairwise alignment of populations across conditions or modalities and do not explicitly incorporate the cell–cell communication patterns that drive cellular state transitions.

To address the above limitations, we introduce a novel flow matching-based framework, *ContextFlow*, that incorporates spatial priors for modeling temporal tissue dynamics (Figure 1). By encoding local tissue organization and ligand-receptor-derived spatial communication patterns into prior-regularized optimal transport formulations, ContextFlow fully exploits the contextual richness of spatial omics data and embeds both structural and functional aspects of tissue organization into its objective, thereby generating more biologically informed trajectories. In summary, our contributions are as follows:

- We leverage local tissue organization and local ligand–receptor communication patterns to extract biologically meaningful features from spatial omics data, and encode them into a biologically-informed transition plausibility matrix to constrain temporal dynamics (Section 3.2).
- We design two novel integration schemes—cost-based and entropy-based—that incorporate the prior knowledge into an OT-coupled flow matching framework, both amenable to efficient Sinkhorn optimization and scalable on modern hardware (Section 3.3).
- Comprehensive experiments on regeneration and developmental datasets demonstrate that ContextFlow consistently outperforms baseline methods under both interpolation and extrapolation settings across metrics that capture biological plausibility and statistical fidelity (Section 4).

## 2 PRELIMINARIES

### 2.1 FLOW MATCHING BASICS

Flow matching (Lipman et al., 2023) is a simulation-free and sample-efficient generative framework for training continuous normalizing flows (Chen et al., 2018). Given a pair of source and target

data distributions over $\mathbb{R}^d$ with densities $q_0 = q(\boldsymbol{x}_0)$ and $q_1 = q(\boldsymbol{x}_1)$, the problem task is to learn a time-varying velocity vector field $u_\theta : [0, 1] \times \mathbb{R}^d \to \mathbb{R}^d$, whose continuous evolution is captured by a function in the form of a neural-net-based model with weights $\theta$, that can transform $q_0$ to $q_1$ through integration via an ordinary differential equation (ODE). To be more specific, *flow matching* (FM) seeks to optimize $\theta$ by minimizing a simple regression loss between $u_\theta$ and a target time-varying velocity vector field $u_t : [0, 1] \times \mathbb{R}^d \to \mathbb{R}^d$ as follows:

$$\min_\theta \ \mathbb{E}_{t \sim \mathcal{U}(0,1), \boldsymbol{x} \sim p_t(\boldsymbol{x})} \big\| u_\theta(t, \boldsymbol{x}) - u_t(\boldsymbol{x}) \big\|^2. \tag{1}$$

Here, $\mathcal{U}(0, 1)$ is the uniform distribution over $[0, 1]$, and $p_t : [0, 1] \times \mathbb{R}^d \to \mathbb{R}_+$ denotes a time-varying probability path induced by $u_t$ such that (i) $p_t$ is a probability density function for any $t \in [0, 1]$, (ii) $p_t$ satisfies the two boundary conditions: $p_{t=0} = q_0$ and $p_{t=1} = q_1$, and (iii) the connection between $p_t$ and $u_t$ can be characterized by the *transport equation* (Villani et al., 2008): $\frac{\partial p_t(\boldsymbol{x})}{\partial t} = -\nabla \cdot (u_t(\boldsymbol{x}) p_t(\boldsymbol{x}))$, where $\nabla$ is the divergence operator. From a dynamical system's view, $u_t$ defines an ODE system $d\boldsymbol{x} = u_t(\boldsymbol{x}) dt$. The corresponding solution to the ODE, usually termed as the probability flow, can then transport any $\boldsymbol{x}_0 \sim q_0$ to a point $\boldsymbol{x}_1 \sim q_1$ along $u_t$ from $t = 0$ to $t = 1$. While the flow matching objective in Equation 1 is simple and intuitive, it is generally intractable in practice: the closed-form velocity vector field $u_t$ is unknown for arbitrary source and target distributions ($q_0$ and $q_1$), and multiple valid probability paths $p_t$ may exist between them.

## 2.2 Conditional Flow Matching

The central idea of conditional flow matching is to express the target probability path $p_t$ via a mixture of more manageable *conditional probability paths* (Lipman et al., 2023). By marginalizing over some conditioning variable $z$, both $p_t$ and $u_t$ can be constructed using their conditional counterparts:

$$p_t(\boldsymbol{x}) = \int p_t(\boldsymbol{x}|\boldsymbol{z}) q(\boldsymbol{z}) d\boldsymbol{z}, \quad u_t(\boldsymbol{x}) = \int u_t(\boldsymbol{x}|\boldsymbol{z}) \frac{p_t(\boldsymbol{x}|\boldsymbol{z}) q(\boldsymbol{z})}{p_t(\boldsymbol{x})} d\boldsymbol{z}, \tag{2}$$

where $q(\boldsymbol{z})$ denotes the distribution of the conditioning variable $\boldsymbol{z}$, and $p_t(\boldsymbol{x}|\boldsymbol{z})$ is selected such that the boundary conditions are satisfied: $\int p_{t=0}(\boldsymbol{x}|\boldsymbol{z}) q(\boldsymbol{z}) = q_0$ and $\int p_{t=1}(\boldsymbol{x}|\boldsymbol{z}) q(\boldsymbol{z}) = q_1$. Theorem 1 of Lipman et al. (2023) proves that $p_t$ and $u_t$ defined by Equation 2 satisfy the transport equation, suggesting that $p_t$ is a valid probability path generated by $u_t$. To avoid the intractable integrals, Lipman et al. (2023) proposed the following optimization of *conditional flow matching* (CFM), and proved its equivalence to the original flow matching objective in terms of gradient computation:

$$\min_\theta \ \mathbb{E}_{t \sim \mathcal{U}(0,1), z \sim q(\boldsymbol{z}), \boldsymbol{x} \sim p_t(\boldsymbol{x}|\boldsymbol{z})} \big\| u_\theta(t, \boldsymbol{x}) - u_t(\boldsymbol{x}|\boldsymbol{z}) \big\|^2. \tag{3}$$

By choosing an appropriate conditional velocity vector field $u_t(\boldsymbol{x}|\boldsymbol{z})$, we can train the neural network using Equation 3 without requiring a closed-form solution of the conditional probability path $p_t(\boldsymbol{x}|\boldsymbol{z})$, thus avoiding the intractable integration operation. Therefore, the remaining task is to define the conditional probability path and velocity vector field properly such that we can sample from $p_t(\boldsymbol{x}|\boldsymbol{z})$ and compute $u_t(\boldsymbol{x}|\boldsymbol{z})$ efficiently for solving the optimization problem in Equation 3.

**Gaussian Conditional Probability Paths.** A specific choice proposed in Lipman et al. (2023) is Gaussian conditional probability paths and their corresponding conditional velocity vector fields:

$$p_t(\boldsymbol{x}|\boldsymbol{z}) = \mathcal{N}(\boldsymbol{x} \mid \mu_t(\boldsymbol{z}), \sigma_t(\boldsymbol{z})^2 \mathbf{I}), \quad u_t(\boldsymbol{x}|\boldsymbol{z}) = \frac{\sigma_t'(\boldsymbol{z})}{\sigma_t(\boldsymbol{z})}(\boldsymbol{x} - \mu_t(\boldsymbol{z})) + \mu_t'(\boldsymbol{z}), \tag{4}$$

where $\mu_t : [0, 1] \times \mathbb{R}^d \to \mathbb{R}^d$ denotes the time-varying mean of the Gaussian distribution, $\mu_t'$ is its derivative with respect to time, $\sigma_t : [0, 1] \times \mathbb{R}^d \to \mathbb{R}_+$ stands for the time-varying scalar standard deviation, and $\sigma_t'$ stand for the corresponding derivative. In particular, Lipman et al. (2023) set $q(\boldsymbol{z}) = q(\boldsymbol{x}_1)$, $\mu_t(\boldsymbol{z}) = t\boldsymbol{x}_1$, and $\sigma_t(\boldsymbol{z}) = 1 - (1 - \sigma)t$. Then, we can see that $u_t(\boldsymbol{x}|\boldsymbol{z})$ transports the standard Gaussian distribution $p_{t=0}(\boldsymbol{x}|\boldsymbol{z}) = \mathcal{N}(\boldsymbol{x}; 0, I)$ to a Gaussian distribution with mean $x_1$ and standard deviation $\sigma$, namely $p_{t=1}(\boldsymbol{x}|\boldsymbol{z}) = \mathcal{N}(\boldsymbol{x}; \boldsymbol{x}_1, \sigma^2)$ for any target point $\boldsymbol{x}_1$. By letting $\sigma \to 0$, the marginal boundary conditions can easily be verified. Tong et al. (2024) further generalized the application scope to arbitrary source distributions, by setting

$$q(\boldsymbol{z}) = q(\boldsymbol{x}_0) q(\boldsymbol{x}_1), \quad \mu_t(\boldsymbol{z}) = (1 - t)\boldsymbol{x}_0 + t\boldsymbol{x}_1, \quad \sigma_t(\boldsymbol{z}) = \sigma. \tag{5}$$

This choice satisfies the boundary conditions $p_{t=0}(\boldsymbol{x}) = q_0$ and $p_{t=1}(\boldsymbol{x}) = q_1$ when $\sigma \to 0$. Based on Equation 4, the conditional velocity vector field has a simple analytical form $u_t(\boldsymbol{x}|\boldsymbol{z}) = \boldsymbol{x}_1 - \boldsymbol{x}_0$.

## 2.3 Flow Matching with Optimal Transport Couplings

The conditionals construction specified by Equation 5 corresponds to the simplest choice of *independent coupling*, where $z = (x_0, x_1)$ with source $x_0$ and target $x_1$ are independently sampled from $q(z) = q(x_0)q(x_1)$. The use of couplings for constructing the sampling paths in the CFM framework naturally connects to the optimal transport theory (Villani et al., 2008). Choosing OT-based couplings has several advantages over independent coupling, including smaller training variance and more efficient sampling (Pooladian et al., 2023; Tong et al., 2024).

Since the classical Kantorovich's formulation (refer Appendix F) has computational complexity that is cubic with respect to the sample size, a popular alternative is to add an extra regularization term, resulting in *entropic optimal transport* (EOT), to approximately solve the optimal transport problem while reducing the computational costs from cubic to quadratic:

$$\pi_{\text{eot}}^*(\epsilon) := \operatorname{argmin}_{\pi \in \Pi(q_0, q_1)} \int_{\mathbb{R}^d \times \mathbb{R}^d} \|x_0 - x_1\|_2^2 \, d\pi(x_0, x_1) + \epsilon H(\pi \mid q_0 \otimes q_1), \qquad (6)$$

where $\epsilon > 0$ is the regularization parameter, and $H(\pi \mid q_0 \otimes q_1)$ denotes the relative entropy (or Kullback-Leibler divergence) with respect to $\pi$ and the product measure $q_0 \otimes q_1$. The optimization problem in Equation 6 can be viewed as a special case of the static Schrödinger bridge problem (Bernton et al., 2022), which can be efficiently solved in a mini-batch fashion via the Sinkhorn algorithm (Cuturi, 2013). Theoretically, one can prove that $\pi_{\text{eot}}^*(\epsilon)$ recovers the Kantorovich's OT coupling $\pi_{\text{ot}}^*$ when $\epsilon \to 0$ (see Equation 16 in Appendix F for its formal definition) and $\pi_{\text{eot}}^*(\epsilon)$ corresponds to the independent coupling $q_0 \otimes q_1$ when $\epsilon \to \infty$.

## 3 Regularizing the Flow with Spatial Priors

### 3.1 Problem Formulation

We focus on the task of inferring spatiotemporal trajectories, i.e., inferring the dynamic evolution of the cell states across time from spatially resolved gene expression data. Let $0 = t_1 < t_2 < \ldots < t_{m+1} = 1$ be a sequence of normalized time points. For any $i \in \{1, 2, \ldots, m + 1\}$, let $q_i$ be the data distribution over $\mathbb{R}^d$ at time point $t_i$. Given $\{\mathbf{X}_{t_i}\}_{i \in \{1, 2, \ldots, m+1\}}$, where $\mathbf{X}_{t_i} = \{x_i(k)\}_{k=1}^{n_i}$ is the gene expressions at time $t_i$ consisting of $n_i$ snapshot data sampled from $q_i$, the objective is to learn a neural velocity vector field $u_\theta : [0, 1] \times \mathbb{R}^d \to \mathbb{R}^d$ to faithfully characterize the temporal evolution of spatially resolved tissues over time, such that the induced probability path $p_t$ can describe the state of each cell at time $t \in [0, 1]$. This task can be viewed as a continuous temporal generalization of the pairwise generative modeling task described in Section 2.1.

A promising candidate solution is *conditional flow matching with entropic OT couplings* (EOT-CFM), by targeting linear conditional velocity vector fields for each pair of consecutive time points. Specifically, for any $t \in [0, 1]$ satisfying $t \in [t_i, t_{i+1}]$, define

$$p(x|z) = \mathcal{N}\left( \frac{t_{i+1} - t}{t_{i+1} - t_i} x_i + \frac{t - t_i}{t_{i+1} - t_i} x_{i+1}, \sigma^2 \mathbf{I} \right), \quad u_t(x|z) = \frac{x_{i+1} - x_i}{t_{i+1} - t_i}, \qquad (7)$$

where the conditioning variable is selected as $z = (x_i, x_{i+1})$, and $p(z)$ is the joint probability measure with marginals $q_i$ and $q_{i+1}$ corresponding to the EOT coupling $\pi_{\text{eot}}^*(\epsilon)$ defined in Equation 6. It can be easily verified that the above construction satisfies the boundary condition at each time point $t_i$. To train $u_\theta$, we can randomly sample a mini-batch of data at each time, run the Sinkhorn algorithm (Cuturi, 2013) to obtain the entropic OT couplings for each consecutive pair, and iteratively update the model weights $\theta$ using stochastic gradient descent with CFM regression loss (Equation 3).

Despite their enhanced ability to model system dynamics, state-of-the-art OT-CFM frameworks lack provisions to fully exploit the contextual richness and integrate the biological prior knowledge that can be inferred from other associated data modalities. Existing approaches can generate statistically optimal trajectories by targeting probability paths induced by (entropic) OT couplings along the temporal dimension. However, they may overlook important functional or structural prior information, leading to biologically implausible trajectories (see Figure 4a in Appendix I.1 for an illustration).

### 3.2 Introducing Spatial Priors & Transitional Plausibility

To faithfully model the spatial context and cellular organization of spatial omics data, we introduce two types of spatial priors and explain how they relate to the transitional plausibility between locations and cell states at different time points.

**Spatial Smoothness.** Tissues are well-organized systems. Within a microenvironment, neighboring cells respond to the same set of external mechanical stimuli and intercellular communication, which affects their states in a similar manner and results in local smoothness of cell-type-specific expression. Due to tissue heterogeneity, we cannot assume a common reference coordinate frame across tissue samples or even slices at $t_i$ and $t_j$ at a larger scale. However, the same heterogeneity allows us to consider the spatial coherence and neighborhood consistency (Greenwald et al., 2024; Ceccarelli et al., 2025) as a proxy for relative cell localization, which cannot change significantly across short time intervals. Therefore the aggregate expression within the microenvironment of each cell can be used to quantify the transitional plausibility in consecutive time points.

Specifically, let $c_i = (\boldsymbol{x}_i, \boldsymbol{s}_i)$ and $c_j = (\boldsymbol{x}_j, \boldsymbol{s}_j)$ be cells at time points $t_i$ and $t_j$, respectively, where $\boldsymbol{x}_i, \boldsymbol{x}_j \in \mathbb{R}^d$ denote their gene expression profiles, and $\boldsymbol{s}_i, \boldsymbol{s}_j \in \mathbb{R}^2$ denote their spatial coordinates in the relative tissue reference frame. Let $\mathrm{TP}(c_i, c_j)$ denote the *transitional plausibility*, i.e., the likelihood that $c_i$ evolves to $c_j$ between $t_i$ and $t_j$. Spatial smoothness suggests that $\mathrm{TP}(c_i, c_j)$ is inversely related to the difference between the average expression profiles of their local neighborhoods:

$$\mathrm{SS}(c_i, c_j) = \left\| \frac{1}{|\mathcal{N}_r(c_i)|} \sum_{c \in \mathcal{N}_r(c_i)} \boldsymbol{x}(c) - \frac{1}{|\mathcal{N}_r(c_j)|} \sum_{c \in \mathcal{N}_r(c_j)} \boldsymbol{x}(c) \right\|_2^2, \tag{8}$$

where $\mathcal{N}_r(c_i) = \{c : \|\boldsymbol{s}(c) - \boldsymbol{s}(c_i)\|_2 \leq r\}$ denotes the set of neighboring cells of $c_i$ in the same tissue slice, $|\mathcal{N}_r(c_i)|$ is the cardinality of $\mathcal{N}_r(c_i)$, and $\boldsymbol{x}(c)$ is the gene expression profile of cell $c$.

**Cell-Cell Communication Patterns.** *Cell–cell communication* (CCC) has a critical role in the regulation of numerous biological processes, including development, apoptosis, and the maintenance of homeostasis in health and disease (Armingol et al., 2024). A major type of CCC is ligand–receptor (LR) signaling, in which ligands expressed by one cell bind to cognate receptors on another, initiating intracellular cascades that ultimately affect the state of the cell (i.e., its expression profile) (Armingol et al., 2021). There are numerous databases of prior knowledge of ligand-receptor binding and computational methods that use these databases to systematically link gene expression with the activity of ligand-receptor-mediated communication.

Specifically, we can represent each cell $c_i$ by a vector $f_{\mathrm{LR}} \in \mathbb{R}^p$, where each entry corresponds to one of $p$ possible ligand–receptor pairs and encodes the extent of $c_i$'s participation in communication through that pair. The $\mathrm{TP}(c_i, c_j)$ between cells in different tissue slices is higher when they exhibit similar ligand-receptor communication patterns $f_{\mathrm{LR}}$ (see Figure 8 for an illustration). We define $\mathrm{LR}(c_i, c_j)$, the dissimilarity between the ligand–receptor communication patterns in the microenvironments of cells $c_i$ and $c_j$, as:

$$\mathrm{LR}(c_i, c_j) = \| f_{\mathrm{LR}}(\mathcal{N}_r(c_i)) - f_{\mathrm{LR}}(\mathcal{N}_r(c_j)) \|_2^2, \tag{9}$$

### 3.3 ContextFlow: CFM with Context-Aware OT Couplings

Our proposed framework, graphically depicted in Figure 1, consists of the following three main steps:

**Transitional Plausibility Matrix.** First, we create a sequence of *transitional plausibility matrices* (TPMs) to encode the biological priors for each pair of consecutive time points. Specifically, let $\mathbf{M}_{i,i+1} \in \mathbb{R}^{n_i \times n_{i+1}}$ be the TPM with respect to the set of cells measured at time $t_i$ and at time $t_{i+1}$, with size $n_i$ and $n_{i+1}$ respectively, where the $(k, l)$-th entry of $\mathbf{M}_{i,i+1}$ indicates how plausibly the $k$-th cell measured at $t_i$ will evolve to the $l$-th cell measured at $t_{i+1}$, defined as follows:

$$\left[\mathbf{M}_{i,i+1}\right]_{kl} = \lambda \cdot \mathrm{SS}\left(c_i(k), c_{i+1}(l)\right) + (1 - \lambda) \cdot \mathrm{LR}\left(c_i(k), c_{i+1}(l)\right), \tag{10}$$

where $\lambda \in [0, 1]$ is a trade-off hyperparameter that balances the contribution of the spatial smoothness prior (SS) and the ligand–receptor communication prior (LR).

**Prior-Regularized OT Couplings.** The transitional plausibility matrices capture our spatially informed prior on cell-cell transitions between consecutive time points, which can naturally be

incorporated in the EOT formulation (Equation 6) to promote couplings that maintain the structural and functional properties of the tissue organization. We propose two techniques for prior integration:

*Prior-Aware Cost Matrix (PACM).* Consider the empirical counterpart of Equation 6 with respect to time $t_i$ and time $t_{i+1}$. Our first approach incorporates the transitional plausibility matrix directly into the transport cost:

$$\min_{\mathbf{\Pi} \in \mathbb{R}^{n_i \times n_{i+1}}} \sum_{k,l} \Pi_{kl} \underbrace{\left[ \alpha \cdot \|\boldsymbol{x}_i(k) - \boldsymbol{x}_{i+1}(l)\|_2^2 + (1-\alpha) \cdot [\mathbf{M}_{i,i+1}]_{kl} \right]}_{\text{Prior-Aware Cost Function}} - \epsilon \sum_{k,l} \Pi_{kl} (\log \Pi_{kl} - 1),$$

(11)

where the transport plan $\Pi$ satisfies the boundary conditions: $\sum_l \Pi_{kl} = 1/n_i$ for any $k \in [n_i]$, and $\sum_k \Pi_{kl} = 1/n_{i+1}$ for any $l \in [n_{i+1}]$, and $\alpha \in [0,1]$ controls the trade-off between the original Euclidean cost and the prior-aware cost derived from the transitional plausibility. If $[\mathbf{M}_{i,i+1}]_{kl}$ is high, Equation 11 will impose a higher transport cost between the $k$-cell at time $i$ to the $j$-cell at time $i+1$. This aligns with our assumption that such transitions are implausible.

*Prior-Aware Entropy Regularization (PAER).* While the prior-aware cost matrix approach penalizes couplings in accordance with our spatial priors, it defines a different OT problem characterized by a modified cost function. Consequently, the standard interpretation of OT as minimizing the transport energy between two transcriptomic distributions no longer holds. Since the scales of the pairwise distances often differ, normalization of the cost terms is required to enable meaningful comparison. This normalization, however, may result in couplings that deviate from their original counterparts (Proposition 1 and Corollary 1 in the Appendix C). Besides, selecting an appropriate $\alpha$ in Equation 11 introduces an additional layer of tuning, increasing computational overhead. Therefore, we propose a second approach to integrate the biological priors without introducing additional hyperparameters:

$$\min_{\mathbf{\Pi} \in \mathbb{R}^{n_i \times n_{i+1}}} \sum_{k,l} \Pi_{kl} \|\boldsymbol{x}_i(k) - \boldsymbol{x}_{i+1}(l)\|_2^2 - \epsilon \sum_{k,l} \underbrace{\Pi_{kl} (\log(\Pi_{kl}/[\widehat{\mathbf{M}}_{i,i+1}]_{kl}) - 1)}_{\text{Prior-Aware Entropy Regularization}},$$

(12)

where $[\widehat{\mathbf{M}}_{i,i+1}]_{kl} = \exp(-[\mathbf{M}_{i,i+1}]_{kl})/\sum_l \exp(-[\mathbf{M}_{i,i+1}]_{kl})$ denotes the prior joint probability matrix induced by $\mathbf{M}_{i,i+1}$. Intuitively, the lower the cost $[\mathbf{M}i, i+1]_{kl}$, the larger the entry $[\widehat{\mathbf{M}}i, i+1]_{kl}$, reflecting a higher plausibility of the transition from cell $k$ at $t_i$ to cell $l$ at $t_{i+1}$. The entropy regularization term in Equation 12 thus biases the learned transport plan toward the prior $\widehat{\mathbf{M}}_{i,i+1}$ rather than a uniform baseline, providing a soft mechanism for incorporating biological prior knowledge.

**ContextFlow.** Finally, we apply the Sinkhorn algorithm (Cuturi, 2013) to solve the optimization problem in Equation 11 or Equation 12 to obtain the spatial context-aware EOT couplings, and train the neural velocity vector field $u_\theta$ based on stochastic gradient descent by minimizing the multi-time generalization of Equation 3 with respect to conditionals $p_t(\boldsymbol{x}|\boldsymbol{z})$ and $u_t(\boldsymbol{x}|\boldsymbol{z})$ defined according to Equation 7. The pseudocode for the proposed method, named *Conditional **Flow** Matching with **Context**-Aware OT Couplings* (ContextFlow), is detailed in Algorithm 1 in Appendix D.

In particular, to apply the Sinkhorn algorithm to solve our prior-aware entropy regularization problem in Equation 12, we make use of the following theorem, a generalized result of Peyré et al. (2019).

**Theorem 1.** *Let $\mathbf{C} \in \mathbb{R}^{n_0 \times n_1}$ be a cost matrix and $\mathbf{M} \in \mathbb{R}^{n_0 \times n_1}$ be a prior transition probability matrix. Suppose $\mathbf{\Pi}^*_{\mathrm{CTF-H}}$ is the solution to the following prior-aware optimal transport problem:*

$$\mathbf{\Pi}^*_{\mathrm{CTF-H}} = \mathrm{argmin}_{\mathbf{\Pi} \in \mathbb{R}^{n_0 \times n_1}} \sum_{k,l} \Pi_{kl} C_{kl} + \epsilon \sum_{k,l} \Pi_{kl}(\log(\Pi_{kl}/M_{kl}) - 1),$$

*where $\epsilon > 0$ is the regularization parameter. Then, we can show that $\mathbf{\Pi}^*_{\mathrm{CTF-H}}$ can be computed by Sinkhorn and takes the form $\mathrm{diag}(\boldsymbol{u}) \cdot \mathbf{M} \odot \exp(-\mathbf{C}/\epsilon) \cdot \mathrm{diag}(\boldsymbol{v})$, where $\odot$ denotes element-wise multiplication, and $\boldsymbol{u} \in \mathbb{R}^{n_0}, \boldsymbol{v} \in \mathbb{R}^{n_1}$ are vectors satisfying the marginalization constraints.*

Theorem 1, proven in Appendix B, suggests a new Gibbs kernel $\mathbf{K} = \mathbf{M} \odot \exp(-\mathbf{C}/\epsilon)$, which combines both the transport cost and the prior joint probability matrices. When $\epsilon \to 0$, $\mathbf{\Pi}^*_{\mathrm{CTF}} \to \mathbf{\Pi}^*_{\mathrm{ot}}$, thereby recovering the standard OT couplings in Equation 16. When $\epsilon \to \infty$, the optimal coupling $\mathbf{\Pi}^*_{\mathrm{CTF}} \to \mathrm{diag}(\boldsymbol{u}) \cdot \mathbf{M} \cdot \mathrm{diag}(\boldsymbol{v})$, which corresponds to a plan that aligns with the prior defined by $\mathbf{M}$ rather than the independent couplings obtained with EOT (Section 2.3). This has the same effect as constraining our transport plan through the proposed prior and, by extension, the flow. By varying the parameter $\epsilon$, we can thus efficiently optimize for a desirable coupling via the Sinkhorn algorithm.

Table 1: Interpolation at the middle holdout time point for the Brain Regeneration dataset.

| Sampling | Method | $\lambda$ | $\alpha$ | Weighted $\mathcal{W}_2$ | $\mathcal{W}_2$ | MMD | Energy |
|---|---|---|---|---|---|---|---|
| Next Step | CFM | – | – | $2.618 \pm 0.142$ | $2.579 \pm 0.197$ | $0.043 \pm 0.003$ | $12.505 \pm 1.271$ |
| | MOTFM | – | – | $2.567 \pm 0.088$ | $2.476 \pm 0.161$ | $0.040 \pm 0.003$ | $11.269 \pm 1.388$ |
| | CTF-C | 1 | 0.8 | $2.423 \pm 0.164$ | $2.293 \pm 0.103$ | $0.037 \pm 0.001$ | $9.874 \pm 0.659$ |
| | | 0 | 0.2 | $2.396 \pm 0.028$ | $2.100 \pm 0.102$ | $0.033 \pm 0.003$ | $8.577 \pm 0.976$ |
| | | 0.5 | 0.8 | $2.442 \pm 0.173$ | $2.353 \pm 0.241$ | $0.035 \pm 0.004$ | $9.008 \pm 2.094$ |
| | CTF-H | 0 | – | $2.528 \pm 0.143$ | $2.534 \pm 0.180$ | $0.040 \pm 0.004$ | $11.192 \pm 1.304$ |
| | | 1 | – | $\mathbf{2.316 \pm 0.141}$ | $\mathbf{1.969 \pm 0.221}$ | $\mathbf{0.030 \pm 0.004}$ | $\mathbf{6.359 \pm 1.336}$ |
| | | 0.5 | – | $2.519 \pm 0.167$ | $2.412 \pm 0.158$ | $0.039 \pm 0.004$ | $10.304 \pm 1.808$ |
| IVP | CFM | – | – | $4.216 \pm 0.463$ | $4.266 \pm 0.308$ | $0.170 \pm 0.029$ | $32.413 \pm 5.122$ |
| | MOTFM | – | – | $4.198 \pm 0.319$ | $4.452 \pm 0.243$ | $0.173 \pm 0.017$ | $33.149 \pm 3.321$ |
| | CTF-C | 1 | 0.8 | $3.603 \pm 0.300$ | $3.816 \pm 0.310$ | $0.127 \pm 0.018$ | $24.271 \pm 3.992$ |
| | | 0 | 0.2 | $\mathbf{3.465 \pm 0.232}$ | $\mathbf{3.641 \pm 0.320}$ | $0.119 \pm 0.025$ | $23.055 \pm 5.939$ |
| | | 0.5 | 0.8 | $4.015 \pm 0.351$ | $3.974 \pm 0.442$ | $0.140 \pm 0.038$ | $27.592 \pm 6.669$ |
| | CTF-H | 0 | – | $3.925 \pm 0.267$ | $4.375 \pm 0.297$ | $0.164 \pm 0.013$ | $32.034 \pm 3.270$ |
| | | 1 | – | $3.905 \pm 0.395$ | $4.188 \pm 0.685$ | $\mathbf{0.074 \pm 0.014}$ | $\mathbf{18.728 \pm 2.689}$ |
| | | 0.5 | – | $3.917 \pm 0.343$ | $4.159 \pm 0.455$ | $0.147 \pm 0.022$ | $29.613 \pm 4.822$ |

## 4 EXPERIMENTS

**Datasets.** We evaluate ContextFlow on three longitudinal spatial transcriptomics datasets: Axolotl Brain Regeneration (Wei et al., 2022), Mouse Embryo Organogenesis (Chen et al., 2022), and Liver Regeneration (Ben-Moshe et al., 2021). For all the datasets, the gene expression values are log-normalized, and we extract the top 50 principal components (PCs) as feature vectors. The strength of ligand–receptor interactions in the microenvironment was inferred using spatially informed bivariate statistics implemented in LIANA+ (Dimitrov et al., 2024), where we applied the cosine similarity metric to gene expression profiles. Interaction evidence was aggregated using the consensus of multiple curated ligand–receptor resources, ensuring robustness of the inferred signals.

**Baselines & Metrics.** We benchmark ContextFlow using its two prior integration strategies—cost-regularized (CTF-C) and entropy-regularized (CTF-H)—against several baselines. As a non-spatial baseline, we include conditional flow matching (CFM), which uses only transcriptomic data with random couplings. We further compare against minibatch-OT flow matching (MOTFM), which leverages OT-derived couplings but does not incorporate spatial priors. For evaluation, we employed 2-Wasserstein distance ($\mathcal{W}_2$), a commonly used OT-based metric, and metrics such as MMD and Energy Distance for statistical fidelity. Furthermore, to assess the biological plausibility of our predicted dynamics, we evaluate them using a cell-type-weighted Wasserstein distance (Weighted $\mathcal{W}_2$), where the weights correspond to the relative frequency of each cell type in the dataset. Exact metric definitions are present in the Appendix G. All reported metrics are averaged across 10 runs.

**Sampling.** A trained velocity field can be evaluated through the samples it generates. We consider two variants. *Initial value problem sampling* (IVP) integrates the learned gradient starting from the first observed batch of cells and evolves them toward a later time point. IVP provides the most comprehensive evaluation of flow quality, as errors can accumulate across steps. In contrast, *next-step sampling* (Next Step) integrates the gradient only from the most recently observed batch of cells, thus limiting error propagation but providing a less stringent test of long-term trajectory fidelity.

### 4.1 AXOLOTL BRAIN REGENERATION

We first evaluate ContextFlow on longitudinal Stereo-seq spatial transcriptomic data coming from a post-traumatic brain regeneration study of the Salamander (axolotl telencephalon) species (Wei et al., 2022). The dataset contains samples from five developmental stages, with replicates collected from different individual organisms at each stage. For our CTF-C method, we present the best ablated $\alpha$ in the main text, with full ablation results across different $\alpha$ values provided in Appendix **??**.

For interpolation, we hold out the middle time point during training and evaluate it using samples generated by the trained velocity field $u_\theta$ via both IVP and next-step sampling. Table 1 presents

Table 2: Extrapolation on the last holdout time point for the Brain Regeneration dataset.

| Sampling | Method | $\lambda$ | $\alpha$ | Weighted $\mathcal{W}_2$ | $\mathcal{W}_2$ | MMD | Energy |
|---|---|---|---|---|---|---|---|
| Next Step | CFM | – | – | $7.124 \pm 0.443$ | $7.133 \pm 0.533$ | $0.275 \pm 0.011$ | $76.947 \pm 5.661$ |
| | MOTFM | – | – | $7.619 \pm 0.611$ | $7.769 \pm 0.763$ | $0.272 \pm 0.007$ | $85.352 \pm 8.140$ |
| | CTF-C | 1 | 0.5 | $6.968 \pm 0.608$ | $6.969 \pm 0.628$ | $0.265 \pm 0.009$ | $77.025 \pm 6.056$ |
| | | 0 | 0.5 | $7.244 \pm 0.804$ | $7.146 \pm 0.775$ | $0.265 \pm 0.003$ | $80.424 \pm 10.376$ |
| | | 0.5 | 0.5 | $7.188 \pm 0.391$ | $\mathbf{6.931 \pm 0.260}$ | $0.267 \pm 0.005$ | $78.992 \pm 6.195$ |
| | CTF-H | 0 | – | $\mathbf{6.914 \pm 0.471}$ | $7.198 \pm 0.726$ | $0.266 \pm 0.009$ | $\mathbf{76.149 \pm 8.436}$ |
| | | 1 | – | $7.505 \pm 0.667$ | $7.338 \pm 0.601$ | $\mathbf{0.263 \pm 0.006}$ | $83.425 \pm 8.793$ |
| | | 0.5 | – | $7.243 \pm 0.479$ | $7.157 \pm 0.641$ | $0.270 \pm 0.007$ | $79.826 \pm 8.067$ |
| IVP | CFM | – | – | $6.633 \pm 1.312$ | $7.116 \pm 1.084$ | $0.143 \pm 0.037$ | $60.573 \pm 21.756$ |
| | MOTFM | – | – | $6.503 \pm 0.720$ | $6.352 \pm 0.592$ | $0.162 \pm 0.038$ | $56.452 \pm 15.932$ |
| | CTF-C | 1 | 0.5 | $6.260 \pm 0.616$ | $7.681 \pm 4.003$ | $0.157 \pm 0.039$ | $52.478 \pm 12.010$ |
| | | 0 | 0.5 | $6.614 \pm 0.710$ | $6.854 \pm 0.740$ | $0.201 \pm 0.023$ | $70.370 \pm 9.099$ |
| | | 0.5 | 0.5 | $6.696 \pm 0.427$ | $6.481 \pm 0.387$ | $0.195 \pm 0.024$ | $66.212 \pm 3.542$ |
| | CTF-H | 0 | – | $6.243 \pm 0.760$ | $6.220 \pm 0.751$ | $0.195 \pm 0.020$ | $61.316 \pm 10.288$ |
| | | 1 | – | $\mathbf{5.277 \pm 0.936}$ | $6.021 \pm 1.192$ | $\mathbf{0.099 \pm 0.007}$ | $\mathbf{27.777 \pm 8.621}$ |
| | | 0.5 | – | $6.254 \pm 0.819$ | $\mathbf{5.973 \pm 0.757}$ | $0.156 \pm 0.025$ | $54.330 \pm 12.089$ |

Table 3: Interpolation (time 5) and extrapolation (time 8) results on the Organogenesis dataset.

| Method | $\lambda$ | $\alpha$ | Next Step (Interpolation) | | IVP (Interpolation) | | Next Step (Extrapolation) | |
|---|---|---|---|---|---|---|---|---|
| | | | Weighted $\mathcal{W}_2$ | $\mathcal{W}_2$ | Weighted $\mathcal{W}_2$ | $\mathcal{W}_2$ | Weighted $\mathcal{W}_2$ | $\mathcal{W}_2$ |
| MOTFM | – | – | $1.892 \pm 0.028$ | $1.873 \pm 0.086$ | $3.251 \pm 0.676$ | $3.418 \pm 0.727$ | $1.626 \pm 0.066$ | $1.682 \pm 0.096$ |
| CTF-C | 1 | 0.5 | $\mathbf{1.865 \pm 0.030}$ | $1.852 \pm 0.093$ | $3.137 \pm 0.407$ | $4.093 \pm 1.187$ | $1.685 \pm 0.096$ | $1.714 \pm 0.160$ |
| | 0 | 0.8 | $1.882 \pm 0.022$ | $1.869 \pm 0.049$ | $2.938 \pm 0.476$ | $3.904 \pm 1.120$ | $1.773 \pm 0.053$ | $1.880 \pm 0.180$ |
| | 0.5 | 0.8 | $1.888 \pm 0.033$ | $\mathbf{1.839 \pm 0.134}$ | $3.200 \pm 0.403$ | $3.555 \pm 0.637$ | $1.768 \pm 0.058$ | $1.858 \pm 0.120$ |
| | 1 | 0.2 | $1.880 \pm 0.020$ | $1.922 \pm 0.078$ | $3.260 \pm 0.880$ | $5.264 \pm 3.060$ | $1.683 \pm 0.058$ | $1.803 \pm 0.117$ |
| | 0 | 0.2 | $1.900 \pm 0.035$ | $1.912 \pm 0.057$ | $2.953 \pm 0.425$ | $3.816 \pm 0.970$ | $1.715 \pm 0.123$ | $1.860 \pm 0.267$ |
| CTF-H | 0 | – | $1.884 \pm 0.027$ | $1.862 \pm 0.123$ | $3.244 \pm 0.713$ | $3.946 \pm 1.671$ | $\mathbf{1.505 \pm 0.057}$ | $\mathbf{1.397 \pm 0.088}$ |
| | 1 | – | $1.898 \pm 0.029$ | $1.866 \pm 0.097$ | $5.200 \pm 0.799$ | $6.306 \pm 1.037$ | $1.890 \pm 0.046$ | $1.877 \pm 0.103$ |
| | 0.5 | – | $1.871 \pm 0.030$ | $1.919 \pm 0.067$ | $\mathbf{2.814 \pm 0.414}$ | $\mathbf{3.233 \pm 0.567}$ | $1.636 \pm 0.060$ | $1.684 \pm 0.099$ |

the results. Across multiple evaluation metrics, ContextFlow with entropy regularization (CTF-H) produces trajectories that most closely match the ground truth. CTF-H consistently achieves the best or comparable performance relative to CTF-C, despite the latter being explicitly tuned across multiple $\alpha$ values. This highlights the computational efficiency and superior generalization ability of CTF-H, as it avoids the need for additional hyperparameter tuning while maintaining strong performance.

For extrapolation, we evaluate generation on the last holdout time point, representing the most challenging test of generalizability for the velocity fields $u_\theta$, as it lies outside the training time horizon. As shown in Table 2, CTF-H again consistently achieves the best overall performance, particularly under IVP-Sampling, where errors are most likely to accumulate. This result further reinforces the robustness and reliability of CTF-H across the entire sampling horizon. Finally, Figure 4 (Appendix I.1) demonstrates that incorporating spatial priors enables ContextFlow to produce substantially fewer biologically implausible couplings compared to its context-free counterpart.

## 4.2 MOUSE EMBRYO ORGANOGENESIS

We further evaluated ContextFlow on the larger Mouse Organogenesis Spatiotemporal Atlas (MOSTA) Stereo-seq dataset (Chen et al., 2022) spanning measurements from 8 developmental time points. For the interpolation study of this dataset, we held out time point 5 during training and evaluated its generation during testing. Table 3 shows the evaluation results. We observe that ContextFlow, with both integration strategies, outperforms MOTFM across all metrics, showcasing the effectiveness of the contextual information. While CTF-C shows stronger performance under next-step sampling—albeit only after fine-tuning the trade-off parameter $\alpha$—CTF-H consistently outperforms it in the more challenging IVP-Sampling setting. On the extrapolation task, integrating to the final time point, CTF-H again achieves the strongest performance, underscoring that the entropy-regularized formulation not only removes the need for additional parameter tuning but also offers more robust generalization to unseen temporal horizons.

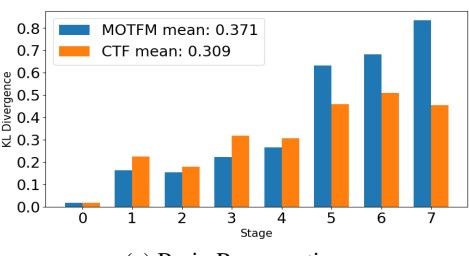 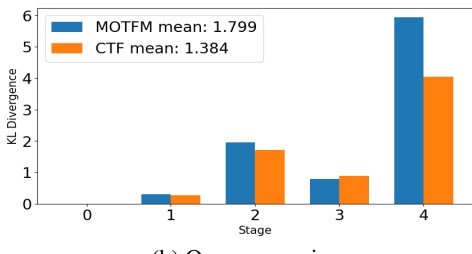

(a) Brain Regeneration            (b) Organogenesis

Figure 2: KL-Divergence between predicted and ground-truth cell type distributions.

Table 4: Interpolation results on the middle holdout time point for the Liver Regeneration dataset.

| | MOTFM | CTF-C | | | CTF-H | | |
|---|---|---|---|---|---|---|---|
| $(\lambda, \alpha)$ | – | (1, 0.5) | (0, 0.5) | (0.5, 0.8) | (0, –) | (1, –) | (0.5, –) |
| $\mathcal{W}_2$ | $34.303 \pm 1.448$ | $33.506 \pm 1.148$ | $32.741 \pm 1.864$ | $33.045 \pm 1.644$ | $\mathbf{32.682 \pm 1.472}$ | $33.481 \pm 1.001$ | $33.414 \pm 0.995$ |

Figure 2 reports the KL-Divergence between normalized histograms of predicted and ground-truth cell types from ContextFlow and MOTFM. In both cases, CTF exhibits lower divergence on average across time points, indicating that the trajectories generated by our model better preserve the biological composition of cell types over time. The cell type progression is further visualized in Figure 10 (Appendix J.6). We show that ContextFlow predicts temporal cell type trajectories that evolve smoothly and consistently across consecutive developmental stages. Early progenitor populations, such as neural crest and mesenchyme, progressively diminish as development advances, while terminal fates, including muscle, cartilage primordium, and liver, emerge at later stages. Major lineages such as brain, heart, and connective tissue remain continuous throughout, demonstrating that ContextFlow captures biologically coherent and temporally consistent developmental dynamics.

## 4.3 LIVER REGENERATION

Finally, we evaluate ContextFlow on a Visium spatial transcriptomics dataset profiling the temporal dynamics of mouse liver regeneration following acetaminophen-induced injury (Ben-Moshe et al., 2021), collected across three distinct regeneration stages. Unlike the earlier datasets resolved at single-cell resolution, Visium data is captured at the level of 55 micron diameter spots, capturing the joint expression of multiple cells. Since direct cell-type information is unavailable, we restrict evaluation to the 2-Wasserstein distance. Moreover, since evaluation is performed on the middle of the three time points, IVP and next-step predictions coincide. Table 4 presents the results. Consistent with the previous findings, CTF-H achieved the lowest reconstruction error, indicating that incorporating contextual information improves trajectory estimation even in aggregated spot-level measurements.

## 5 CONCLUSION

We introduced ContextFlow, a contextually aware flow matching framework that leverages spatial priors and biologically motivated constraints to learn more plausible trajectories from snapshot spatial transcriptomic data, addressing a central challenge of existing methods. The entropic variant of ContextFlow is theoretically grounded, which always yields OT couplings constrained by prior knowledge, promoting stability and consistency with the imposed contextual constraints. Across three diverse datasets, we showed that ContextFlow consistently improves over state-of-the-art baselines even in challenging *Initial Value Problem* sampling settings, underscoring the importance of our contextually informed priors. In addition, we demonstrated that our framework reduces the number of biologically implausible couplings and results in coherent and temporally consistent developmental trajectories while maintaining strong quantitative performance across Wasserstein, MMD, and Energy metrics. These results highlight the value of embedding biological context into generative flow models. Future works can adapt our methods to reconstruct tissues and learn spatial latent dynamics by formulating the flow in space (rather than time), or leverage multi-marginal OT formulations for optimizing temporal flows. Looking forward, ContextFlow offers a principled foundation for modeling perturbations and disease progression, bridging generative power with biological interpretability.

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

Table 5: Comparison of spatiotemporal OT and flow matching methods.

| Method | Generative | Prior Knowledge | Dynamic Gen. | OT Runtime |
|---|---|---|---|---|
| DeST-OT (Halmos et al., 2025) | ✗ | ✗ | ✗ | $O(N^3)$ |
| TOAST (Ceccarelli et al., 2025) | ✗ | ✗ | ✗ | $O(N^3)$ |
| PASTE (Zeira et al., 2022) | ✗ | ✗ | ✗ | $O(N^3)$ |
| CFM (Lipman et al., 2022) | ✓ | ✗ | ✓ | — |
| MOTFM (Tong et al., 2024) | ✓ | ✗ | ✓ | $O(N^2)$ |
| **ContextFlow (ours)** | ✓ | ✓ | ✓ | $O(N^2)$ |

## A  RELATED WORK

### A.1  FLOW MATCHING

Normalizing flows provide a parametric framework for characterizing transformations of a random variable into desired distributions (Papamakarios et al., 2021). These transformations can be realized through either finite (Rezende & Mohamed, 2015) or continuous compositions (Chen et al., 2018). The loss functions used in such formulations typically require computing Jacobians or integrating the flows at each forward pass, making them computationally expensive. Flow matching (FM) (Lipman et al., 2023; Albergo & Vanden-Eijnden, 2022; Liu et al., 2022) addresses this limitation by reducing the training of the velocity field to a regression problem, thereby making normalizing flows substantially more scalable. To ensure valid conditional paths at intermediate time points, samples are coupled either randomly or via optimal transport (Pooladian et al., 2023; Tong et al., 2024). Owing to this scalability, FM has been rapidly adopted across scientific domains, including biology and the life sciences (Li et al., 2025). In transcriptomics, for example, Klein et al. (2024) employed an FM backbone to approximate OT maps for drug response modeling and cross-modal translation tasks. Entropic OT formulations have also been applied to infer cellular trajectories (Tong et al., 2024; Rohbeck et al., 2025), generate imaging-based cell morphology changes (Zhang et al., 2025), and simulate spatial transcriptomics data from histology images (Huang et al., 2025). Despite these advances, existing work does not address how to meaningfully incorporate *biological prior knowledge* to constrain the velocity field, limiting the biological plausibility of inferred trajectories.

### A.2  OPTIMAL TRANSPORT

Omics studies frequently generate uncoupled measurements across conditions, modalities, or time points, which must be integrated into a unified representation to provide a more comprehensive view of the underlying biology. Optimal transport (OT) has recently gained popularity for this task, as it provides a geometry-based approach to couple probability distributions (Bunne et al., 2024; Klein et al., 2025). In spatial transcriptomics (ST), several OT formulations have been introduced depending on context. For instance, Zeira et al. (2022) and Liu et al. (2023) proposed PASTE and PASTE2 to align ST data from adjacent tissue slices, while DeST-OT (Halmos et al., 2025) integrates spatio-temporal slices by modeling cell growth and differentiation. Rahimi et al. (2024) developed DOT, a multi-objective OT framework for mapping features across scRNA-seq and spatially resolved assays, and Ceccarelli et al. (2025) introduced TOAST, a spatially regularized OT framework for slice alignment and annotation transfer. While these methods are primarily designed to *align* biological data across space, time, or modality, they do not address the problem of trajectory inference toward biologically plausible solutions, leveraging biological priors to *constrain* or *bias* the transport plan. Table 5 summarizes the key features of our method, compared with the aforementioned existing work.

## B  PROOFS OF MAIN THEORETICAL RESULTS

**Proposition 1.** *Let* $\mathbf{C} \in \mathbb{R}^{n_0 \times n_1}$ *be a cost matrix and* $\mathbf{M} \in \mathbb{R}^{n_0 \times n_1}$ *a prior transition matrix with positive entries. Consider the entropy-regularized OT formulation:*

$$\Pi^* = \operatorname{argmin}_{\Pi \geq 0} \sum_{k,l} \Pi_{kl} C_{kl} + \epsilon \sum_{k,l} \Pi_{kl}(\log(\Pi_{kl}) - 1).$$

*Let $\tilde{\Pi}^*$ be the EOT-coupling where the cost is scaled by a normalization constant $c$ or $\tilde{C}_{ij} = \frac{C_{ij}}{c}$. Let the regularization parameter $\epsilon > 0$ be the same in both cases. Then, for indices $(i, j)$ and $(k, l)$,*

$$\frac{\tilde{\Pi}^*_{ij}}{\tilde{\Pi}^*_{kl}} \leq \gamma \left(\frac{\Pi^*_{ij}}{\Pi^*_{kl}}\right)^{\frac{1}{c}},$$

*where $\gamma$ depends on $\Pi^*_{ij}, c$ and the OT marginal constraints $a, b$.*

*Proof.* For the original optimal transport (OT) formulation, we note:

$$\Pi^*_{ij} = u_i K_{ij} v_j, \quad K_{ij} = e^{-C_{ij}/\epsilon},$$

with the constraints $\Pi^* \mathbf{1} = a$ and $\Pi^{*\top} \mathbf{1} = b$.

Let

$$\Pi^{*1/c}_{ij} = u_i^{1/c} K_{ij}^{1/c} v_j^{1/c},$$

where:

$$\tilde{K}_{ij} = K_{ij}^{1/c} = \exp\left(-C_{ij}/(c\epsilon)\right)$$

is the kernel for the scaled/normalized OT formulation. Let $\tilde{\Pi}^*_{ij}$ be the coupling for the scaled version, then:

$$\tilde{\Pi}^*_{ij} = \tilde{u}_i \tilde{K}_{ij} \tilde{v}_j.$$

Thus, there exist scaling factors $\alpha_i, \beta_j \in \mathbb{R}$ such that:

$$\tilde{u}_i = \alpha_i u_i^{\frac{1}{c}},$$

$$\tilde{v}_j = \beta_j v_j^{\frac{1}{c}}.$$

This implies:

$$\tilde{\Pi}^*_{ij} = (\alpha_i u_i^{1/c}) \tilde{K}_{ij} (\beta_j v_j^{1/c}),$$

$$\implies \tilde{\Pi}^* = \text{diag}(\alpha u^{1/c}) \tilde{K} \, \text{diag}(\beta v^{1/c}), \tag{g1}$$

$$\implies \tilde{\Pi}^* = \text{diag}(\alpha) \Pi^{1/c} \, \text{diag}(\beta).$$

Subject to the constraints:

$$\sum_i \alpha_i \beta_j \Pi^{*1/c}_{ij} = a_i, \quad \sum_i \alpha_i \beta_j \Pi^{*1/c}_{ij} = b_j.$$

For any pair $(i, j)\&(k, l)$, we can express:

$$\frac{\tilde{\Pi}^*_{ij}}{\tilde{\Pi}^*_{kl}} = \frac{\alpha_i}{\alpha_k} \frac{\beta_j}{\beta_l} \left(\frac{\Pi^*_{ij}}{\Pi^*_{kl}}\right)^{1/c}.$$

Taking logarithms on both sides, we have:

$$\log\left(\frac{\tilde{\Pi}^*_{ij}}{\tilde{\Pi}^*_{kl}}\right) = \log(\alpha_i) - \log(\alpha_k) + \log(\beta_j) - \log(\beta_l) + \frac{1}{c}\log\left(\frac{\Pi^*_{ij}}{\Pi^*_{kl}}\right).$$

Let $\log(\alpha) = \phi$ and $\log(\beta) = \psi$, then:

$$\log\left(\frac{\tilde{\Pi}^*_{ij}}{\tilde{\Pi}^*_{kl}}\right) = (\phi_i - \phi_k) + (\psi_j - \psi_l) + \frac{1}{c}\log\left(\frac{\Pi^*_{ij}}{\Pi^*_{kl}}\right).$$

This implies:

$$\left|\log\left(\frac{\tilde{\Pi}^*_{ij}}{\tilde{\Pi}^*_{kl}}\right) - \frac{1}{c}\log\left(\frac{\Pi^*_{ij}}{\Pi^*_{kl}}\right)\right| \leq |\phi_i| + |\phi_k| + |\psi_j| + |\psi_l|.$$

From Proposition 3 B, we have:

$$\max_i \phi_i \leq E, \quad \max_i \psi_i \leq E.$$

Thus:

$$\left| \log\left( \frac{\tilde{\Pi}^*_{ij}}{\tilde{\Pi}^*_{kl}} \right) - \frac{1}{c} \log\left( \frac{\Pi^*_{ij}}{\Pi^*_{kl}} \right) \right| \leq 4E.$$

Therefore:

$$-4E + \frac{1}{c} \log\left( \frac{\Pi^*_{ij}}{\Pi^*_{kl}} \right) \leq \log\left( \frac{\tilde{\Pi}^*_{ij}}{\tilde{\Pi}^*_{kl}} \right) \leq 4E + \frac{1}{c} \log\left( \frac{\Pi^*_{ij}}{\Pi^*_{kl}} \right).$$

This implies:

$$\frac{\tilde{\Pi}^*_{ij}}{\tilde{\Pi}^*_{kl}} \leq \exp(4E) \left( \frac{\Pi^*_{ij}}{\Pi^*_{kl}} \right)^{1/c}.$$

Let $\gamma = \exp(4E)$, then:

$$\frac{\tilde{\Pi}^*_{ij}}{\tilde{\Pi}^*_{kl}} \leq \gamma \left( \frac{\Pi^*_{ij}}{\Pi^*_{kl}} \right)^{1/c}.$$

$\square$

**Corollary 1.** *Let* $\mathbf{C} \in \mathbb{R}^{n_0 \times n_1}$ *be a cost matrix and* $\mathbf{M} \in \mathbb{R}^{n_0 \times n_1}$ *a prior transition matrix with positive entries. Consider the entropy-regularized OT formulation:*

$$\Pi^* = \operatorname{argmin}_{\Pi \geq 0} \sum_{k,l} \Pi_{kl} C_{kl} + \epsilon \sum_{k,l} \Pi_{kl}(\log(\Pi_{kl}) - 1).$$

*Let* $\tilde{\Pi}^*$ *be the EOT-coupling in the case when cost is scaled by a normalization constant* $c$ *or* $\tilde{C}_{ij} = \frac{C_{ij}}{c}$. *Let the regularization parameter* $\epsilon > 0$ *be the same in both cases. Then:*

$$H(\tilde{\Pi}_{ij}) \geq mH(\Pi_{ij}) - s,$$

*where* $m$ *and* $s$ *are constants that depend on* $\Pi^*$, *the marginalization constants* $a, b$ *and the normalization constant* $c$.

*Proof.* From equation (g1) in Proposition 1 above, we know that:

$$\tilde{\Pi}^*_{ij} = (\Pi^*_{ij})^{1/c} \cdot \exp(\phi_i, \psi_j)$$

and from Proposition 2, we have that,

$$\tilde{\Pi}^*_{ij} \leq (\Pi^*_{ij})^{1/c} \cdot e^{2E}$$

$$\Rightarrow \log(\tilde{\Pi}^*_{ij}) \leq \frac{1}{c} \log(\Pi^*_{ij}) + 2E$$

$$\Rightarrow -\tilde{\Pi}^*_{ij} \log(\tilde{\Pi}^*_{ij}) \geq -\frac{1}{c}(\Pi^*_{ij})^{1/c-1} \cdot \Pi^*_{ij} \log(\Pi^*_{ij}) \cdot e^{2E} - 2E \cdot e^{2E} \cdot (\Pi^*_{ij})^{1/c}$$

For $c \gg 1$, $\frac{1}{c} \to 0$:

$$\Rightarrow -\tilde{\Pi}^*_{ij} \log(\tilde{\Pi}^*_{ij}) \geq -\frac{1}{c\Pi^*_{ij}} \cdot \Pi^*_{ij} \log(\Pi^*_{ij}) \cdot e^{2E} - 2E \cdot e^{2E} \cdot (\Pi^*_{ij})^{1/c}$$

$$\Rightarrow -\tilde{\Pi}^*_{ij} \log(\tilde{\Pi}^*_{ij}) \geq -\frac{1}{c\Pi^*_{\min}} \cdot \Pi^*_{ij} \log(\Pi^*_{ij}) \cdot e^{2E} - 2E \cdot e^{2E} \cdot (\Pi^*_{ij})^{1/c}$$

Summing for all $(i, j)$ we get,

$$H(\tilde{\Pi}^*) \geq mH(\Pi^*) - s,$$

where $m = \frac{e^{2E}}{c\Pi^*_{\min}}$ and $s = 2E \cdot e^{2E}$.

$\square$

**Proposition 2.** *Let $\mathbf{C} \in \mathbb{R}^{n_0 \times n_1}$ be a cost matrix and $\mathbf{M} \in \mathbb{R}^{n_0 \times n_1}$ a prior transition matrix with positive entries. Consider the entropy-regularized OT formulation:*

$$\Pi^* = \operatorname{argmin}_{\Pi \geq 0} \sum_{k,l} \Pi_{kl} C_{kl} + \epsilon \sum_{k,l} \Pi_{kl}(\log(\Pi_{kl}) - 1).$$

*Let $\tilde{\Pi}^*$ be the EOT-coupling in the case when cost is scaled by a normalization constant $c$ or $\tilde{C}_{ij} = \frac{C_{ij}}{c}$. Let the regularization parameter $\epsilon > 0$ be the same in both cases. Consider the scaling factors $\alpha, \beta$ such that: $\tilde{u}_i = \alpha_i u_i^{1/c}$, $\tilde{v}_j = \beta_j v_j^{1/c}$ where $u, v$ are the Sinkhorn algorithm converged vectors for the original setting and $\tilde{u}, \tilde{v}$ are for the cost-scaled version. Then, we have*

$$\max\{\|\phi\|_\infty, \|\psi\|_\infty\} \leq \|M^{-1}\|_\infty \cdot \left\| \begin{pmatrix} \Delta_a \\ \Delta_b \end{pmatrix} \right\|_\infty,$$

*where $\phi = \log(\alpha)$ and $\psi = \log(\beta)$. We also have that,*

$$\max_i |\alpha_i - 1|, \max_i |\beta_i - 1| \leq \|M^{-1}\|_\infty \max(\|\Delta_a\|_\infty, \|\Delta_b\|_\infty),$$

*where $M, \Delta_a, \Delta_b$ depend on $\Pi^*$, marginalization constants $a, b$ and normalization constant $c$.*

*Proof.* Let $X_{ij} = \Pi_{ij}^{*1/c}$ and $X = \Pi^{*1/c}$. Consider the exponentiated versions of $\alpha$ and $\beta$:

$$\phi = \log(\alpha) \in \mathbb{R}^n, \quad \psi = \log(\beta) \in \mathbb{R}^m.$$

From the marginal constraints, we have:

$$\sum_j X_{ij} e^{\phi_i + \psi_j} = a_i, \quad \sum_i X_{ij} e^{\phi_i + \psi_j} = b_j.$$

Applying a first-order Taylor expansion gives:

$$\sum_j X_{ij}(1 + \phi_i + \psi_j) = a_i \implies \sum_j X_{ij}(\phi_i + \psi_j) = a_i - \sum_j X_{ij},$$

$$\sum_i X_{ij}(1 + \phi_i + \psi_j) = b_j \implies \sum_i X_{ij}(\phi_i + \psi_j) = b_j - \sum_i X_{ij}.$$

Define:

$$\Delta_{a_i} = a_i - \sum_j X_{ij}, \quad \Delta_{b_j} = b_j - \sum_i X_{ij}.$$

Thus, we have:

$$\sum_j X_{ij}(\phi_i + \psi_j) = \Delta a_i, \quad \sum_i X_{ij}(\phi_i + \psi_j) = \Delta b_j.$$

This implies:

$$\phi_i \left( \sum_j X_{ij} \right) + \sum_j X_{ij} \psi_j = \Delta a_i,$$

$$\sum_i X_{ij} \phi_i + \psi_j \left( \sum_i X_{ij} \right) = \Delta b_j.$$

Let:

$$D_r = \operatorname{diag}(X\mathbf{1}) \in \mathbb{R}^{n \times n}, \quad D_c = \operatorname{diag}(X^T\mathbf{1}) \in \mathbb{R}^{m \times m}.$$

Then we can express the system as:

$$\begin{pmatrix} D_r & X \\ X^T & D_c \end{pmatrix} \begin{pmatrix} \phi \\ \psi \end{pmatrix} = \begin{pmatrix} \Delta a \\ \Delta b \end{pmatrix}.$$

Let:

$$M = \begin{pmatrix} D_r & X \\ X^T & D_c \end{pmatrix}.$$

Thus:

$$\begin{pmatrix} \phi \\ \psi \end{pmatrix} = M^{-1} \begin{pmatrix} \Delta_a \\ \Delta_b \end{pmatrix}.$$

This implies:

$$\left\| \begin{pmatrix} \phi \\ \psi \end{pmatrix} \right\| \leq \|M^{-1}\| \cdot \left\| \begin{pmatrix} \Delta_a \\ \Delta_b \end{pmatrix} \right\|.$$

Since $\alpha = \exp(\phi)$ and $\beta = \exp(\psi)$, by assumption:

$$|\alpha_i - 1| \approx |\exp(\phi_i) - 1| \approx \phi_i,$$
$$|\beta_j - 1| \approx |\exp(\psi_j) - 1| \approx \psi_j.$$

Therefore:

$$\max_i |\alpha_i - 1|, \max_j |\beta_j - 1| \leq \|M^{-1}\|_\infty \cdot \max(\|\Delta a\|_\infty, \|\Delta b\|_\infty).$$

$\square$

**Theorem 1.** *Let $\mathbf{C} \in \mathbb{R}^{n_0 \times n_1}$ be a general cost matrix and $\mathbf{M} \in \mathbb{R}^{n_0 \times n_1}$ be a prior transition probability matrix. Suppose $\Pi_{\mathrm{CTF-H}}^*$ is the solution to the following prior-aware optimal transport problem:*

$$\Pi_{\mathrm{CTF-H}}^* = \underset{\Pi \in \mathbb{R}^{n_0 \times n_1}}{\arg\min} \sum_{k,l} \Pi_{kl} C_{kl} + \epsilon \sum_{k,l} \Pi_{kl} \log((\Pi_{kl}/M_{kl}) - 1),$$

*where $\epsilon > 0$ is the regularization parameter. Then, we can show that $\Pi_{\mathrm{CTF-H}}^*$ can be computed by the Sinkhorn algorithm and takes the form $\mathrm{diag}(\boldsymbol{u}) \cdot \mathbf{M} \odot \exp(-\mathbf{C}/\epsilon) \cdot \mathrm{diag}(\boldsymbol{v})$, where $\odot$ stands for the elementwise multiplication, and $\boldsymbol{u} \in \mathbb{R}^{n_0}, \boldsymbol{v} \in \mathbb{R}^{n_1}$ are vectors satisfying the marginalization constraints.*

*Proof.* We have that:

$$\Pi_{\mathrm{CTF-H}}^* = \underset{\Pi \in \mathbb{R}^{n_0 \times n_1}}{\arg\min} \sum_{k,l} \Pi_{kl} C_{kl} + \epsilon \sum_{k,l} \Pi_{kl} \log(\Pi_{kl}/M_{kl}),$$

Subject to:

$$\Pi \mathbf{1} = a, \quad \Pi^\top \mathbf{1} = b.$$

This formulation is a standard convex optimization setting with constraints. The Lagrangian of this setting is:

$$\mathcal{L}(\Pi, f, g) = \sum_{k,l} C_{kl} \Pi_{kl} + \epsilon \sum_{k,l} \Pi_{kl} \left( \log\left(\frac{\Pi_{kl}}{M_{kl}}\right) - 1 \right) - \sum_k f_k \left( \sum_l \Pi_{kl} - a_k \right) - \sum_l g_l \left( \sum_k \Pi_{kl} - b_l \right)$$

Differentiating with respect to $\Pi_{kl}, f_k, g_l$, we get:

$$\frac{\partial \mathcal{L}}{\partial \Pi_{kl}} = C_{kl} + \epsilon \log\left(\frac{\Pi_{kl}}{M_{kl}}\right) - f_k - g_l$$

Setting the derivative to zero:

$$\epsilon \log\left(\frac{\Pi_{kl}^*}{M_{kl}}\right) = f_k - C_{kl} + g_l$$

$$\implies \frac{\Pi_{kl}^*}{M_{kl}} = e^{\frac{f_k}{\epsilon}} e^{-\frac{C_{kl}}{\epsilon}} e^{\frac{g_l}{\epsilon}}$$

$$\implies \Pi_{kl}^* = e^{\frac{f_k}{\epsilon}} M_{kl} e^{-\frac{C_{kl}}{\epsilon}} e^{\frac{g_l}{\epsilon}}$$

Let $u \in \mathbb{R}^n$ and $v \in \mathbb{R}^m$ such that:

$$u_k = e^{\frac{f_k}{\epsilon}}, \quad v_l = e^{\frac{g_l}{\epsilon}}$$

Let $K_{kl}$ be the kernel $M_{kl} e^{-C_{kl}/\epsilon}$.

Then, we have:

$$\Pi_{kl}^* = u_k K_{kl} v_l$$

$$\Pi^* = \text{diag}(u) \cdot K \cdot \text{diag}(v) \tag{13}$$

Differentiating the Lagrangian with respect to $f_k$ and $g_l$, we get:

$$\frac{\partial \mathcal{L}}{\partial f_k} = 1 \cdot \left( \sum_l \Pi_{kl}^* - a_k \right) = 0$$

$$\implies \Pi^* \mathbf{1} = a \tag{14}$$

$$\frac{\partial \mathcal{L}}{\partial g_l} = 1 \cdot \left( \sum_i \Pi_{kl}^* - b_l \right) = 0$$

$$\implies \Pi^{*\top} \mathbf{1} = b \tag{15}$$

From equations 16 B, 17 B, and 18 B above, we get:

$$\text{diag}(u) \cdot K \cdot \text{diag}(v) \cdot \mathbf{1} = a$$
$$(\text{diag}(u) \cdot K \cdot \text{diag}(v))^\top \mathbf{1} = b$$

Which can be rewritten as:

$$u \odot (Kv) = a$$
$$K^\top u \odot v = b$$

This is the usual matrix scaling formulation for which the Iterative Proportional Fitting (IPF) updates are:

$$u_k^{t+1} = \frac{a_k}{(Kv^t)_k}, \quad v_l^{t+1} = \frac{b_l}{(K^\top u^{t+1})_l}$$

Sinkhorn Algorithm uses these updates, iteratively, and these updates are shown to converge in Franklin & Lorenz (1989). Thus, Sinkhorn Algorithm can be used for the ContextFlow's Prior Aware Entropy Regularized (*PAER*) (CTF-H) formulation.

From equation (9) B, we get:

$$\Pi_{kl}^* = e^{f_k/\epsilon} M_{kl} e^{-C_{kl}/\epsilon} e^{g_l/\epsilon}$$

When $\epsilon \to \infty$, we have $C_{kl}/\epsilon \to 0$.

$$e^{-C_{kl}/\epsilon} \to 1$$

$$\implies \Pi_{kl}^* \to u_k M_{kl} v_l$$
$$\implies \Pi_{\text{CTF-H}}^* \to \text{diag}(\boldsymbol{u}) \cdot \mathbf{M} \cdot \text{diag}(\boldsymbol{v})$$

Such that marginal constraints, $\Pi_{\text{CTF-H}}^* \mathbf{1} = a$ and $\Pi_{\text{CTF-H}}^{*\top} \mathbf{1} = b$ are satisfied. $\qquad \square$

## C  EFFECTS OF NORMALIZATION ON PRIOR AWARE COST MATRIX

From Peyré et al. (2019), we know that optimal MOTFM coupling takes the form $\Pi^*_{\text{EOT}} = \text{diag}(u) \cdot K \cdot \text{diag}(v)$, where $K$ is the kernel matrix such that $[K]_{ij} = \exp(\frac{-c_{ij}}{\epsilon})$, with $u, v$ satisfying marginalization constraints $u \odot Kv = a$ and $K^T u \odot v = b$. Sinkhorn updates are given by:

$$u^{l+1} = \frac{a}{Kv^l}; v^{l+1} = \frac{b}{K^T u^{l+1}}.$$

In cases where the OT cost function consists of information from different modalities the distances are usually normalized to have distances of a similar scale. Normalizing the cost results $\tilde{c}_{ij} = \frac{c_{ij}}{\epsilon}$ such that the new kernel matrix $[K_{\text{norm}}]_{ij} = \exp(\frac{-c_{ij}}{C_{\max}\epsilon})$ can cause numerical issues if $C_{\max} \gg 1$. The cost normalization should be performed mindfully, when considering different pairwise distances, as in *PACM* Section 3. Intuitively, scaling the cost has the same effect as that of increasing $\epsilon$, making solutions more diffused.

**Proposition 1.** *Let $\mathbf{C} \in \mathbb{R}^{n_0 \times n_1}$ be a cost matrix and $\mathbf{M} \in \mathbb{R}^{n_0 \times n_1}$ a prior transition matrix with positive entries. Consider the entropy-regularized OT formulation:*

$$\Pi^* = \text{argmin}_{\mathbf{\Pi} \geq 0} \sum_{k,l} \Pi_{kl} C_{kl} + \epsilon \sum_{k,l} \Pi_{kl}(\log(\Pi_{kl}) - 1).$$

*Let $\tilde{\Pi}^*$ be the EOT-coupling where the cost is scaled by a normalization constant $c$ or $\tilde{C}_{ij} = \frac{C_{ij}}{c}$. Let the regularization parameter $\epsilon > 0$ be the same in both cases. Then, for any indices $(i,j)$ and $(k,l)$ we have*

$$\frac{\widetilde{\Pi}^*_{ij}}{\widetilde{\Pi}^*_{kl}} \leq \gamma \left( \frac{\Pi^*_{ij}}{\Pi^*_{kl}} \right)^{\frac{1}{c}},$$

*where $\gamma$ depends on $\Pi^*_{ij}, c$ and OT marginal constraints $a, b$.*

From Proposition 1, let $\frac{\Pi^*_{ij}}{\Pi^*_{kl}} = m$, such that $m > 1$ ($\Pi^*_{ij} > \Pi^*_{kl}$ or entries are faraway) then, for $c > 1$, we have $\frac{\tilde{\Pi}^*_{ij}}{\tilde{\Pi}^*_{kl}} < m^{\frac{1}{c}} < m$, for $\gamma < 1$, implying that faraway entries are squeezed together. This results in bringing probabilities that are far apart closer to each other or, in essence, in creating more diffused and less sharp couplings.

**Corollary 1.** *Let $\mathbf{C} \in \mathbb{R}^{n_0 \times n_1}$ be a cost matrix and $\mathbf{M} \in \mathbb{R}^{n_0 \times n_1}$ a prior transition matrix with positive entries. Consider the entropy-regularized OT formulation:*

$$\mathbf{\Pi}^* = \text{argmin}_{\mathbf{\Pi} \geq 0} \sum_{k,l} \Pi_{kl} C_{kl} + \epsilon \sum_{k,l} \Pi_{kl}(\log(\Pi_{kl}) - 1)$$

*and $\tilde{\Pi}^*$ be EOT-coupling in the case when cost is scaled by a normalization constant $c$ or $\tilde{C}_{ij} = \frac{C_{ij}}{c}$. Let the regularization parameter $\epsilon > 0$ be the same in both cases. Then we have:*

$$H(\tilde{\Pi}_{ij}) \geq mH(\Pi_{ij}) - s$$

*where $m$ and $s$ are constants, that depend on $\Pi^*$, marginalization constants $a, b$ and normalization constant $c$.*

Corollary 1 can also be interpreted as supporting the results of Proposition 1 and our intuition that normalizing has the same effect on the kernel matrix as increasing $\epsilon$, leading to more diffused couplings or couplings with increased entropy.

# D CONTEXTFLOW ALGORITHM

---

**Algorithm 1** ContextFlow (CTF): Flow Matching with Spatial-Context-Aware OT Couplings

---

1: **Input:** gene data $\{\mathbf{X}_{t_1}, \cdots, \mathbf{X}_{t_{m+1}}\}$, spatial data $\{\mathbf{S}_{t_1}, \ldots, \mathbf{S}_{t_{m+1}}\}$, parameters $\lambda, \alpha, \epsilon, \sigma, r, \eta$

2: **Data-Preprocessing:** Compute local neighborhood means using Nearest Neighbor Algorithm and Ligand-Receptor features $f_{\text{LR}}$ using LIANA+      ▷ As defined in Equation 8 and 9

3: **Output:** neural velocity vector field $u_\theta$

4: Initialize $\theta$

5: **while** training **do**

6:      **for** $i = 1, 2, \ldots, m$ **do**

7:          Sample a batch $\mathcal{B} = \{(\boldsymbol{x}_i, \boldsymbol{x}_{i+1}) : (\boldsymbol{x}_i, \boldsymbol{x}_{i+1}) \sim (\mathbf{X}_{t_i}, \mathbf{X}_{t_{i+1}})\}$

8:          Construct TPM: $\mathbf{M}_{i,i+1}(\mathcal{B})$          ▷ $\mathbf{M}_{i,i+1}$ is defined in Equation 10

9:          **if** "prior-aware cost matrix" **then**

10:              $C_{kl} \leftarrow \alpha \cdot \|\boldsymbol{x}_i(k) - \boldsymbol{x}_{i+1}(l)\|_2^2 + (1-\alpha) \cdot [\mathbf{M}_{i,i+1}]_{kl}$ for any pair $(k, l)$

11:              $\mathbf{K} \leftarrow \exp(-\mathbf{C}/\epsilon)$

12:          **else if** "prior-aware entropy regularization" **then**

13:              $C_{kl} \leftarrow \|\boldsymbol{x}_i(k) - \boldsymbol{x}_{i+1}(l)\|_2^2$ for any pair $(k, l)$

14:              $\mathbf{K} \leftarrow \widehat{\mathbf{M}}_{i,i+1} \odot \exp(-\mathbf{C}/\epsilon)$          ▷ $\widehat{\mathbf{M}}_{i,i+1}$ is defined in Equation 12

15:          **end if**

16:          Initialize $\boldsymbol{a} \leftarrow \frac{1}{n_i}\mathbf{1}_{n_i}, \boldsymbol{b} \leftarrow \frac{1}{n_{i+1}}\mathbf{1}_{n_{i+1}}, \boldsymbol{u} \leftarrow \mathbf{1}_{n_i}, \boldsymbol{v} \leftarrow \mathbf{1}_{n_{i+1}}$

17:          **while** not converged **do**

18:              $\boldsymbol{u} \leftarrow \boldsymbol{a} \oslash (\mathbf{K}\boldsymbol{v}), \ \boldsymbol{v} \leftarrow \boldsymbol{b} \oslash (\mathbf{K}^\top \boldsymbol{u})$          ▷ Run Sinkhorn algorithm

19:          **end while**

20:          Obtain spatial-prior-aware OT couplings $\mathbf{\Pi}_{i,i+1}^{\text{CTF}} \leftarrow \text{diag}(\boldsymbol{u})\mathbf{K}\,\text{diag}(\boldsymbol{v})$

21:          Sample $t \sim \mathcal{U}(t_i, t_{i+1})$ and $\{(\boldsymbol{x}_i, \boldsymbol{x}_{i+1}) : (\boldsymbol{x}_i, \boldsymbol{x}_{i+1}) \sim \mathbf{\Pi}_{i,i+1}^{\text{CTF}}\}$

22:          Sample $\boldsymbol{x}_t \sim \mathcal{N}\left(\frac{t_{i+1}-t}{t_{i+1}-t_i}\boldsymbol{x}_i + \frac{t-t_i}{t_{i+1}-t_i}\boldsymbol{x}_{i+1}, \sigma^2\mathbf{I}\right)$

23:          $L_{\text{CFM}} \leftarrow \frac{1}{|\mathcal{B}|}\sum_{t,(\boldsymbol{x}_i,\boldsymbol{x}_{i+1})} \left\|u_\theta(\boldsymbol{x}_t, t) - \frac{\boldsymbol{x}_{i+1}-\boldsymbol{x}_i}{t_{i+1}-t_i}\right\|_2^2$

24:      **end for**

25:      $\theta \leftarrow \theta - \eta \cdot \nabla_\theta L_{\text{CFM}}$

26: **end while**

---

# E TIME COMPLEXITY ANALYSIS

The training time of ContextFlow is comparable to that of Minibatch-OT FM (Tong et al., 2024) (Figure 3), as both methods solve an entropic variant of optimal transport using the GPU-optimized Sinkhorn algorithm, alongside forward and backward propagation steps that are also GPU-accelerated. Although ContextFlow incorporates prior knowledge, such as spatial smoothness (Equation 8) and cell-cell communication patterns (Equation 9), their corresponding features are computed once during preprocessing, resulting in a one-time cost. The precomputed features can be reused across multiple hyperparameter settings and model variants, making ContextFlow highly scalable and efficient.

## E.1 DATA PREPROCESSING

The following preprocessing steps generate additional biologically informed features that complement the original transcriptomic profiles. These features incur a one-time computational cost and can be reused across different experiments and model configurations.

**Spatial Smoothness (SS).** We employ a nearest neighbor (NN) algorithm for calculating the mean of local transcriptomic features for each cell. The computational complexity of the NN search is known to be $O(N^2d)$, where $N$ denotes the total points considered, and $d$ represents the data dimension.

**Cell-Cell Communication Patterns (LR).** We employ spatially informed bivariate statistics implemented in LIANA+ (Dimitrov et al., 2024), for computing LR features, where we applied the cosine

Table 6: Runtime for computing cell-cell communication patterns.

| Dataset | Total Number of Cells | Runtime (seconds) |
|---|---|---|
| Brain Regeneration (Wei et al., 2022) | 28,780 | 23.35 |
| Mouse Organogenesis (Chen et al., 2022) | 399,248 | 200.40 |

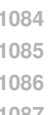

Figure 3: Training time comparisons for different flow matching algorithms with a mini-batch size of 256 on the Brain Regeneration dataset (Wei et al., 2022).

similarity metric to gene expression profiles and used the recommended hyperparameters. The exact runtime complexity for LIANA+ is unknown. Table 6 summarizes the total time taken for the Brain Regeneration and Mouse Organogenesis datasets in our case.

### E.2 TRAINING OF CONTEXTFLOW

The training time complexity largely depends on the total training epochs ($E$), mini-batch size ($B$), the time taken for forward and backward passes ($P$), transcriptomic feature dimension ($d$), and total LR pairs ($l$). Below, we compute the time complexity for each individual step in ContextFlow:

**TPM Construction.** The construction of the transition plausibility matrix involves a calculation of pairwise distances for each mini-batch, resulting in the runtime of $O(B^2(l + d))$.

**Sinkhorn Algorithm.** According to Theorem 1, we know that Sinkhorn iterations can be adapted to solve the prior-aware entropy regularization problem (Equation 12). Since the Sinkhorn algorithm has a well-known quadratic time complexity (Cuturi, 2013), the runtime for computing minibatch OT couplings in ContextFlow is $O(B^2)$.

**Total Runtime.** Putting pieces together, across all the training epochs, the total runtime complexity of ContextFlow turns out to be $O(E \times (B^2(d + l) + P))$. As shown in Figure 3, the runtime is linearly dependent on the total epochs $E$, with different linear rates for different configurations. CFM is the fastest because it bypasses the optimal transport coupling step required by the other methods.

## F KANTAROVICH-OT FORMULATION

Kantorovich's formulation (Peyré et al., 2019) is a classical definition of the *optimal transport* (OT) problem that seeks a joint coupling to move a probability measure to another that minimizes the Euclidean distance cost, corresponding to the following minimization problem with respect to the *2-Wasserstein distance*:

$$\pi_{\text{ot}}^* := \operatorname{argmin}_{\pi \in \Pi(q_0, q_1)} \int_{\mathbb{R}^d \times \mathbb{R}^d} \|\boldsymbol{x}_0 - \boldsymbol{x}_1\|_2^2 \, d\pi(\boldsymbol{x}_0, \boldsymbol{x}_1), \tag{16}$$

where $\Pi(q_0, q_1)$ denotes the set of joint probability measures such that the left and right marginals are $q_0$ and $q_1$. Equation 16 can be solved in a mini-batch fashion using standard solvers such as POT (Flamary et al., 2021); however, the computational complexity is cubic in batch size.

# G EVALUATION METRICS

## G.1 2-WASSERSTEIN

The *2-Wasserstein distance* ($\mathcal{W}_2$ between empirical distributions $\mu, \nu$ is defined as:

$$\mathcal{W}_2(\mu, \nu) = \inf_{\gamma \in \Pi(\mu,\nu)} \left( \sum_{(\boldsymbol{x},\boldsymbol{y})} \gamma(\boldsymbol{x},\boldsymbol{y}) \cdot \|\boldsymbol{x} - \boldsymbol{y}\|_2^2 \right)^{1/2},$$

where $\Pi(\mu, \nu)$ denotes the set of couplings between $\mu$ and $\nu$.

## G.2 WEIGHTED 2-WASSERSTEIN

Implausible velocity fields can steer a cell's transcriptional trajectory in unrealistic directions, potentially leading to entirely different terminal cell types. We thus employ the weighted 2-Wasserstein metric, which ensures the evaluation accounts for both transcriptional similarity and the distributional balance of cell types. We define the *weighted 2-Wasserstein distance* (Weighted $\mathcal{W}_2$) between true and predicted distributions as:

$$\text{Weighted-}\mathcal{W}_2(\mu, \nu) = \sum_{i=1}^{C} \frac{n_i^{\text{true}}}{N} \cdot \mathcal{W}_2 \left( \frac{1}{n_i^{\text{true}}} \sum_{j:y_j=i} \delta_{\boldsymbol{x}_j}, \frac{1}{n_i^{\text{pred}}} \sum_{j:\hat{y}_j=i} \delta_{\boldsymbol{x}_j} \right),$$

where $n_i^{\text{true}}, n_i^{\text{pred}}$ are the number of true and predicted cells of type $i$, and $N$ is the total number of samples. To determine the cell type of generated trajectories, we employ a multi-class classifier $M_\phi$, implemented as an XGBoost model (Chen & Guestrin, 2016) trained for each dataset.

## G.3 ENERGY DISTANCE

Let $\mu$ and $\nu$ be probability distributions with samples $X = \{\boldsymbol{x}_i\}_{i=1}^{m} \sim \mu$ and $Y = \{\boldsymbol{y}_j\}_{j=1}^{n} \sim \nu$. The squared empirical *energy distance* (Energy) is defined as:

$$\text{ED}(\mu, \nu) = \frac{2}{mn} \sum_{i=1}^{m} \sum_{j=1}^{n} \|\boldsymbol{x}_i - \boldsymbol{y}_j\| - \frac{1}{m^2} \sum_{i=1}^{m} \sum_{i'=1}^{m} \|\boldsymbol{x}_i - \boldsymbol{x}_{i'}\| - \frac{1}{n^2} \sum_{j=1}^{n} \sum_{j'=1}^{n} \|\boldsymbol{y}_j - \boldsymbol{y}_{j'}\|,$$

where $\| \cdot \|$ is the Euclidean norm. The distance is non-negative and equals zero if and only if $\mu = \nu$.

## G.4 MAXIMUM MEAN DISCREPANCY

For the same samples, the unbiased empirical estimate of the squared *maximum mean discrepancy* (MMD) with kernel $\kappa$ is defined as:

$$\text{MMD}(\mu, \nu; \kappa) = \frac{1}{m(m-1)} \sum_{i \neq i'} \kappa(\boldsymbol{x}_i, \boldsymbol{x}_{i'}) + \frac{1}{n(n-1)} \sum_{j \neq j'} \kappa(\boldsymbol{y}_j, \boldsymbol{y}_{j'}) - \frac{2}{mn} \sum_{i=1}^{m} \sum_{j=1}^{n} \kappa(\boldsymbol{x}_i, \boldsymbol{y}_j).$$

In our evaluations, we use a multi-kernel variant with radial basis function (RBF) kernels $\kappa_\gamma(\boldsymbol{x}, y) = \exp(-\gamma \|\boldsymbol{x} - \boldsymbol{y}\|^2)$, and average over $\gamma \in [2, 1, 0.5, 0.1, 0.01, 0.005]$.

# H SPATIOTEMPORAL OPTIMAL TRANSPORT

In this section, we compare the recent state-of-the-art spatiotemporal alignment methods, including DeST-OT (Halmos et al., 2025) and TOAST (Ceccarelli et al., 2025), with our prior-aware entropy regularized (PAER) OT objective used in ContextFlow (CTF-H). It is important to note that these OT methods are not generative models and are only used for pairwise alignment tasks. ContextFlow, on the other hand, is a generative model that learns a dynamic flow across the time horizon and utilizes OT couplings to design better conditional paths for regression. In particular, we compute metrics described in DeST-OT on the Axolotl Brain Regeneration dataset following the same setup as used in flow matching. Specifically, for each time step, we randomly sample a batch of 1000 cells and compute the corresponding coupling matrix $\boldsymbol{\Pi}$, which is then used to derive the metrics. We use the CTF-H ($\lambda = 0.8$) version of ContextFlow for comparison.

## H.1 METRIC COMPARISON

DeST-OT introduces an OT objective for aligning spatial transcriptomic tissue slices from different developmental timesteps, with an emphasis on modeling cell growth and tissue expansion/contraction. The growth distortion metric is designed to assess whether the inferred growth pattern aligns with the changes in cell-type abundance across timesteps. As shown in Table 7, for the growth distortion metric, we find that our CTF-H OT is competitive with DeST-OT and TOAST, despite DeST-OT being specifically developed with consideration for cell growth.

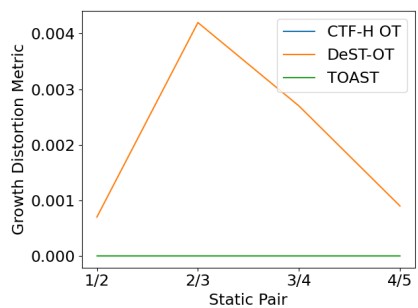

Table 7: Comparison on growth distortion.

| Static Pair | DeST-OT | TOAST | CTF-H OT |
|---|---|---|---|
| 1/2 | 0.0007 | 0.0000 | 0.0000 |
| 2/3 | 0.0042 | 0.0000 | 0.0000 |
| 3/4 | 0.0027 | 0.0000 | 0.0000 |
| 4/5 | 0.0009 | 0.0000 | 0.0000 |

Migration metric is another important metric introduced in DeST-OT, which measures whether the coupling implies realistic cell movements between timesteps. As seen in Table 8, DeST-OT achieves the best performance, highlighting the advantage of its growth-aware objective compared to TOAST and CTF-H OT, which do not explicitly model tissue expansion or contraction.

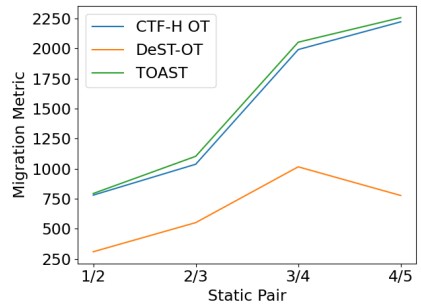

Table 8: Comparison on migration.

| Static Pair | DeST-OT | TOAST | CTF-H OT |
|---|---|---|---|
| 1/2 | 308.97 | 793.71 | 780.16 |
| 2/3 | 551.53 | 1103.29 | 1037.29 |
| 3/4 | 1015.73 | 2052.65 | 1991.43 |
| 4/5 | 777.13 | 2257.02 | 2222.07 |

Lastly, we compute how similar the transcriptomic values of coupled cells are using a *Coupled Transcriptomic Distance* metric, which is defined as $\sum_{k=1}^{N} \sum_{l=1}^{M} \left\| X_{t_i}[k,:] - X_{t_{i+1}}[l,:] \right\|^2 \times \Pi_{i,j}$, where $X_{t_i}[k,:]$ represents the transcriptomic feature of cell $k$ from timestep $t_i$ and $X_{t_{i+1}}[l,:]$ represents the transcriptomic feature of cell $l$ from timestep $t_{i+1}$, and $\Pi$ is the OT coupling matrix. From Table 9, we can observe that CTF-H OT is competitive with both DeST and TOAST.

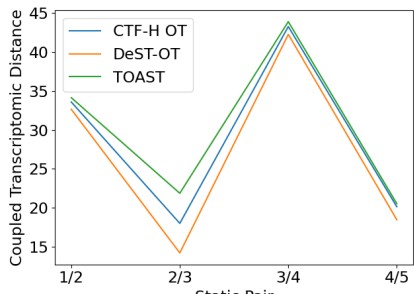

Table 9: On coupled transcriptomic distance.

| Static Pair | DeST-OT | TOAST | CTF-H OT |
|---|---|---|---|
| 1/2 | 32.64 | 34.13 | 33.58 |
| 2/3 | 14.22 | 21.87 | 18.01 |
| 3/4 | 42.26 | 43.89 | 43.26 |
| 4/5 | 18.47 | 20.54 | 20.14 |

## H.2 RUNTIME ANALYSIS WITH VARYING SAMPLE SIZE

We also compare the runtime complexity of the above-mentioned OT methods, with the results shown in Table 10. CTF-H OT is the fastest among the three, followed by DeST-OT and TOAST, while being competitive in the metrics above. We also observe that DeST-OT is the slowest, as expected, since its OT objective involves a Gromov-Wasserstein term, which has an $O(n^3)$ runtime, along with other growth and tissue distortion-specific terms.

Table 10: Runtime (s) with varying sample size.

| Sample Size | DeST-OT | TOAST | CTF-OT |
|---|---|---|---|
| 10 | 0.0111 | 0.0227 | 0.0009 |
| 50 | 0.0342 | 0.0114 | 0.0018 |
| 100 | 0.1892 | 0.0167 | 0.0024 |
| 150 | 0.4035 | 0.0247 | 0.0053 |
| 200 | 0.4913 | 0.0337 | 0.0059 |
| 250 | 0.4571 | 0.0426 | 0.0061 |
| 300 | 0.6252 | 0.0543 | 0.0074 |
| 350 | 0.6974 | 0.0656 | 0.0078 |
| 400 | 0.8612 | 0.0817 | 0.0117 |
| 450 | 1.1028 | 0.0983 | 0.0110 |
| 500 | 1.5478 | 0.1197 | 0.0107 |
| 550 | 1.7448 | 0.1468 | 0.0165 |
| 600 | 1.9295 | 0.1905 | 0.0209 |
| 650 | 2.1282 | 0.2077 | 0.0201 |
| 700 | 2.7013 | 0.2309 | 0.0235 |
| 750 | 3.2951 | 0.2574 | 0.0327 |
| 800 | 4.0964 | 0.3001 | 0.0382 |
| 850 | 4.2001 | 0.3229 | 0.0339 |
| 900 | 4.4582 | 0.3798 | 0.0483 |
| 950 | 5.0965 | 0.4206 | 0.0509 |
| 1000 | 6.1452 | 0.4931 | 0.0375 |

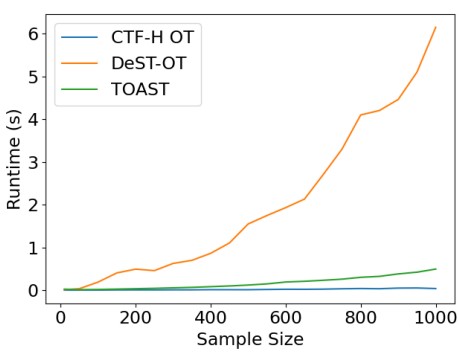

Tables 7-10 demonstrate that the design choices of ContextFlow enable it to be highly scalable compared to existing state-of-the-art spatiotemporal alignment methods, while remaining competitive across several spatiotemporal OT alignment metrics.

## I BIOLOGICAL PRIORS AND DATASET VISUALIZATIONS

### I.1 (IM-)PLAUSIBILITY OF OT-COUPLINGS

To demonstrate the need of integrating biological priors within a generative framework, we computed the Entropic-OT plan (Section 2.3) for the MOTFM framework and the PAER-OT plan (Section 3.3) for the ContextFlow framework. From these transport plans, we sampled couplings corresponding to the first two stages of the Brain Regeneration dataset (Wei et al., 2022) together with their associated cell types. Figures 4a and 4b illustrate the Excitatory–Inhibitory lineage switches present in these sampled couplings. Since excitatory and inhibitory neurons have mutually exclusive neurotransmitter functions and originate from distinct progenitor populations with different transcription factor profiles, a transition from excitatory to inhibitory identity is considered biologically implausible.

In our transport plan couplings, we observed the following cell type lineage switches:

- Immature MSN → Immature nptxEX
- Immature MSN → Immature dpEX
- Immature MSN → Immature CMPN
- Immature nptxEX → Immature cckIN
- Immature nptxEX → Immature MSN

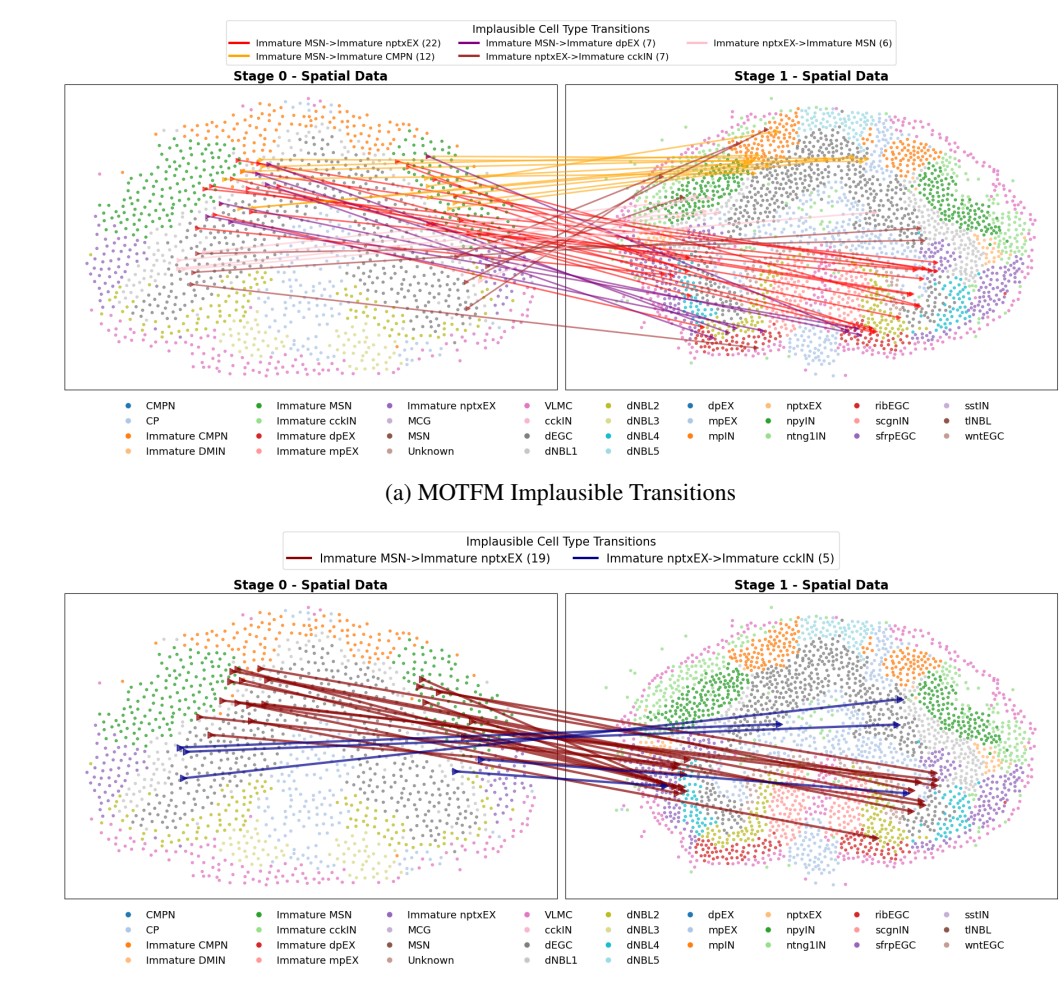

(a) MOTFM Implausible Transitions

(b) ContextFlow Implausible Transitions

Figure 4: Comparison of biologically implausible cell type couplings between Stage 0 and Stage 1 of the Brain Regeneration Dataset (Wei et al., 2022), under the Entropic-OT and ContextFlow Regularized-OT formulations. Biological implausibility is defined here as transitions involving excitatory–inhibitory lineage switches. Our formulation produces substantially fewer biologically implausible couplings (24) compared to MOTFM (54).

Of these, 54 implausible transitions arose from the Entropic-OT plan compared to the 24 under the PAER-OT plan, with the specific transitions detailed in the figure legends. We also observed that the Entropic-OT formulation produced implausible transitions across brain hemispheres, for example, coupling cells from the left hemisphere with those from the right. In contrast, the PAER-OT formulation typically restricted transitions to within the same hemisphere, reflecting its integration of spatially aware contextual information. These observations provide strong motivation for incorporating biological priors through ContextFlow as a principled approach to learning biologically consistent developmental trajectories.

## I.2 CELL TYPE DISTRIBUTIONS OVER TIME

Figures 5–7 present the spatial maps of the transcriptomics datasets across different time points, illustrating how tissue organization and cell type distributions evolve during development and regeneration. These maps highlight not only changes in cellular composition but also the preservation of spatial neighborhoods and geometrical arrangements of specific cell types over time. Such contextual information, specific to spatial transcriptomics, remains inaccessible to standard flow-matching frameworks. By contrast, ContextFlow is designed to exploit these spatial features, enabling the inference of trajectories that are both temporally smooth and spatially coherent.

### I.2.1 BRAIN REGENERATION

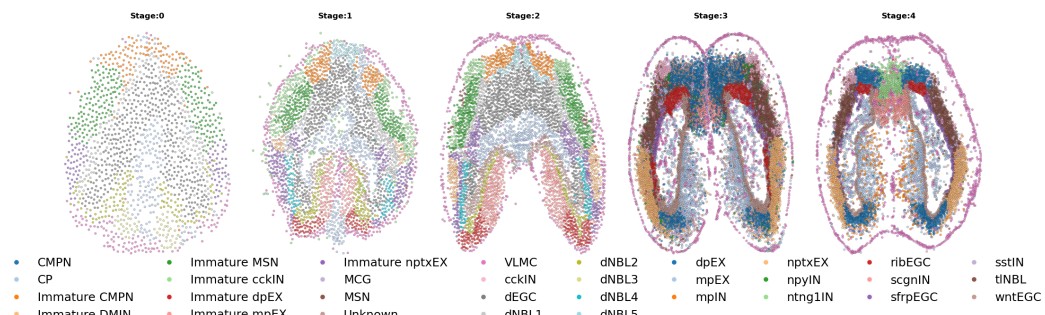

Figure 5: Temporal progression of spatial distribution of different cell types for Brain Regeneration.

### I.2.2 MOUSE EMBRYO ORGANOGENESIS

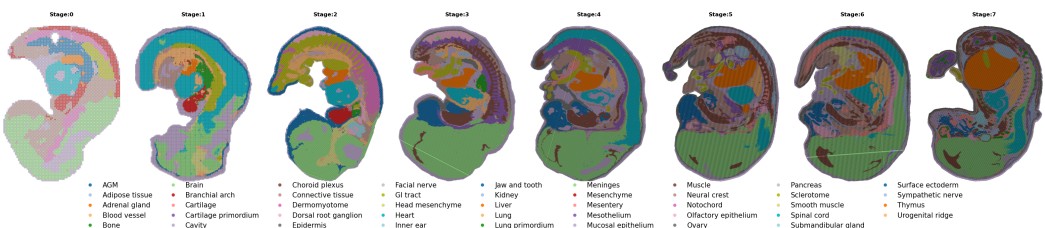

Figure 6: Temporal progression of spatial distribution of different cell types for Mouse Organogenesis.

### I.2.3 LIVER REGENERATION

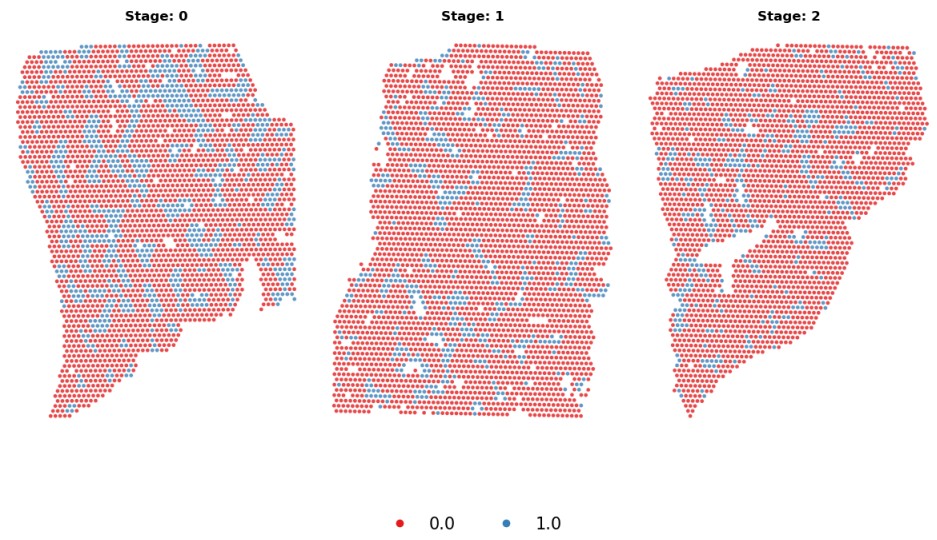

Figure 7: Temporal progression of spatial distribution of fibrogenic states for Liver Regeneration. Here, $0/1$ refers to the absence or presence of fibrogenic spots.

### I.3 LIGAND RECEPTOR INTERACTIONS

Figure 8 shows the ligand-receptor score of the NPTX2-NPTXR pair in two consecutive slides from the Brain regeneration dataset (Wei et al., 2022). Similar activities are visible bilaterally in the

cerebral cortex, suggesting that ligand–receptor interactions are preserved across time and spatially aligned with underlying tissue structure. This observation provides strong evidence that including LR interactions as contextual priors is biologically meaningful, as they capture functional communication signals between cells that remain stable across short time intervals.

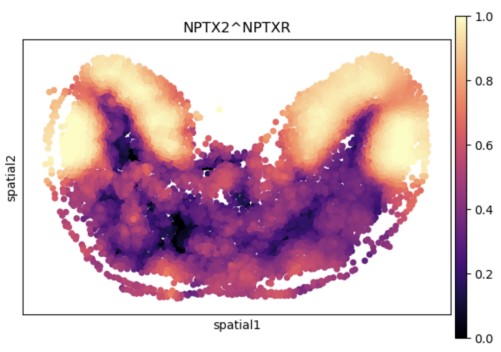
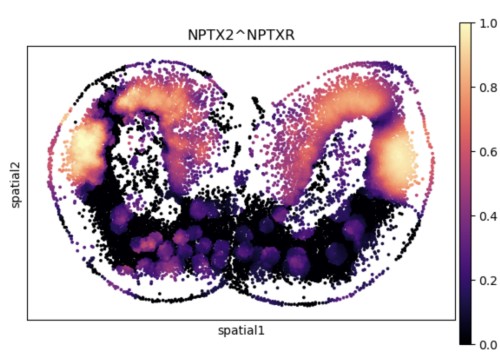

(a) NPTX2-NPTXR LR pair activation on Stage 3     (b) NPTX2-NPTXR LR pair activation on Stage 4

Figure 8: Spatial distributions of LR activation for NPTX2-NPTXR in two consecutive slides from the Brain regeneration dataset. Similar activations are visible at structurally equal positions.

Based on the activation of NPTX2–NPTXR in Figure 8, we observe that the corresponding communication pattern naturally biases the optimal couplings towards transitions such as Immature dpEX → dpEX and Immature nptxEX → nptxEX (Figure 9). These transitions are biologically plausible, as they preserve cell type identity within excitatory neuronal lineages while reflecting maturation within the same functional context. This example highlights the richness of the contextual information captured by our proposed biological prior, and demonstrates how incorporating such ligand–receptor–driven cues into the coupling process leads to more interpretable and biologically consistent trajectories.

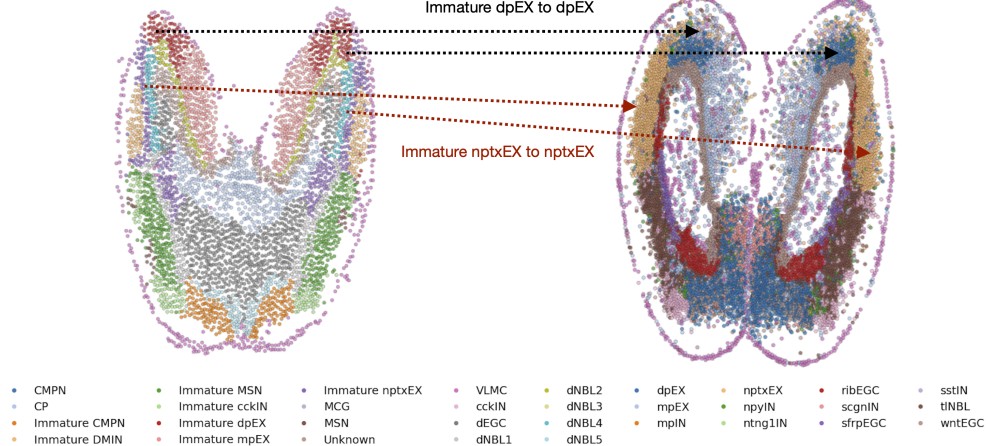

Figure 9: Visual translation of the bias that NPTX2–NPTXR LR pattern provides in terms of cell type coupling for the two consecutive slides.

## J    ADDITIONAL EXPERIMENTS & ABLATIONS

### J.1    NEXT STEP SAMPLING FOR AXOLOTL BRAIN REGENERATION

Table 11: Interpolation via Next Step Sampling at holdout time 3 for the Brain Regeneration dataset.

| Sampling | Method | $\lambda$ | $\alpha$ | Weighted $\mathcal{W}_2$ | $\mathcal{W}_2$ | MMD | Energy |
|---|---|---|---|---|---|---|---|
| | CFM | – | – | $2.618 \pm 0.142$ | $2.579 \pm 0.197$ | $0.043 \pm 0.003$ | $12.505 \pm 1.271$ |
| | MOTFM | – | – | $2.567 \pm 0.088$ | $2.476 \pm 0.161$ | $0.040 \pm 0.003$ | $11.269 \pm 1.388$ |
| | | 1 | 0.2 | $2.503 \pm 0.071$ | $2.425 \pm 0.239$ | $0.037 \pm 0.003$ | $9.868 \pm 1.293$ |
| | | 1 | 0.5 | $2.467 \pm 0.107$ | $2.301 \pm 0.163$ | $0.037 \pm 0.002$ | $9.532 \pm 1.093$ |
| | | 1 | 0.8 | $2.423 \pm 0.164$ | $2.293 \pm 0.103$ | $0.037 \pm 0.001$ | $9.874 \pm 0.659$ |
| | | 0 | 0.2 | $2.396 \pm 0.028$ | $2.100 \pm 0.102$ | $0.033 \pm 0.003$ | $8.577 \pm 0.976$ |
| Next Step | CTF-C | 0 | 0.5 | $2.447 \pm 0.142$ | $2.337 \pm 0.216$ | $0.036 \pm 0.005$ | $9.696 \pm 1.882$ |
| | | 0 | 0.8 | $2.413 \pm 0.099$ | $2.293 \pm 0.161$ | $0.036 \pm 0.002$ | $9.114 \pm 1.092$ |
| | | 0.5 | 0.2 | $2.460 \pm 0.118$ | $2.342 \pm 0.144$ | $0.036 \pm 0.003$ | $9.500 \pm 1.067$ |
| | | 0.5 | 0.5 | $2.504 \pm 0.094$ | $2.309 \pm 0.139$ | $0.036 \pm 0.003$ | $9.394 \pm 1.431$ |
| | | 0.5 | 0.8 | $2.442 \pm 0.173$ | $2.353 \pm 0.241$ | $0.035 \pm 0.004$ | $9.008 \pm 2.094$ |
| | | 0 | – | $2.528 \pm 0.143$ | $2.534 \pm 0.180$ | $0.040 \pm 0.004$ | $11.192 \pm 1.304$ |
| | CTF-H | 1 | – | $\mathbf{2.316 \pm 0.141}$ | $\mathbf{1.969 \pm 0.221}$ | $\mathbf{0.030 \pm 0.004}$ | $\mathbf{6.359 \pm 1.336}$ |
| | | 0.5 | – | $2.519 \pm 0.167$ | $2.412 \pm 0.158$ | $0.039 \pm 0.004$ | $10.304 \pm 1.808$ |

Table 12: Extrapolation via Next Step Sampling at holdout time 5 for the Brain Regeneration dataset.

| Sampling | Method | $\lambda$ | $\alpha$ | Weighted $\mathcal{W}_2$ | $\mathcal{W}_2$ | MMD | Energy |
|---|---|---|---|---|---|---|---|
| | CFM | – | – | $7.124 \pm 0.443$ | $7.133 \pm 0.533$ | $0.276 \pm 0.011$ | $76.947 \pm 5.661$ |
| | MOTFM | – | – | $7.487 \pm 0.698$ | $7.449 \pm 0.931$ | $0.266 \pm 0.010$ | $81.965 \pm 9.812$ |
| | | 1 | 0.2 | $7.257 \pm 0.597$ | $7.077 \pm 0.473$ | $0.257 \pm 0.004$ | $79.562 \pm 7.787$ |
| | | 1 | 0.5 | $6.968 \pm 0.608$ | $6.969 \pm 0.628$ | $0.265 \pm 0.009$ | $77.025 \pm 6.056$ |
| | | 1 | 0.8 | $7.695 \pm 0.443$ | $7.792 \pm 0.463$ | $0.266 \pm 0.007$ | $87.179 \pm 6.690$ |
| | | 0 | 0.2 | $8.170 \pm 0.663$ | $8.079 \pm 0.723$ | $0.269 \pm 0.008$ | $91.572 \pm 8.802$ |
| Next Step | CTF-C | 0 | 0.5 | $7.244 \pm 0.804$ | $7.146 \pm 0.775$ | $0.265 \pm 0.003$ | $80.424 \pm 10.376$ |
| | | 0 | 0.8 | $7.382 \pm 1.068$ | $7.234 \pm 0.852$ | $0.267 \pm 0.009$ | $81.635 \pm 14.135$ |
| | | 0.5 | 0.2 | $7.194 \pm 0.239$ | $7.171 \pm 0.422$ | $0.266 \pm 0.001$ | $78.924 \pm 3.715$ |
| | | 0.5 | 0.5 | $7.188 \pm 0.391$ | $\mathbf{6.931 \pm 0.260}$ | $0.267 \pm 0.005$ | $78.992 \pm 6.195$ |
| | | 0.5 | 0.8 | $7.242 \pm 0.804$ | $7.166 \pm 0.980$ | $0.267 \pm 0.006$ | $80.509 \pm 10.304$ |
| | | 0 | – | $\mathbf{6.914 \pm 0.471}$ | $7.198 \pm 0.726$ | $0.266 \pm 0.009$ | $\mathbf{76.149 \pm 8.436}$ |
| | CTF-H | 1 | – | $7.505 \pm 0.667$ | $7.338 \pm 0.601$ | $\mathbf{0.263 \pm 0.006}$ | $83.425 \pm 8.793$ |
| | | 0.5 | – | $7.243 \pm 0.479$ | $7.157 \pm 0.641$ | $0.270 \pm 0.007$ | $79.826 \pm 8.067$ |

### J.2    IVP SAMPLING ON AXOLOTL BRAIN REGENERATION

Table 13: Interpolation via IVP Sampling at time point 3 for the Brain Regeneration dataset.

| Sampling | Method | $\lambda$ | $\alpha$ | Weighted $\mathcal{W}_2$ | $\mathcal{W}_2$ | MMD | Energy |
|---|---|---|---|---|---|---|---|
| | CFM | – | – | $4.216 \pm 0.463$ | $4.266 \pm 0.308$ | $0.170 \pm 0.029$ | $32.413 \pm 5.122$ |
| | MOTFM | – | – | $4.198 \pm 0.319$ | $4.452 \pm 0.243$ | $0.173 \pm 0.017$ | $33.149 \pm 3.321$ |
| | | 1 | 0.2 | $4.011 \pm 0.276$ | $4.048 \pm 0.321$ | $0.147 \pm 0.021$ | $30.337 \pm 4.713$ |
| | | 1 | 0.5 | $3.932 \pm 0.377$ | $4.356 \pm 0.398$ | $0.156 \pm 0.025$ | $31.524 \pm 4.875$ |
| | | 1 | 0.8 | $3.603 \pm 0.300$ | $3.816 \pm 0.310$ | $0.127 \pm 0.018$ | $24.271 \pm 3.992$ |
| | | 0 | 0.2 | $\mathbf{3.465 \pm 0.232}$ | $\mathbf{3.641 \pm 0.320}$ | $0.119 \pm 0.025$ | $23.055 \pm 5.939$ |
| IVP | CTF-C | 0 | 0.5 | $3.943 \pm 0.413$ | $4.241 \pm 0.435$ | $0.150 \pm 0.039$ | $29.221 \pm 5.713$ |
| | | 0 | 0.8 | $3.881 \pm 0.368$ | $4.094 \pm 0.551$ | $0.139 \pm 0.026$ | $27.941 \pm 6.676$ |
| | | 0.5 | 0.2 | $4.152 \pm 0.341$ | $4.322 \pm 0.291$ | $0.166 \pm 0.014$ | $33.299 \pm 3.629$ |
| | | 0.5 | 0.5 | $4.013 \pm 0.187$ | $4.138 \pm 0.297$ | $0.153 \pm 0.020$ | $30.941 \pm 3.685$ |
| | | 0.5 | 0.8 | $4.015 \pm 0.351$ | $3.974 \pm 0.442$ | $0.140 \pm 0.038$ | $27.592 \pm 6.669$ |
| | | 0 | – | $3.925 \pm 0.267$ | $4.375 \pm 0.297$ | $0.164 \pm 0.013$ | $32.034 \pm 3.270$ |
| | CTF-H | 1 | – | $3.905 \pm 0.395$ | $4.188 \pm 0.685$ | $\mathbf{0.074 \pm 0.014}$ | $\mathbf{18.728 \pm 2.689}$ |
| | | 0.5 | – | $3.917 \pm 0.343$ | $4.159 \pm 0.455$ | $0.147 \pm 0.022$ | $29.613 \pm 4.822$ |

Table 14: Extrapolation via IVP Sampling at holdout time 5 for the Brain Regeneration dataset.

| Sampling | Method | $\lambda$ | $\alpha$ | Weighted $\mathcal{W}_2$ | $\mathcal{W}_2$ | MMD | Energy |
|---|---|---|---|---|---|---|---|
| | CFM | – | – | $6.633 \pm 1.312$ | $7.116 \pm 1.084$ | $0.143 \pm 0.037$ | $60.573 \pm 21.756$ |
| | MOTFM | – | – | $6.503 \pm 0.720$ | $6.352 \pm 0.592$ | $0.162 \pm 0.038$ | $56.452 \pm 15.932$ |
| | | 1 | 0.2 | $6.403 \pm 0.959$ | $6.558 \pm 1.297$ | $0.160 \pm 0.024$ | $61.051 \pm 16.594$ |
| | | 1 | 0.5 | $6.260 \pm 0.616$ | $7.681 \pm 4.003$ | $0.157 \pm 0.039$ | $52.478 \pm 12.010$ |
| | | 1 | 0.8 | $6.875 \pm 0.643$ | $6.920 \pm 0.796$ | $0.159 \pm 0.045$ | $62.838 \pm 16.897$ |
| | | 0 | 0.2 | $6.722 \pm 0.905$ | $6.782 \pm 1.003$ | $0.154 \pm 0.034$ | $53.996 \pm 15.617$ |
| IVP | CTF-C | 0 | 0.5 | $6.614 \pm 0.710$ | $6.854 \pm 0.740$ | $0.201 \pm 0.023$ | $70.370 \pm 9.099$ |
| | | 0 | 0.8 | $6.504 \pm 0.925$ | $6.744 \pm 1.336$ | $0.174 \pm 0.037$ | $56.687 \pm 18.118$ |
| | | 0.5 | 0.2 | $6.514 \pm 0.504$ | $5.998 \pm 0.803$ | $0.155 \pm 0.032$ | $51.329 \pm 15.080$ |
| | | 0.5 | 0.5 | $6.696 \pm 0.427$ | $6.481 \pm 0.387$ | $0.195 \pm 0.024$ | $66.212 \pm 3.542$ |
| | | 0.5 | 0.8 | $6.550 \pm 0.975$ | $6.563 \pm 1.029$ | $0.188 \pm 0.037$ | $63.014 \pm 14.173$ |
| | | 0 | – | $6.243 \pm 0.760$ | $6.220 \pm 0.751$ | $0.195 \pm 0.020$ | $61.316 \pm 10.288$ |
| | CTF-H | 1 | – | $\mathbf{5.277 \pm 0.936}$ | $6.021 \pm 1.192$ | $\mathbf{0.099 \pm 0.007}$ | $\mathbf{27.777 \pm 8.621}$ |
| | | 0.5 | – | $6.254 \pm 0.819$ | $\mathbf{5.973 \pm 0.757}$ | $0.156 \pm 0.025$ | $54.330 \pm 12.089$ |

## J.3 NEXT STEP SAMPLING FOR MOUSE EMBRYO ORGANOGENESIS

Table 15: Interpolation via Next Step Sampling at holdout time 5 for the Mouse Organogenesis dataset.

| Sampling | Method | $\lambda$ | $\alpha$ | Weighted $\mathcal{W}_2$ | $\mathcal{W}_2$ | MMD | Energy |
|---|---|---|---|---|---|---|---|
| | MOTFM | – | – | $1.892 \pm 0.028$ | $1.873 \pm 0.086$ | $0.164 \pm 0.002$ | $11.615 \pm 0.092$ |
| | | 1 | 0.2 | $1.881 \pm 0.020$ | $1.922 \pm 0.078$ | $0.158 \pm 0.003$ | $11.529 \pm 0.197$ |
| | | 1 | 0.5 | $\mathbf{1.865 \pm 0.030}$ | $1.852 \pm 0.093$ | $0.159 \pm 0.001$ | $11.482 \pm 0.108$ |
| | | 1 | 0.8 | $1.889 \pm 0.024$ | $1.888 \pm 0.082$ | $0.161 \pm 0.002$ | $11.552 \pm 0.166$ |
| | | 0 | 0.2 | $1.893 \pm 0.035$ | $1.912 \pm 0.057$ | $0.159 \pm 0.001$ | $11.462 \pm 0.154$ |
| Next Step | CTF-C | 0 | 0.5 | $1.877 \pm 0.039$ | $1.933 \pm 0.088$ | $0.162 \pm 0.002$ | $11.528 \pm 0.110$ |
| | | 0 | 0.8 | $1.882 \pm 0.022$ | $1.869 \pm 0.049$ | $0.161 \pm 0.001$ | $\mathbf{11.399 \pm 0.119}$ |
| | | 0.5 | 0.2 | $1.886 \pm 0.022$ | $1.927 \pm 0.111$ | $\mathbf{0.157 \pm 0.002}$ | $11.430 \pm 0.131$ |
| | | 0.5 | 0.5 | $1.899 \pm 0.027$ | $1.899 \pm 0.072$ | $0.160 \pm 0.002$ | $11.517 \pm 0.097$ |
| | | 0.5 | 0.8 | $1.888 \pm 0.033$ | $\mathbf{1.839 \pm 0.134}$ | $0.161 \pm 0.002$ | $11.475 \pm 0.159$ |
| | | 0 | – | $1.884 \pm 0.027$ | $1.862 \pm 0.123$ | $0.164 \pm 0.001$ | $11.499 \pm 0.123$ |
| | CTF-H | 1 | – | $1.898 \pm 0.029$ | $1.866 \pm 0.097$ | $0.167 \pm 0.002$ | $11.795 \pm 0.170$ |
| | | 0.5 | – | $1.871 \pm 0.030$ | $1.919 \pm 0.067$ | $0.164 \pm 0.002$ | $11.639 \pm 0.182$ |

Table 16: Extrapolation via Next Step Sampling at holdout time 8 for Mouse Organogenesis.

| Sampling | Method | $\lambda$ | $\alpha$ | Weighted $\mathcal{W}_2$ | $\mathcal{W}_2$ | MMD | Energy |
|---|---|---|---|---|---|---|---|
| | MOTFM | – | – | $1.626 \pm 0.066$ | $1.682 \pm 0.096$ | $0.084 \pm 0.007$ | $7.418 \pm 0.749$ |
| | | 1 | 0.2 | $1.683 \pm 0.058$ | $1.803 \pm 0.117$ | $0.087 \pm 0.006$ | $7.830 \pm 0.551$ |
| | | 1 | 0.5 | $1.685 \pm 0.096$ | $1.714 \pm 0.159$ | $0.089 \pm 0.006$ | $8.056 \pm 1.033$ |
| | | 1 | 0.8 | $1.703 \pm 0.063$ | $1.830 \pm 0.131$ | $0.095 \pm 0.005$ | $8.928 \pm 0.723$ |
| | | 0 | 0.2 | $1.715 \pm 0.123$ | $1.860 \pm 0.267$ | $0.094 \pm 0.009$ | $9.021 \pm 1.740$ |
| Next Step | CTF-C | 0 | 0.5 | $1.725 \pm 0.082$ | $1.856 \pm 0.191$ | $0.093 \pm 0.006$ | $8.806 \pm 0.749$ |
| | | 0 | 0.8 | $1.774 \pm 0.053$ | $1.897 \pm 0.175$ | $0.094 \pm 0.007$ | $9.466 \pm 0.957$ |
| | | 0.5 | 0.2 | $1.818 \pm 0.096$ | $2.089 \pm 0.222$ | $0.084 \pm 0.008$ | $8.875 \pm 0.976$ |
| | | 0.5 | 0.5 | $1.774 \pm 0.104$ | $1.899 \pm 0.280$ | $0.093 \pm 0.007$ | $9.139 \pm 1.437$ |
| | | 0.5 | 0.8 | $1.768 \pm 0.058$ | $1.858 \pm 0.120$ | $0.101 \pm 0.006$ | $9.303 \pm 0.634$ |
| | | 0 | – | $\mathbf{1.505 \pm 0.057}$ | $\mathbf{1.397 \pm 0.088}$ | $0.087 \pm 0.005$ | $\mathbf{5.954 \pm 0.492}$ |
| | CTF-H | 1 | – | $1.890 \pm 0.046$ | $1.877 \pm 0.103$ | $0.147 \pm 0.006$ | $10.752 \pm 0.405$ |
| | | 0.5 | – | $1.636 \pm 0.060$ | $1.684 \pm 0.099$ | $\mathbf{0.081 \pm 0.005}$ | $7.088 \pm 0.692$ |

## J.4 IVP SAMPLING FOR MOUSE EMBRYO ORGANOGENESIS

Extrapolating to the last holdout time point of the mouse organogenesis dataset (Chen et al., 2022), particularly under IVP-Sampling, represents the most challenging setting among all our experiments.

This difficulty arises because the target time point lies entirely outside the training horizon, requiring integration from the initial samples through to the end. As a result, the velocity field has more opportunity to drift in incorrect directions, often leading to generations that deviate substantially from the true dynamics. In our experiments, this instability was evident: across 10 runs, several produced highly unstable trajectories, reflecting the sensitivity of the system to initial conditions and numerical solvers. This variability is also captured in the performance metrics reported in Table 18.

Table 17: Interpolation via IVP Sampling at holdout time 5 for the Mouse Organogenesis dataset.

| Sampling | Method | $\lambda$ | $\alpha$ | Weighted $\mathcal{W}_2$ | $\mathcal{W}_2$ | MMD | Energy |
|---|---|---|---|---|---|---|---|
| | MOTFM | – | – | $3.251 \pm 0.676$ | $3.418 \pm 0.727$ | $0.090 \pm 0.003$ | $9.226 \pm 0.648$ |
| IVP | CTF-C | 1 | 0.2 | $3.261 \pm 0.880$ | $5.264 \pm 3.060$ | $0.089 \pm 0.003$ | $10.724 \pm 1.288$ |
| | | 1 | 0.5 | $3.137 \pm 0.407$ | $4.093 \pm 1.187$ | $0.086 \pm 0.004$ | $11.948 \pm 1.393$ |
| | | 1 | 0.8 | $3.392 \pm 0.757$ | $4.716 \pm 2.079$ | $0.089 \pm 0.005$ | $9.547 \pm 0.752$ |
| | | 0 | 0.2 | $2.953 \pm 0.425$ | $3.816 \pm 0.973$ | $0.083 \pm 0.002$ | $9.816 \pm 0.715$ |
| | | 0 | 0.5 | $2.938 \pm 0.476$ | $3.904 \pm 1.120$ | $0.088 \pm 0.005$ | $9.864 \pm 0.764$ |
| | | 0 | 0.8 | $3.101 \pm 0.539$ | $3.855 \pm 0.946$ | $0.087 \pm 0.004$ | $9.280 \pm 0.551$ |
| | | 0.5 | 0.2 | $3.771 \pm 0.862$ | $5.457 \pm 1.704$ | $\mathbf{0.079 \pm 0.004}$ | $9.262 \pm 1.134$ |
| | | 0.5 | 0.5 | $3.090 \pm 0.635$ | $4.596 \pm 2.357$ | $0.084 \pm 0.005$ | $9.786 \pm 1.067$ |
| | | 0.5 | 0.8 | $3.200 \pm 0.403$ | $3.555 \pm 0.637$ | $0.084 \pm 0.004$ | $9.269 \pm 0.541$ |
| | CTF-H | 0 | – | $3.244 \pm 0.713$ | $3.946 \pm 1.671$ | $0.089 \pm 0.005$ | $\mathbf{8.797 \pm 0.612}$ |
| | | 1 | – | $5.200 \pm 0.799$ | $6.306 \pm 1.037$ | $0.123 \pm 0.008$ | $45.862 \pm 13.765$ |
| | | 0.5 | – | $\mathbf{2.814 \pm 0.414}$ | $\mathbf{3.233 \pm 0.567}$ | $0.093 \pm 0.005$ | $10.319 \pm 0.817$ |

Table 18: Extrapolation via IVP Sampling at holdout time 8 for the Mouse Organogenesis dataset.

| Sampling | Method | $\lambda$ | $\alpha$ | Weighted $\mathcal{W}_2$ | $\mathcal{W}_2$ | MMD | Energy |
|---|---|---|---|---|---|---|---|
| | MOTFM | – | – | $110835 \pm 211671$ | $1021005 \pm 2063905$ | $0.086 \pm 0.002$ | $14178 \pm 29475$ |
| IVP | CTF-C | 1 | 0.2 | $785586 \pm 1318212$ | $7598321 \pm 13497483$ | $0.088 \pm 0.002$ | $98199 \pm 150412$ |
| | | 1 | 0.5 | $2691 \pm 3931$ | $28480 \pm 36483$ | $0.087 \pm 0.002$ | $1632 \pm 2090$ |
| | | 1 | 0.8 | $2473 \pm 3349$ | $19537 \pm 26306$ | $0.087 \pm 0.003$ | $517 \pm 616$ |
| | | 0 | 0.2 | $1493 \pm 2497$ | $14563 \pm 24858$ | $0.087 \pm 0.001$ | $800 \pm 1158$ |
| | | 0 | 0.5 | $218018 \pm 471298$ | $1820788 \pm 3994886$ | $0.086 \pm 0.001$ | $2170 \pm 4697$ |
| | | 0 | 0.8 | $12736 \pm 34766$ | $118089 \pm 310135$ | $0.084 \pm 0.002$ | $27013 \pm 60065$ |
| | | 0.5 | 0.2 | $8114720 \pm 16270274$ | $69458305 \pm 140579849$ | $0.088 \pm 0.002$ | $901074 \pm 1775139$ |
| | | 0.5 | 0.5 | $2414338 \pm 6009993$ | $23103811 \pm 56863018$ | $0.086 \pm 0.001$ | $261335 \pm 663279$ |
| | | 0.5 | 0.8 | $1158 \pm 3023$ | $11138 \pm 30025$ | $0.084 \pm 0.002$ | $445 \pm 1085$ |
| | CTF-H | 0 | – | $353428 \pm 952168$ | $3011396 \pm 8057131$ | $0.095 \pm 0.004$ | $22990 \pm 58936$ |
| | | 1 | – | $\mathbf{15 \pm 10}$ | $\mathbf{53 \pm 53}$ | $0.098 \pm 0.006$ | $\mathbf{48 \pm 32}$ |
| | | 0.5 | – | $107889 \pm 275882$ | $994606 \pm 2772756$ | $0.087 \pm 0.002$ | $8875 \pm 24264$ |

## J.5 LIVER REGENERATION

Table 19: Wasserstein distances for different model configurations

| Variant | $\lambda$ | $\alpha$ | $\mathcal{W}_2$ |
|---|---|---|---|
| EOT | – | – | $34.30348 \pm 1.44797$ |
| CTF-C | 1 | 0.2 | $34.44455 \pm 1.19306$ |
| CTF-C | 1 | 0.5 | $33.95671 \pm 1.64415$ |
| CTF-C | 1 | 0.8 | $34.62812 \pm 0.98181$ |
| CTF-C | 0 | 0.2 | $34.24147 \pm 1.16930$ |
| CTF-C | 0 | 0.5 | $32.74147 \pm 1.86351$ |
| CTF-C | 0 | 0.8 | $33.71729 \pm 1.23057$ |
| CTF-C | 0.5 | 0.2 | $33.56646 \pm 1.04376$ |
| CTF-C | 0.5 | 0.5 | $33.84199 \pm 1.71408$ |
| CTF-C | 0.5 | 0.8 | $33.04534 \pm 1.64399$ |
| CTF-H | 0 | – | $\mathbf{32.68215 \pm 1.47185}$ |
| CTF-H | 1 | – | $33.48050 \pm 1.00149$ |
| CTF-H | 0.5 | – | $33.41444 \pm 0.99501$ |

## J.6 IVP CELL TYPE PROGRESSION OVER TIME

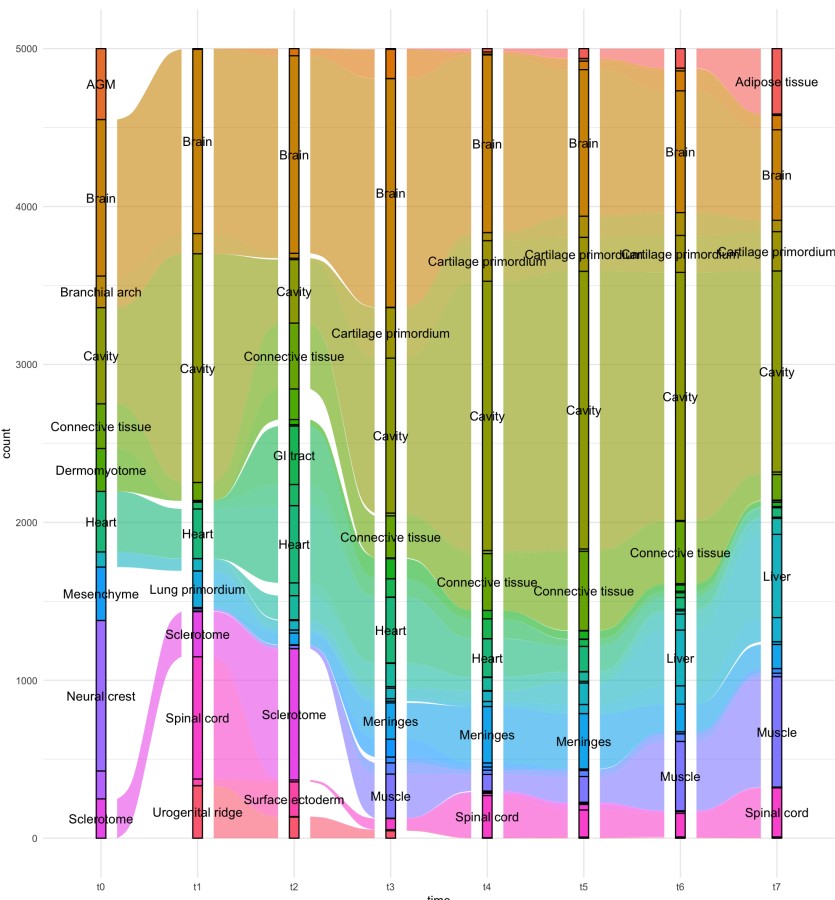

Figure 10: Temporal cell type predictions from ContextFlow for the major cell types in the Organogenesis dataset (Chen et al., 2022). Early progenitor populations (neural crest and mesenchyme) progressively diminish as development advances, while terminal fates (muscle, cartilage primordium, and liver) emerge at later stages. Major lineages such as brain, heart, and connective tissue remain continuous throughout. Overall, ContextFlow captures biologically coherent and temporally consistent developmental dynamics.

## K HYPERPARAMETER SENSITIVITY ANALYSIS

### K.1 ABLATION ON $\lambda$

First, we conduct a sensitivity analysis on all three datasets for the trade-off parameter $\lambda$, which controls the relative importance of spatial smoothness (SS) and cell-cell communication (LR) priors. All the other hyperparameters are kept constant. From Tables 20-24 and their corresponding Figures 11-15, we observe that the best performance is usually achieved towards the extremities, at $\lambda = 0$ or $\lambda = 1$, with values near the latter dominating more often. We hypothesize that the SS prior (at $\lambda = 1$ only the SS prior is considered), acting as a proxy for spatial distance between cells from different slices, always carries relevant information, encoding the structural information present in the data. On the other hand, the informativeness of cell–cell communication patterns depends on how distinct the ligand–receptor features are at a given time step compared with those of neighboring ones. When LR features remain highly similar across consecutive timesteps, they contribute little to the discriminative signal that the OT objective can leverage. Consequently, the influence of communication priors is strongly dataset- and timestep-dependent. This effect is also reflected in the observation that

settings with $\lambda = 0$ (where only the LR prior is used) tend to perform worse when using Next-Step sampling—where local, immediate effects dominate—than under IVP sampling, which integrates information over the entire preceding trajectory.

We therefore recommend experimenting with different values of $\lambda$ (e.g., 0, 0.8, or 1) depending on the specific use case and context. Due to the scalability of ContextFlow, hyperparameter exploration can be performed efficiently, allowing for a rapid assessment of the effect of $\lambda$ on model performance.

### K.2 ABLATION ON $r$

We conduct an ablation study on the Brain Regeneration dataset, examining the effect of varying the neighborhood radius $r$ used to define the boundary for computing the Spatial Smoothness Prior. We evaluate two settings: $\lambda = 1$ and $\lambda = 0.8$, corresponding to the use of only the spatial prior and to a setting with a modest contribution from the cell–cell communication prior, respectively. From Tables 25-28 and their corresponding Figures 16-19, we observe that the optimal neighborhood radius tends to lie toward the lower end of the tested range. Radii smaller than this optimum degrade performance by failing to capture sufficient local context, resulting in neighborhood means that are overly similar to individual cellular profiles. Conversely, increasing the radius beyond the optimal range also reduces performance, as the neighborhood begins to include cells from distinct types or spatial regions, thereby diluting the local signal. While certain deviations from this trend occur, likely reflecting underlying biological complexity, this behavior is consistent with the trade-off between spatial specificity and contextual coverage inherent to neighborhood-based priors.

For our case, we set the radius by considering the timestep with the least number of cells, dividing it by half (to account for different hemispheres), and dividing by the order of cell types present in that timestep. For the dataset considered in this study, Stage 44 had the fewest number of 1400 cells, with approximately 10 cell types present. We thus set the radius at 50 in our studies.

### K.3 ABLATION ON $\epsilon$

Additionally, we conduct an ablation study on the Brain Regeneration dataset by varying the parameter $\epsilon$, which is used to weigh the entropic term in the OT objective. The ContextFlow configuration we consider here is CTF-H ($\lambda = 1$), which includes only the spatial smoothness prior. As observed from Tables 29-30 and their corresponding Figures 20-21, drastically decreasing $\epsilon$ results in the OT formulation to ignore the relative entropic term containing the prior information and only to consider the transport cost resulting in higher Wasserstein values. Furthermore, in accordance with results from Theorem 1, increasing $\epsilon$ too much still does not drastically degrade the performance, as the prior matrix $M$ acts as a soft filter and prohibits uniform couplings.

While setting $\epsilon$ values, one must look at the Gibbs kernel used in the Sinkhorn Algorithm $\exp(-\mathbf{C}/\epsilon)$, since lower $\epsilon$ values can cause potential numerical issues. We thus set $\epsilon$ by examining the order of the cost matrix $\mathbf{C}$, and for the studies above, we set it to 100 after considering the order of the median of all the elements in the cost matrix.

Table 20: Extrapolation on the last holdout timestep on the Brain Regeneration dataset.

| $\lambda$ | Next Step Sampling | | IVP Sampling | |
|---|---|---|---|---|
| | Weighted $\mathcal{W}_2$ | $\mathcal{W}_2$ | Weighted $\mathcal{W}_2$ | $\mathcal{W}_2$ |
| 0 | $6.968 \pm 0.608$ | $7.198 \pm 0.726$ | $6.243 \pm 0.760$ | $6.220 \pm 0.751$ |
| 0.2 | $7.313 \pm 0.384$ | $7.331 \pm 0.467$ | $6.502 \pm 0.634$ | $6.039 \pm 0.733$ |
| 0.5 | $7.243 \pm 0.479$ | $7.157 \pm 0.641$ | $6.254 \pm 0.819$ | $5.973 \pm 0.757$ |
| 0.8 | $7.333 \pm 0.605$ | $7.334 \pm 0.622$ | $6.598 \pm 0.892$ | $6.402 \pm 1.039$ |
| 1 | $7.505 \pm 0.667$ | $7.338 \pm 0.601$ | $5.277 \pm 0.936$ | $6.021 \pm 1.192$ |

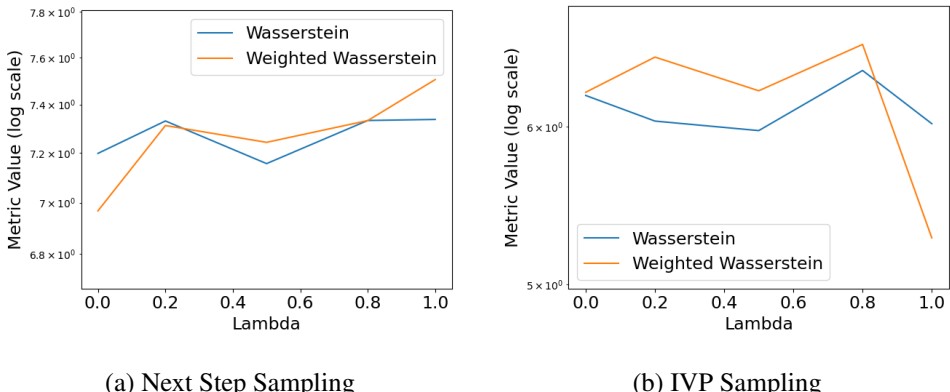

(a) Next Step Sampling   (b) IVP Sampling

Figure 11: Performance variation with $\lambda$ for extrapolation on the Brain Regeneration dataset.

Table 21: Interpolation on the middle holdout timestep 3 on the Brain Regeneration dataset.

| $\lambda$ | Next Step Sampling | | IVP Sampling | |
|---|---|---|---|---|
| | Weighted $\mathcal{W}_2$ | $\mathcal{W}_2$ | Weighted $\mathcal{W}_2$ | $\mathcal{W}_2$ |
| 0 | $2.528 \pm 0.143$ | $2.534 \pm 0.180$ | $3.925 \pm 0.267$ | $4.375 \pm 0.297$ |
| 0.2 | $2.544 \pm 0.093$ | $2.389 \pm 0.183$ | $4.153 \pm 0.432$ | $4.393 \pm 0.369$ |
| 0.5 | $2.519 \pm 0.167$ | $2.412 \pm 0.158$ | $3.917 \pm 0.343$ | $4.159 \pm 0.455$ |
| 0.8 | $2.533 \pm 0.137$ | $2.352 \pm 0.142$ | $4.151 \pm 0.193$ | $4.408 \pm 0.285$ |
| 1 | $2.316 \pm 0.141$ | $1.969 \pm 0.221$ | $3.905 \pm 0.395$ | $4.188 \pm 0.685$ |

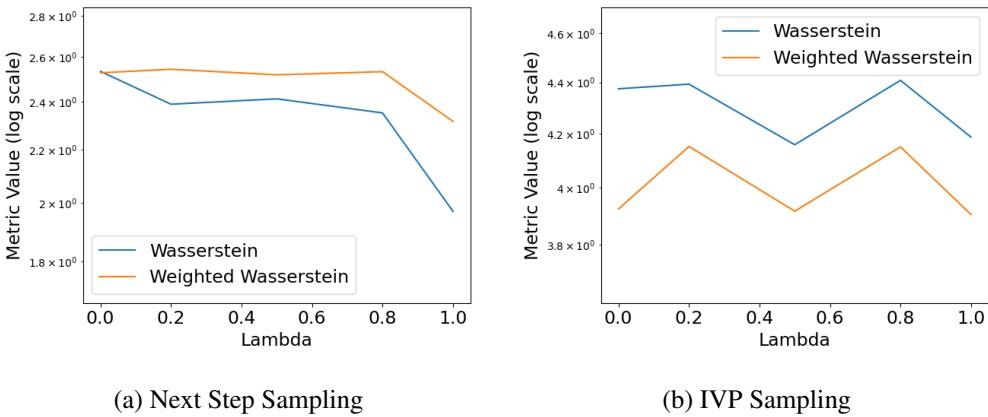

(a) Next Step Sampling   (b) IVP Sampling

Figure 12: Performance variation with $\lambda$ for interpolation on the Brain Regeneration dataset.

Table 22: Interpolation for the holdout timestep 5 on the Mouse Organogenesis dataset.

| $\lambda$ | Next Step Sampling | | IVP Sampling | |
|---|---|---|---|---|
| | Weighted $\mathcal{W}_2$ | $\mathcal{W}_2$ | Weighted $\mathcal{W}_2$ | $\mathcal{W}_2$ |
| 0 | $1.884 \pm 0.027$ | $1.862 \pm 0.123$ | $3.244 \pm 0.713$ | $3.946 \pm 1.671$ |
| 0.2 | $1.896 \pm 0.028$ | $1.899 \pm 0.078$ | $2.990 \pm 0.205$ | $3.273 \pm 0.518$ |
| 0.5 | $1.871 \pm 0.030$ | $1.919 \pm 0.067$ | $2.814 \pm 0.414$ | $3.233 \pm 0.567$ |
| 0.8 | $1.878 \pm 0.031$ | $1.890 \pm 0.064$ | $2.966 \pm 0.411$ | $3.345 \pm 0.508$ |
| 1 | $1.898 \pm 0.029$ | $1.866 \pm 0.097$ | $5.200 \pm 0.799$ | $6.306 \pm 1.037$ |

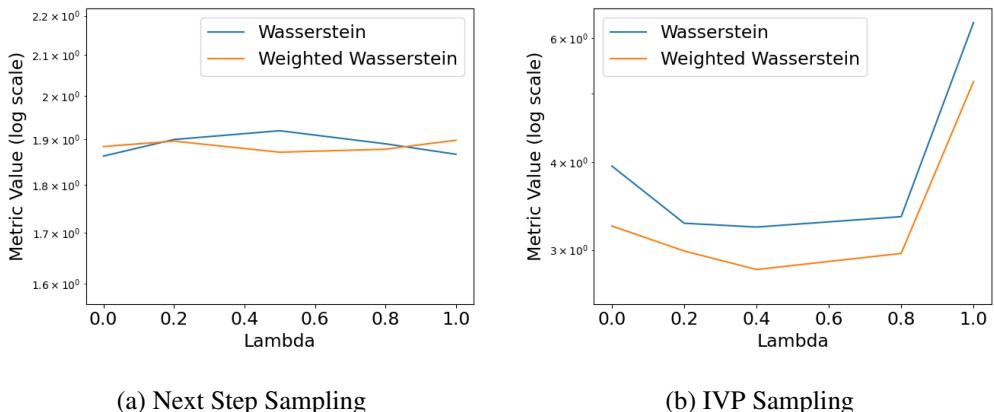

(a) Next Step Sampling               (b) IVP Sampling

Figure 13: Performance variation with $\lambda$ for interpolation on the Mouse Organogenesis dataset.

Table 24: Interpolation for holdout timestep 3 with IVP Sampling on the Liver Regeneration dataset.

| $\lambda$ | $\mathcal{W}_2$ |
|---|---|
| 0 | $32.682 \pm 1.472$ |
| 0.2 | $34.647 \pm 1.461$ |
| 0.5 | $33.414 \pm 0.995$ |
| 0.8 | $33.512 \pm 0.786$ |
| 1 | $33.481 \pm 1.001$ |

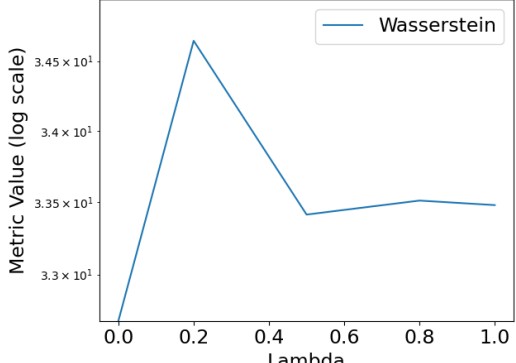

Figure 15: Performance variation with $\lambda$ for interpolation on the Liver Regeneration dataset.

Table 23: Extrapolation for holdout timestep 5 on the Mouse Organogenesis dataset.

| $\lambda$ | Next Step Sampling | | IVP Sampling | |
|---|---|---|---|---|
| | Weighted $\mathcal{W}_2$ | $\mathcal{W}_2$ | Weighted $\mathcal{W}_2$ | $\mathcal{W}_2$ |
| 0 | $1.508 \pm 0.047$ | $1.386 \pm 0.088$ | $6.33 \times 10^5 \pm 1.72 \times 10^6$ | $3.95 \times 10^6 \pm 1.05 \times 10^7$ |
| 0.2 | $1.614 \pm 0.081$ | $1.642 \pm 0.136$ | $9.18 \times 10^2 \pm 1.69 \times 10^3$ | $4.29 \times 10^4 \pm 1.05 \times 10^5$ |
| 0.5 | $1.638 \pm 0.069$ | $1.676 \pm 0.114$ | $1.08 \times 10^5 \pm 2.76 \times 10^5$ | $9.95 \times 10^5 \pm 2.77 \times 10^6$ |
| 0.8 | $1.617 \pm 0.042$ | $1.680 \pm 0.094$ | $1.66 \times 10^5 \pm 3.92 \times 10^5$ | $1.43 \times 10^6 \pm 3.43 \times 10^6$ |
| 1 | $1.906 \pm 0.071$ | $1.892 \pm 0.092$ | $1.89 \times 10^1 \pm 1.77 \times 10^1$ | $6.89 \times 10^1 \pm 7.36 \times 10^1$ |

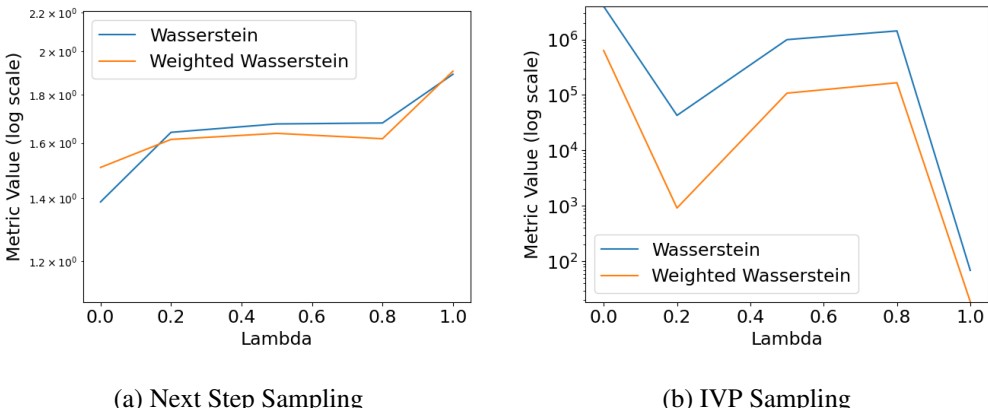

(a) Next Step Sampling        (b) IVP Sampling

Figure 14: Performance variation with $\lambda$ for extrapolation on the Mouse Organogenesis dataset.

Table 25: Extrapolation with CTF-H at $\lambda = 1$ (only using the spatial smoothness prior) for the last holdout timestep on the Brain Regeneration dataset.

| Radius | Next Step Sampling | | IVP Sampling | |
|---|---|---|---|---|
| | Weighted $\mathcal{W}_2$ | $\mathcal{W}_2$ | Weighted $\mathcal{W}_2$ | $\mathcal{W}_2$ |
| 12 | $6.228 \pm 1.276$ | $6.163 \pm 1.490$ | $4.415 \pm 0.580$ | $6.843 \pm 4.812$ |
| 25 | $6.244 \pm 1.066$ | $6.231 \pm 1.043$ | $6.500 \pm 1.751$ | $5.613 \pm 1.561$ |
| 50 | $7.505 \pm 0.667$ | $7.338 \pm 0.601$ | $5.277 \pm 0.936$ | $6.021 \pm 1.192$ |
| 100 | $6.892 \pm 0.930$ | $6.702 \pm 0.631$ | $7.061 \pm 1.677$ | $6.860 \pm 1.880$ |
| 150 | $7.747 \pm 0.923$ | $7.793 \pm 0.934$ | $9.796 \pm 3.847$ | $10.656 \pm 6.591$ |
| 200 | $6.039 \pm 0.282$ | $5.764 \pm 0.272$ | $5.630 \pm 0.793$ | $5.000 \pm 0.735$ |
| 250 | $6.804 \pm 1.011$ | $6.834 \pm 1.124$ | $6.578 \pm 1.611$ | $7.379 \pm 2.864$ |

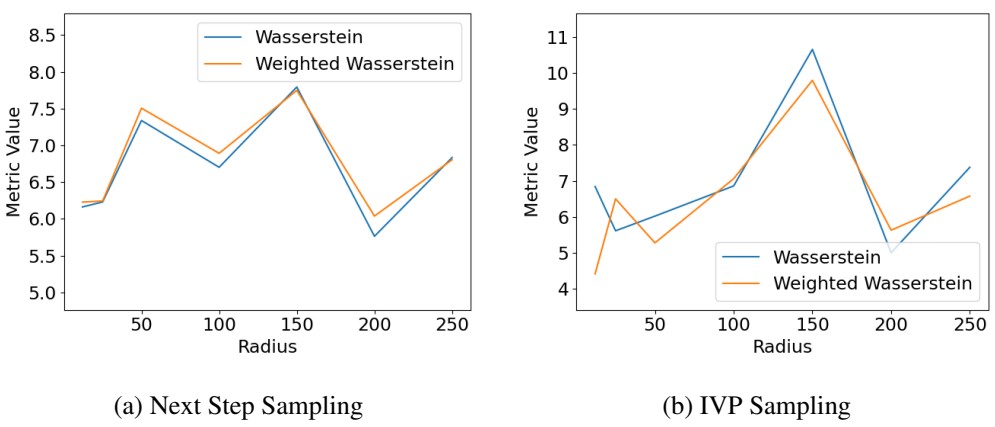

(a) Next Step Sampling        (b) IVP Sampling

Figure 16: Performance variation with radius for extrapolation on the Brain Regeneration dataset.

Table 26: Interpolation for the middle holdout timestep 3 for CTF-H at $\lambda = 1$ (only using the spatial smoothness prior) on the Brain Regeneration dataset.

| Radius | Next Step Sampling | | IVP Sampling | |
| --- | --- | --- | --- | --- |
| | Weighted $\mathcal{W}_2$ | $\mathcal{W}_2$ | Weighted $\mathcal{W}_2$ | $\mathcal{W}_2$ |
| 12 | $4.293 \pm 0.318$ | $3.547 \pm 0.343$ | $2.650 \pm 0.204$ | $2.346 \pm 0.251$ |
| 25 | $5.019 \pm 0.270$ | $3.968 \pm 0.274$ | $2.408 \pm 0.239$ | $1.808 \pm 0.257$ |
| 50 | $2.316 \pm 0.141$ | $1.969 \pm 0.221$ | $3.905 \pm 0.395$ | $4.188 \pm 0.685$ |
| 100 | $4.590 \pm 0.360$ | $3.359 \pm 0.166$ | $2.812 \pm 0.240$ | $2.220 \pm 0.231$ |
| 150 | $4.731 \pm 0.424$ | $3.819 \pm 0.239$ | $3.533 \pm 0.220$ | $3.290 \pm 0.778$ |
| 200 | $4.548 \pm 0.780$ | $4.249 \pm 1.315$ | $3.751 \pm 0.725$ | $3.677 \pm 1.016$ |
| 250 | $4.768 \pm 1.994$ | $4.782 \pm 4.129$ | $4.281 \pm 0.985$ | $4.103 \pm 1.081$ |

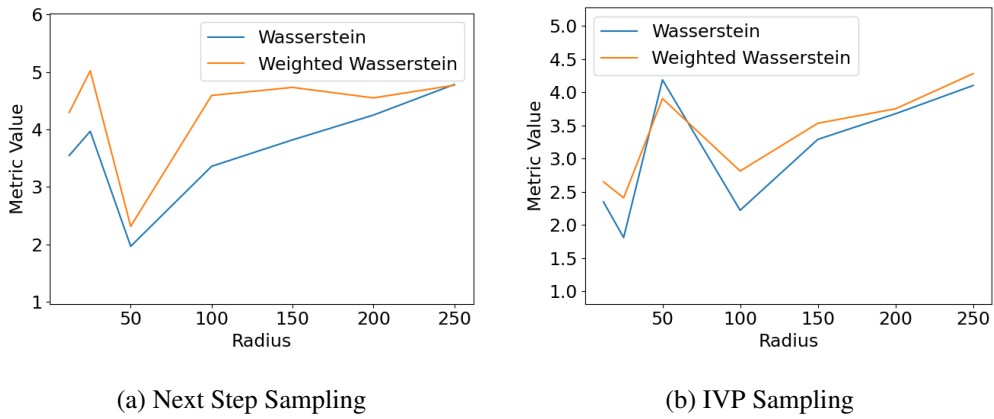

(a) Next Step Sampling

(b) IVP Sampling

Figure 17: Performance variation with radius for interpolation on the Brain Regeneration dataset.

Table 27: Extrapolation for the last holdout timestep with CTF-H at $\lambda = 0.8$ on the Brain Regeneration dataset.

| Radius | Next Step Sampling | | IVP Sampling | |
|---|---|---|---|---|
| | Weighted $\mathcal{W}_2$ | $\mathcal{W}_2$ | Weighted $\mathcal{W}_2$ | $\mathcal{W}_2$ |
| 12 | $6.924 \pm 1.178$ | $6.831 \pm 1.132$ | $8.049 \pm 2.162$ | $13.223 \pm 12.164$ |
| 25 | $6.633 \pm 0.780$ | $6.350 \pm 0.654$ | $6.621 \pm 1.608$ | $9.223 \pm 8.396$ |
| 50 | $7.130 \pm 0.389$ | $7.260 \pm 0.632$ | $5.971 \pm 0.461$ | $5.836 \pm 1.181$ |
| 100 | $6.411 \pm 0.522$ | $6.351 \pm 0.456$ | $5.932 \pm 0.264$ | $6.434 \pm 0.840$ |
| 150 | $6.498 \pm 1.056$ | $6.501 \pm 1.098$ | $6.033 \pm 0.882$ | $7.203 \pm 2.443$ |
| 200 | $6.052 \pm 0.873$ | $6.129 \pm 1.052$ | $5.852 \pm 1.085$ | $6.247 \pm 1.731$ |
| 250 | $6.449 \pm 0.909$ | $6.278 \pm 0.726$ | $6.151 \pm 0.986$ | $11.261 \pm 7.063$ |

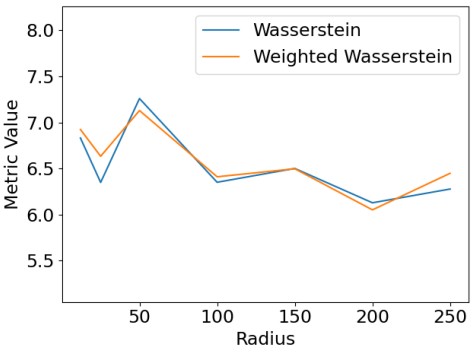
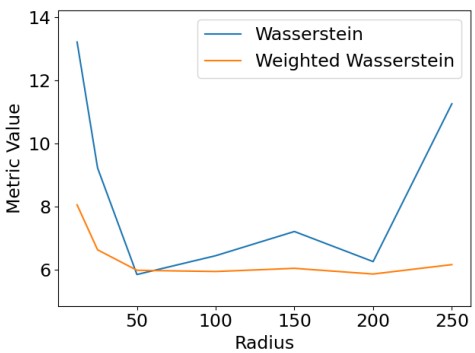

(a) Next Step Sampling

(b) IVP Sampling

Figure 18: Performance variation with radius for extrapolation on the Brain Regeneration dataset.

Table 28: Interpolation for the middle holdout timestep 3 with CTF-H at $\lambda = 0.8$ on the Brain Regeneration dataset.

| Radius | Next Step Sampling | | IVP Sampling | |
|---|---|---|---|---|
| | Weighted $\mathcal{W}_2$ | $\mathcal{W}_2$ | Weighted $\mathcal{W}_2$ | $\mathcal{W}_2$ |
| 12 | $5.320 \pm 1.714$ | $4.709 \pm 2.260$ | $3.722 \pm 1.114$ | $3.656 \pm 1.327$ |
| 25 | $4.943 \pm 1.384$ | $4.467 \pm 1.821$ | $3.350 \pm 1.548$ | $3.112 \pm 1.418$ |
| 50 | $2.440 \pm 0.090$ | $2.302 \pm 0.137$ | $4.181 \pm 0.035$ | $4.238 \pm 0.068$ |
| 100 | $4.028 \pm 0.648$ | $3.417 \pm 0.869$ | $2.956 \pm 0.580$ | $2.678 \pm 0.535$ |
| 150 | $5.408 \pm 0.889$ | $4.669 \pm 1.364$ | $4.535 \pm 0.823$ | $4.209 \pm 0.884$ |
| 200 | $7.110 \pm 2.581$ | $6.490 \pm 3.543$ | $4.043 \pm 1.441$ | $3.754 \pm 1.350$ |
| 250 | $4.502 \pm 0.573$ | $3.689 \pm 1.204$ | $3.532 \pm 1.148$ | $3.457 \pm 1.217$ |

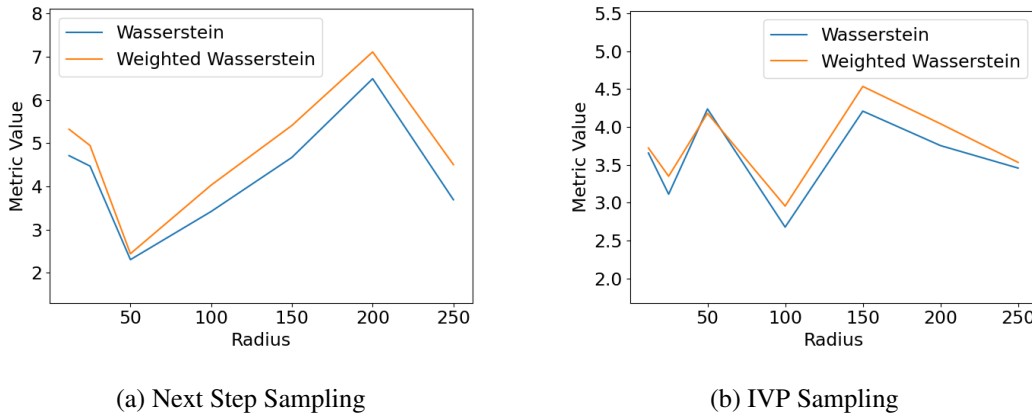

(a) Next Step Sampling                    (b) IVP Sampling

Figure 19: Performance variation with radius for interpolation on the Brain Regeneration dataset

Table 29: Extrapolation for the last holdout timestep on the Brain Regeneration dataset.

| $\epsilon$ | Next Step Sampling | | IVP Sampling | |
|---|---|---|---|---|
| | Weighted $\mathcal{W}_2$ | $\mathcal{W}_2$ | Weighted $\mathcal{W}_2$ | $\mathcal{W}_2$ |
| 0.001 | $6.007 \pm 0.516$ | $5.939 \pm 0.286$ | $6.260 \pm 1.123$ | $7.301 \pm 2.935$ |
| 0.01 | $6.240 \pm 0.870$ | $6.254 \pm 1.111$ | $6.396 \pm 0.236$ | $7.231 \pm 0.968$ |
| 0.1 | $6.579 \pm 0.744$ | $6.861 \pm 0.845$ | $6.758 \pm 1.826$ | $7.283 \pm 2.068$ |
| 1 | $5.648 \pm 0.471$ | $5.721 \pm 0.595$ | $6.010 \pm 0.674$ | $5.905 \pm 0.737$ |
| 10 | $6.841 \pm 0.597$ | $6.940 \pm 0.671$ | $5.532 \pm 1.775$ | $6.646 \pm 1.926$ |
| 100 | $7.166 \pm 0.991$ | $7.094 \pm 1.148$ | $6.455 \pm 3.047$ | $5.650 \pm 1.928$ |
| 1000 | $6.291 \pm 1.041$ | $6.300 \pm 1.052$ | $7.382 \pm 2.553$ | $7.626 \pm 3.204$ |
| 10000 | $6.587 \pm 0.805$ | $6.641 \pm 1.083$ | $5.754 \pm 0.741$ | $7.546 \pm 3.599$ |

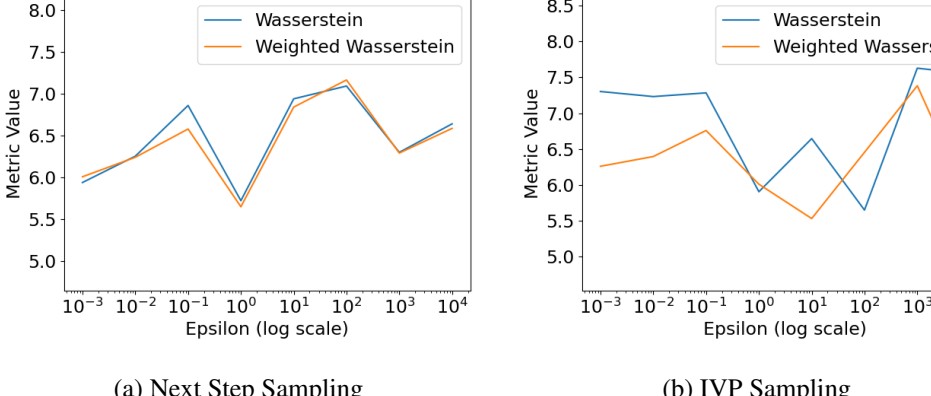

(a) Next Step Sampling

(b) IVP Sampling

Figure 20: Performance variation with $\epsilon$ for extrapolation on the Brain Regeneration Dataset.

Table 30: Interpolation for the middle holdout timestep on the Brain Regeneration dataset.

| $\epsilon$ | Next Step Sampling | | IVP Sampling | |
|---|---|---|---|---|
| | Weighted $\mathcal{W}_2$ | $\mathcal{W}_2$ | Weighted $\mathcal{W}_2$ | $\mathcal{W}_2$ |
| 0.001 | $2.899 \pm 0.582$ | $2.715 \pm 0.653$ | $4.056 \pm 0.542$ | $3.286 \pm 0.289$ |
| 0.01 | $4.520 \pm 2.066$ | $4.589 \pm 2.298$ | $6.915 \pm 3.573$ | $7.125 \pm 5.289$ |
| 0.1 | $2.573 \pm 0.476$ | $2.472 \pm 0.507$ | $3.772 \pm 0.642$ | $3.046 \pm 0.537$ |
| 1 | $2.865 \pm 0.612$ | $2.785 \pm 0.576$ | $4.255 \pm 0.679$ | $3.355 \pm 0.584$ |
| 10 | $2.899 \pm 0.865$ | $2.833 \pm 0.984$ | $4.908 \pm 1.130$ | $4.159 \pm 1.526$ |
| 100 | $2.338 \pm 0.101$ | $1.835 \pm 0.171$ | $5.069 \pm 0.985$ | $4.322 \pm 1.461$ |
| 1000 | $3.104 \pm 0.663$ | $2.321 \pm 0.521$ | $5.109 \pm 0.948$ | $3.974 \pm 1.227$ |
| 10000 | $2.838 \pm 0.281$ | $2.176 \pm 0.315$ | $4.557 \pm 0.710$ | $3.373 \pm 0.833$ |

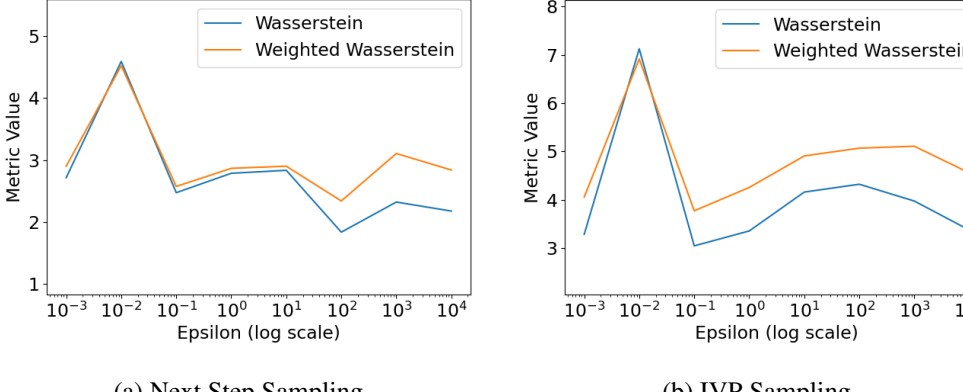

(a) Next Step Sampling        (b) IVP Sampling

Figure 21: Performance variation with $\epsilon$ for interpolation on the Brain Regeneration dataset.

