# OpenReview forum: "ContextFlow: Context-Aware Flow Matching for Trajectory Inference from Spatial Omics Data"
_ICLR.cc/2026/Conference — Submitted to ICLR 2026_

### Official Review · Reviewer_ZUfA · 2025-10-23

**Soundness:** 2
**Presentation:** 3
**Contribution:** 2
**Rating:** 6
**Confidence:** 4

**Summary:**

This paper introduces ContextFlow, a flow matching framework that incorporates spatial context from spatially-resolved omics data to infer biologically plausible cellular trajectories over time. The key innovation is integrating two spatial priors—(1) local tissue organization via spatial smoothness and (2) ligand-receptor communication patterns—into a transition plausibility matrix that regularizes optimal transport couplings. The authors propose two integration schemes: Prior-Aware Cost Matrix (PACM) and Prior-Aware Entropy Regularization (PAER). They demonstrate that ContextFlow outperforms baseline methods (CFM, MOTFM) across three datasets (axolotl brain regeneration, mouse organogenesis, liver regeneration) under both interpolation and extrapolation settings, using metrics including Wasserstein distance, MMD, energy distance, and a weighted Wasserstein metric based on cell type compositions.

**Strengths:**

1. **Well-motivated problem**: Incorporating spatial context into trajectory inference addresses a real limitation of existing methods and is timely given advances in spatial transcriptomics.

2. **Principled approach**: The integration of biological priors through OT regularization is theoretically grounded, and Theorem 1 ensures computational tractability via Sinkhorn.

3. **Comprehensive experiments**: Three diverse datasets with both interpolation and extrapolation settings. The biological analysis (e.g., excitatory-inhibitory transitions in Figure 3) provides qualitative validation.

4. **Two integration schemes**: Offering both PACM and PAER gives practitioners flexibility, and showing PAER often performs well without extra hyperparameter tuning is valuable.

5. **Reproducibility**: Algorithm 1 and extensive ablations in the appendix support reproducibility.

**Weaknesses:**

1. **Limited novelty**: The core contribution—adding prior knowledge to OT objectives—is incremental. PACM is a straightforward weighted cost, and PAER follows naturally from KL divergence regularization literature. The connection to existing spatial transcriptomics OT methods (TOAST, DeST-OT) needs clearer differentiation.

2. **Experimental limitations**:
   - **Modest improvements**: Many results show <5-10% improvement over MOTFM, often within error bars
   - **Extreme instability**: Table 12 shows catastrophic failures in hard extrapolation (errors 10^5-10^6), severely limiting practical utility
   - **Limited baselines**: No comparison to established trajectory inference methods (e.g., Waddington-OT, CellRank, RNA velocity-based methods)
   - **Missing ablations**: No systematic study of spatial smoothness vs. LR communication independently

3. **Evaluation concerns**:
   - **Weighted W2 confounding**: Depends on XGBoost classifier quality; unclear if improvements reflect better trajectories or classifier biases
   - **No statistical testing**: Despite reporting standard deviations, no significance tests are provided
   - **Biological validation**: The excitatory-inhibitory analysis (Figure 3) is interesting but limited to one dataset and one specific biological phenomenon

4. **Hyperparameter sensitivity**: Despite claims that PAER avoids tuning, λ still needs selection. Tables 5-13 show considerable variance, and there's no principled guidance for setting these parameters.

5. **Scalability and generalizability**:
   - Computational complexity not discussed
   - Limited to three datasets from specific biological contexts
   - Unclear how to set the neighborhood radius r or select relevant LR pairs for new datasets

6. **Technical issues**:
   - The normalization analysis (Propositions 1-2) seems tangential
   - Equation 10 assumes a linear combination of SS and LR—no justification for this specific form
   - The prior matrix construction (Equation 10) is somewhat ad-hoc

**Questions:**

1. **Ablation studies**: Can you provide results using only spatial smoothness (λ=1) or only LR communication (λ=0) systematically across all datasets and metrics? This would clarify the relative importance of each prior.

2. **Baseline comparisons**: How does ContextFlow compare to other trajectory inference methods like Waddington-OT, CellRank, or RNA velocity approaches? Even if they don't use spatial information, they provide important benchmarks.

3. **Weighted W2 metric**: How is the XGBoost classifier trained? What is its accuracy? Could you provide results without this metric to separate trajectory quality from classifier performance?

4. **Stability**: Table 12 shows extreme instability. Can you characterize when and why these failures occur? Are there early warning signs or ways to detect when the method will fail?

5. **Hyperparameter selection**: Can you provide principled guidance for setting λ, α, and ε for new datasets? Cross-validation results would be helpful.

6. **Computational cost**: What is the runtime comparison vs. MOTFM? How does it scale with dataset size?

7. **Generalization**: Can you provide evidence that the learned velocity fields capture meaningful biological dynamics beyond the specific datasets tested?

8. **LR database sensitivity**: How sensitive are results to the choice of LR database or communication inference method?

---

> ### Author Response · Authors · 2025-11-22
> **Official Response to Reviewer ZUfA**
>
> We thank the reviewer for their feedback.
>
> > W1: Limited novelty
>
> We've included the advantages and novelty of ContextFlow compared to existing methods in the related work section, presented in a table. We've also benchmarked the prior-aware OT formulation from ContextFlow with SOTA spatiotemporal alignment methods such as DeST-OT and TOAST in Appendix H. In summary, the existing OT spatiotemporal alignment methods are not generative models, relying on a costly Gromov-Wasserstein term. In contrast, existing flow-matching techniques, although generative, do not fully exploit the structural information present in spatiotemporal omics datasets. ContextFlow bridges all these gaps, which is a generative model that learns the underlying dynamics across the entire time horizon and is not limited to two consecutive tissue slices.
>
> > W2: Experimental limitations. (Modest improvements) Many results show <5-10\% improvement over MOTFM, often within error bars; (Extreme instability) Table 12 shows catastrophic failures in hard extrapolation, severely limiting practical utility; (Limited baselines) No comparison to established trajectory inference methods (e.g., Waddington-OT, CellRank, RNA velocity-based methods); (Missing ablations) No systematic study of spatial smoothness vs. LR communication independently
>
> The failure/instability occurs when we generate samples using IVP sampling for the hard extrapolation setting in the mouse organogenesis dataset. And it's valid for all the techniques we benchmark against, with ContextFlow still being the best-performing among them. This instability is highly dependent on the initial points chosen for to integrate the velocity field over. Although this doesn't limit the applicability of ContextFlow or flow-based methods, since, practically, the model can still be used to **generate samples using Next-Step sampling**. We've mentioned in the paper that you can integrate the velocity field using samples from the previous time instead of the very first timestep as used in IVP sampling.
>
> In Appendix H, we've added a benchmark of the prior-aware OT we use for better conditional paths to train ContextFlow with respect to DeST-OT and TOAST, which are improved methods over Waddington-OT. And Minibatch-OT Flow-Matching (MOTFM) already covers a neural network equivalent for trajectory inference, like RNA-velocity, and we've benchmarked against it.
>
> We've also more explicitly added ablations in Appendix K.1, with respect to $\lambda$ for the ContextFlow entropic (CTF-H) variation, where the relative effect of Spatial Smoothness (SS) and LR communication patterns can be observed.  Overall, we observe that SS outperforms LR communication patterns in most cases, because a distinct communication pattern occurs only at some timesteps, where it can be exploited. In contrast, SS, which acts as a proxy for spatial distance, is always informative, as it captures spatial context.
>
> > W3: Evaluation concerns. (Weighted W2 confounding) Depends on XGBoost classifier quality, unclear if improvements reflect better trajectories or classifier biases; (No statistical testing) Despite reporting standard deviations, no significance tests are provided.
>
> The XGBoost model we trained for cell-type classification achieved approximately $100\%$ accuracy on the training set and approximately $92\%$ on the holdout test set. Additionally, XGBoost has been recently shown to be most suitable for cell-type classification tasks, as demonstrated in this repository: https://huggingface.co/spaces/Arozhada/spcelleval. Having said that, we also evaluate using Wasserstein Distance, along with Energy Distance and MMD score, in addition to the weighted-Wasserstein metric.
>
> > W4: Biological validation. The excitatory-inhibitory analysis (Figure 3) is interesting but limited to one dataset and one specific biological phenomenon
>
> In this work, we aimed to propose a new generative flow-matching technique for spatial omics datasets meant to overcome the lack of generative capabilities of SOTA spatiotemporal OT methods and the lack of structure/prior exploitation in existing flow-matching literature. The excitatory-inhibitory analysis was an illustrative example of prior limitations. ContextFlow, owing to its scalability, can be modified accordingly for different datasets and can potentially be used to generate such novel insights depending on the biological context, something we aim to explore more in future works.
>
> > W5: Hyperparameter sensitivity. Despite claims that PAER avoids tuning, $\lambda$ still needs selection. Tables 5-13 show considerable variance, and there's no principled guidance for setting these parameters.
>
> We've added an ablation study with respect to $\lambda$ in Appendix K.1, along with our recommendations for setting $\lambda$ appropriately. And since ContextFlow is scalable, multiple runs can be performed to determine the optimal setting.

---

> > ### Author Response · Authors · 2025-11-22
> > **Official Response to Reviewer ZUfA**
> >
> > > W6: Scalability and generalizability. Computational complexity not discussed. Limited to three datasets from specific biological contexts. Unclear how to set the neighborhood radius r or select relevant LR pairs for new datasets
> >
> > We've added ablations on the radius $r$ and several other hyperparameters in Appendix  K, along with a time-complexity analysis in Appendix E. We've also discussed how we set these hyperparameters, along with recommendations on setting these.
> >
> > > W7: Technical issues: The normalization analysis (Propositions 1-2) seems tangential. Equation 10 assumes a linear combination of SS and LR—no justification for this specific form. The prior matrix construction (Equation 10) is somewhat ad-hoc.
> >
> > Both SS and LR are intended to provide a biological prior indicating which couplings are more biologically feasible than others. SS acts as a proxy for spatial distance, while LR acts as a proxy for cell-cell communication patterns. To elaborate biologically, two cells with similar neighborhood expression (SS distance) are more likely to have a feasible transition between them than otherwise, the same being true for cell-cell LR communication patterns.
> >
> > The linear combination with $\lambda$ is meant to capture the relative importance of both depending on the dataset and setting considered. However, we agree that better formulations can potentially be explored in the future.
> >
> > > Q1: Can you provide results using only spatial smoothness ($\lambda$=1) or only LR communication ($\lambda$=0) systematically across all datasets and metrics? This would clarify the relative importance of each prior.
> >
> > We've included an ablation study on $\lambda$ in Appendix K, which considers extreme values of 0 and 1 to examine the effects of only LR and only SS.
> >
> > We observe that the best performance is often achieved at $\lambda = 0$ or $1$, and we think that this is dataset- and timestep-dependent. For certain consecutive timesteps, the LR communication patterns are similar, meaning that there's no discernible signal that can biologically indicate that two cells should be coupled based on their LR patterns. However, for certain datasets and timesteps, the LR communication pattern is quite distinct, as shown in Figure 8 of Appendix I of the revised manuscript, where LR is extremely informative. While the spatial smoothness prior is more informative in a homogenous manner, as it serves as a proxy for spatial distance, which is always informative and contains the structural signal present in spatial omics datasets. The relative importance thus depends on how informative the LR communication patterns are for a given dataset, for a certain time horizon. And this is the reason why optimal performance usually occurs at $\lambda=0$ when LR is very informative, $\lambda=1$ when LR isn't informative, and $\lambda$ near 1 (e.g., 0.8) when LR is slightly informative but not more than the local neighborhood expression (SS).
> >
> > > Q2: How does ContextFlow compare to other trajectory inference methods like Waddington-OT, CellRank, or RNA velocity approaches? Even if they don't use spatial information, they provide important benchmarks
> >
> > It is important notice that Waddington-OT and other OT methods from the related section are primarily designed for _aligning_ spatiotemporal omics data rather than generative modelling. In contrast, ContextFlow is a generative model where a single neural network models the dynamics across the time horizon. Thus, direct comparison between ContextFlow and Waddington is not appropriate; however, we added a comprehensive comparison between the OT formulation used to generate conditional paths in ContextFlow and SOTA spatiotemporal alignment methods, such as Dest-OT and TOAST, in Appendix H. We observe that our proposed prior-aware OT formulation is competitive with DeST-OT/TOAST across metrics while being significantly faster than them.
> >
> > Both DeST-OT/TOAST are improvements over Waddington-OT. And Minibatch-OT Flow-Matching (MOTFM) already covers a neural network equivalent for trajectory inference, like RNA-velocity, and we've also benchmarked against it.
> >
> > > Q3: How is the XGBoost classifier trained? What is its accuracy? Could you provide results without this metric to separate trajectory quality from classifier performance?
> >
> > The XGBoost multi-class classifier is trained in its default setting, using cell features as input and cell-type labels as categorical outputs. We maintain an 80/20 split for the train and test sets across all datasets, and we observe that, across datasets, the train accuracy approaches $100\%$, while the test accuracy approaches $92\%$. XGBoost has been independently validated as a cell-type classifier compared to other approaches in the repository: https://huggingface.co/spaces/Arozhada/spcelleval. We've also included metrics such as Wasserstein, MMD, and Energy distance in the ablation tables.

---

> > > ### Author Response · Authors · 2025-11-22
> > > **Official Response to Reviewer ZUfA**
> > >
> > > > Q4: Table 12 shows extreme instability. Can you characterize when and why these failures occur? Are there early warning signs or ways to detect when the method will fail?
> > >
> > > The failure/instability only occurs when we generate samples using IVP sampling for the hard extrapolation setting in the mouse organogenesis dataset. And it's valid for all the techniques we benchmark against, with ContextFlow still being the best-performing among them. This instability is highly dependent on the initial points chosen for to integrate the velocity field over.
> > >
> > > This doesn't limit the applicability of ContextFlow or flow-based methods since, practically, the model can still be used to **generate samples using Next-Step sampling** (Table 3, extrapolation) we've mentioned in the paper, wherein, you integrate the velocity field using samples from the previous time instead of the very first timestep as in IVP (initial value problem) sampling.
> > >
> > > > Q5: Can you provide principled guidance for setting $\lambda$, $\alpha$, and $\epsilon$ for new datasets? Cross-validation results would be helpful.
> > >
> > > Yes, we've now included ablation studies regarding hyperparameters, along with a detailed discussion on how to set these values, in Appendix K.
> > >
> > > > Q6: What is the runtime comparison vs. MOTFM? How does it scale with dataset size?
> > >
> > > We've added a runtime analysis of ContextFlow compared to MOTFM in Appendix E. Along with runtime analysis of the prior aware OT used in ContextFlow with other SOTA spatiotemporal OT methods, such as DeST-OT/TOAST, in Appendix H.2. Importantly, we would like to clarify that the calculation of Ligand-Receptor interaction (LR) features and local spatial smoothness (SS) features is included in the data pre-processing step and is a one-time cost. While running ContextFlow, we do not recompute them, making our approach scalable.
> > >
> > > To summarize, excluding the pre-processing steps, the total time complexity for training ContextFlow is $O(E \times P \times (B^{2}(d+l)))$, where $E$ is the training epochs, $P$ is the time taken for forward/backward pass using gradient descent, $B$ is the mini-batch size for computing minibatch OT couplings, $d$ is the transcriptomic feature dimension, and $l$ is the total number LR pairs.
> > >
> > > > Q7: Can you provide evidence that the learned velocity fields capture meaningful biological dynamics beyond the specific datasets tested?
> > >
> > > The current scope of the work aims to provide a scalable generative model capable of generating new samples within the same context as the training data, which can be further utilized in downstream applications as needed. While the biological dynamics are inherently present in the trained vector field, extracting mechanistic biological insights from them and validating their generalizability in terms of something more fundamental requires further efforts and experiments on other datasets, which we aim to explore in future work.
> > >
> > > > Q8: How sensitive are results to the choice of LR database or communication inference method?
> > >
> > > We utilize the recommended Consensus resource from LIANA+, which consolidates several LR databases to determine LR communication patterns. Since we don't rely on any particular database, the pattern determination is already robust by design.

---

### Official Review · Reviewer_w9VN · 2025-10-27

**Soundness:** 3
**Presentation:** 3
**Contribution:** 2
**Rating:** 4
**Confidence:** 4

**Summary:**

This paper introduces ContextFlow, a flow-matching framework designed to infer cellular trajectories from snapshot spatial omics data. ContextFlow integrates two types of spatial priors (Spatial Smoothness, Cell-Cell Communication Patterns) to the computation of OT couplings, and then performs conditional flow matching, which can help avoid biologically implausible trajectories. Specifically, these priors are encoded into a Transitional Plausibility Matrix (TPM) and incorporated into the OT objective via two schemes: Prior-Aware Cost Matrix (PACM), and Prior-Aware Entropy Regularization (PAER). Through experiments on three datasets, ContextFlow outperforms CFM and OT-CFM in interpolation and extrapolation tasks, and reduces biologically invalid cell lineage transitions.

**Strengths:**

- Inferring trajectories from spatiotemporal data is a critical and challenging task in computational biology.
- This paper is well-motivated, as it aims to tackle the limitations of other OT-based methods, where these methods may induce biologically implausible transitions.
- Incorporating biological priors into the OT coupling is a natural extension. By modifying the Gibbs kernel, the entropy-based regularization guides the transport plan in a theoretically sound manner while simplifying the hyperparameter tuning process.

**Weaknesses:**

Despite its promising direction, the paper suffers from several critical weaknesses that question the validity and significance of its central claims.
- **Omission of Spatial Coordinates:** The first concern is the disregard for Euclidean spatial distance in the OT cost function for both the proposed method and the baseline. For spatial data, the most direct and powerful prior is that cells cannot transport across the tissue in a short time interval. I believe it is important to penalize the physical distance between matched cells. The paper's "spatial smoothness" prior is an indirect and potentially weak proxy for this physical constraint. It is possible for two distant regions to have similar neighborhood expression profiles, which could lead to **biologically impossible long-range couplings.** The authors' claim in Appendix G.1 that ContextFlow can "restrict transitions to within the same hemisphere" is questionable when visually inspecting Figure 3(b). This figure still appears to show several long-range couplings that may not be biologically reasonable. One way to examine this is to evaluate the reconstructed spatial coordinates at hold-out time points.
- **Inadequate Baselines:** As said, it seems that the spatial information is not incorporated into baselines, which may be unfair. I also recommend that the authors compare their proposed method against some newer flow-matching baselines, such as Metric Flow Matching [1], some neural-ODE baselines, such as DeepRUOT[2], and **methods specifically designed for mapping spatially resolved data**, such as moscot [3].
- **Theoretical Gap Between Static Coupling and Dynamic Trajectory Generation:** The work relies on the Conditional Flow Matching framework, which assumes a linear interpolation path between two coupled endpoints. While ContextFlow constrains the choice of these endpoints, it provides **no guarantee about the validity of the path between them.** A straight line in high-dimensional gene expression space is likely to break those proposed constraints and induce biologically implausible transitions. This paper fails to provide any theoretical or compelling empirical justification for this point. I think the authors should examine the biologically implausible transitions along the whole trajectory instead of only endpoints.
- **Hyperparameter Selection:** It seems that the paper does not specify the value of entropy regularization parameter $\epsilon$ used in the experiments, nor does it describe the strategy for its selection or provide a sensitivity analysis. I assume this is an important parameter to constrain the OT coupling, so this omission is significant for the reproducibility and a full understanding of the method's robustness. I recommend providing a guidance on how to select all these hyperparameters in this paper for different datasets, and discussing the efforts for hyperparameter tuning.
- **Minor Points:**
    1. The notation CTF-C and PACM (or CTF-H vs. PAER) is a bit confusing. If they refer to the same concept, I recommend using one consistent term throughout the paper.
    2. Equation 4, should that be $(x-\mu_{t}(z))$?
    3. Equation 9 should be more clearly stated. It is unclear how this is computed in practice.
    4. The construction of the TPM involves neighborhood calculations and ligand-receptor interaction inference, which can be computationally intensive as the number of cells grows. The authors should provide runtime of the whole pipeline.
    5. I think the concept of constrained OT is a well-established field. The authors should provide a more thorough discussion in the related work section to better position their specific contribution.

[1] Kapusniak, Kacper, et al. "Metric flow matching for smooth interpolations on the data manifold." NeurIPS
[2] Zhang, Zhenyi et al., "Learning stochastic dynamics from snapshots through regularized unbalanced optimal transport". ICLR
[3] Klein, Dominik, et al. "Mapping cells through time and space with moscot." Nature

**Questions:**

Following the above weaknesses:
- How well can ContextFlow reconstruct spatial coordinates in hold-out time points?
- Could the authors provide comparison results against more appropriate baselines?
- Can the authors provide justification (theoretical or empirical) for why the linear path assumption in CFM is sufficient to produce biologically reasonable trajectories, given that the proposed constraints are only applied at the endpoints?
- How is the parameter $\epsilon$ set for different datasets?
- How is the runtime of ContextFlow?

If the authors can address the weaknesses and questions outlined, I will consider revising my score.

The reviewer wrote this review. LLM was utilized only to correct grammar and enhance the clarity of this review.

---

> ### Author Response · Authors · 2025-11-22
> **Official Response to Reviewer w9VN**
>
> We thank the reviewer for their feedback.
>
> > W1: Omission of Spatial Coordinates
>
> We thank the reviewer for bringing this to our attention. The direct spatial distance is excluded because, based on current technologies, no two slices are guaranteed to have the same global frame of reference; as such, calculating direct spatial distance doesn't necessarily imply actual physical distance between cells from different tissue slices. We thus employ spatial smoothness as a proxy for relative location, with the hypothesis that cells physically closer to each other have similar neighborhood transcriptomic expression levels.
>
> > W2: Inadequate Baselines: As said, it seems that the spatial information is not incorporated into baselines, which may be unfair. I also recommend that the authors compare their proposed method against some newer flow-matching baselines, such as Metric Flow Matching, some neural-ODE baselines, such as DeepRUOT, and methods specifically designed for mapping spatially resolved data, such as moscot
>
> We agree that comparing our approach with Metric-FM and DeepRUOT would indeed be interesting; however, due to the limited rebuttal window and the substantial engineering effort required, we were unable to include comparisons with Metric-FM and DeepRUOT. We see these as promising baselines and plan to explore them in future work.
>
> MOSCOT uses a slight variation of the PASTE OT formulation for spatiotemporal alignment and isn't a generative model. We've provided benchmarks of our proposed prior-regularized OT used in ContextFlow in Appendix H, alongside SOTA OT methods such as DeST-OT and TOAST, where we observe that we're competitive with them while being significantly faster.
>
> > W3: Theoretical Gap Between Static Coupling and Dynamic Trajectory Generation: The work relies on the Conditional Flow Matching framework, which assumes a linear interpolation path between two coupled endpoints. While ContextFlow constrains the choice of these endpoints, it provides no guarantee about the validity of the path between them.
>
> We thank the reviewer for their deeply insightful comment. While it's true that for each mini-batch, post OT coupling, the conditional path is linear interpolation between the coupled points, it's important to note that such linear interpolations are calculated across several non-overlapping mini-batches, resulting in several linear paths that cross each other. And since a single _non-linear_ neural network is being regressed over these cross-over paths, the velocity field learned by the neural network is not a linear function. Additionally, a single neural network is regressed over multiple time horizons, resulting in the network learning whatever non-linear function best fits the complicated overall paths.
>
> The applicability of any flow-based generative algorithm is naturally limited to datasets where the temporal transcriptomic variations aren't far apart.  When the variation across timesteps becomes too large, the data may not contain enough signal to reliably recover the intermediate dynamics, unless it is externally provided by some inductive prior or implicitly present in the network via pretraining on large datasets. If the data from holdout timesteps deviates too much from the presented samples, then even with additional contextual information, domain prior, stochasticity, or curvature, the model cannot reconstruct transitions that are not supported by the underlying data distribution.
>
> ContextFlow is designed to demonstrate that for spatiotemporal omics data, the inherent structure and domain knowledge can be leveraged for enhanced generative capabilities. Existing flow-matching methods do not explicitly leverage this structure, while the spatiotemporal OT methods that capture these aren't generative. We've also included a benchmark with SOTA spatiotemporal OT alignment methods in Appendix H.
>
> > W4: It seems that the paper does not specify the value of entropy regularization parameter $\epsilon$. I recommend providing a guidance on how to select all these hyperparameters in this paper for different datasets, and discussing the efforts for hyperparameter tuning.
>
> We've added an ablation on $\epsilon$ in Appendix K.3, and discuss on how we set $\epsilon$. Briefly, we set the $\epsilon$ to be of the same order as the transcriptomic cost $C$ in order to ensure that the Gibbs kernel $\exp(-\mathbf{C}/{\epsilon})$ is numerically stable. In our experiments, we notice that performance is relatively stable with increasing $\epsilon$ owing to the result from Theorem 1 where transition plausibility matrix constrains the coupling appropriately when $\epsilon \rightarrow \infty$.

---

> > ### Author Response · Authors · 2025-11-22
> > **Official Response to Reviewer w9VN**
> >
> > > W5: Minor points about notation/runtime/related works
> >
> >
> > We thank the reviewer for pointing out the notational inconsistencies. We've now corrected it. We've also modified our related work section, adding a table to position ContextFlow more clearly in relation to the existing literature.
> >
> > Moreover, we would like to clarify that the calculation of spatial smoothness (neighborhood expression) and LR communication patterns is included in the pre-processing steps and does not occur during training. We've modified the algorithm to reflect the same, and have included a time complexity analysis in Appendix E.
> >
> >
> > > Q1: How well can ContextFlow reconstruct spatial coordinates in hold-out time points?
> >
> > ContextFlow uses the spatial context for designing better conditional paths, and its velocity field is meant to be the transcriptomic embedding velocity, and not of spatial coordinates. Thus, reconstructing spatial coordinates is outside the scope of this work. Having said that, spatial coordinate reconstruction is also made much harder by the fact that the frames of reference aren't aligned between slices.
> >
> > > Q2: Could the authors provide comparison results against more appropriate baselines?
> >
> > We agree that comparing our approach with Metric-FM and DeepRUOT would indeed be interesting; however, due to the limited rebuttal window and the substantial engineering effort required, we were unable to include comparisons with Metric-FM and DeepRUOT. We see these as promising baselines and plan to explore them in future work.
> >
> > MOSCOT uses a slight variation of the PASTE OT formulation for spatiotemporal alignment and isn't a generative model. We've provided benchmarks of our proposed prior-regularized OT used in ContextFlow in Appendix H, alongside SOTA OT methods such as DeST-OT and TOAST, where we observe that we're competitive with them while being significantly faster.
> >
> > > Q3: Can the authors provide justification (theoretical or empirical) for why the linear path assumption in CFM is sufficient to produce biologically reasonable trajectories, given that the proposed constraints are only applied at the endpoints?
> >
> > In addition to our response to a similar comment earlier, we note that the constraints are applied between each consecutive timestep present in the training data, not just at the endpoints. The velocity field at every section of the time horizon is trained with biologically informed couplings, meaning that the overall dynamics learned are guided by biologically meaningful priors.
> >
> > > Q4: How is the parameter set for different datasets?
> >
> > As mentioned earlier, we've added an ablation on $\epsilon$ in Appendix K.3, and discuss how we set $\epsilon$. Briefly, we set the $\epsilon$ to be of the same order as the transcriptomic cost $\mathbf{C}$ in order to ensure that the Gibbs kernel $\exp (-\mathbf{C}/{\epsilon})$ is numerically stable. In our ablations we notice that performance is relatively stable with increasing $\epsilon$ owing to the result from Theorem 1 where transition plausibility matrix constrains the coupling appropriately when $\epsilon \rightarrow \infty$.
> >
> > > Q5: How is the runtime of ContextFlow?
> >
> > We've included a runtime complexity analysis of Context in Appendix E, which is shown to be similar to MOTFM.

---

### Official Review · Reviewer_s8Za · 2025-10-30

**Soundness:** 3
**Presentation:** 3
**Contribution:** 2
**Rating:** 4
**Confidence:** 4

**Summary:**

This paper addresses the challenge of inferring cellular trajectories from longitudinal, spatially-resolved omics data. In this setting, the vanilla Optimal Transport (OT) relies solely on transcriptomic similarity, potentially leading to biologically implausible trajectories. The authors propose ContextFlow to integrate spatial and biological priors for the OT batch sampling step of CFM. This is achieved by constructing a Transitional Plausibility Matrix (TPM) encoding Spatial Smoothness (SS) and Cell-Cell Communication (CCC) patterns. ContextFlow introduces two integration mechanisms: cost-based (CTF-C) and entropy-based (CTF-H). Experiments demonstrate improved statistical fidelity and biological plausibility over baselines.

**Strengths:**

- **S1: Motivation.** Incorporating spatial context and biological priors (LR interactions) is highly relevant for trajectory inference in spatial biology.

- **S2: Principled Integration (CTF-H).** The PAER (CTF-H) approach elegantly incorporates priors via entropy regularization, avoiding normalization issues associated with modifying the cost matrix (CTF-C).

- **S3: Comprehensive Evaluation.** Experiments on three diverse datasets demonstrate consistent quantitative improvements across various metrics and sampling scenarios (Interpolation, Extrapolation, IVP).

- **S4: Biological Plausibility.** Qualitative analysis (Appendix G.1) convincingly shows a substantial reduction in biologically implausible couplings (e.g., lineage switches) compared to MOTFM.

**Weaknesses:**

- **W1: Sensitivity and Interpretation of Hyperparameters ($\lambda, r$).** The method relies heavily on the neighborhood radius $r$ and the trade-off $\lambda$. The paper lacks ablation on $r$. Crucially, empirical results (e.g., Table 2) show the best performance often occurs at extremal values ($\lambda=0$ or $\lambda=1$), suggesting either SS or LR dominates. The paper does not provide biological justification for why one prior dominates in specific datasets or guidance on how to select $\lambda$ a priori.

- **W2: Ambiguity in PAER Formulation.** The normalization of the prior matrix $\hat{\mathbf{M}}$ in PAER (Line 301) appears row-wise, creating conditional probabilities rather than a joint distribution. The interpretation of the regularization term $H(\Pi \mid \hat{\mathbf{M}})$ requires clarification if $\hat{\mathbf{M}}$ is not a valid joint measure.

- **W3: Exclusion of Direct Spatial Distance.** The method uses neighborhood features but does not explicitly use physical distance ($||\mathbf{s}(c_i) - \mathbf{s}(c_j)||$) to constrain cell movement.

**Questions:**

1. **(W1) Lambda Interpretation:** In the results (e.g., Tables 1 and 2), the best performance is often achieved when $\lambda=0$ or $\lambda=1$. What does this imply about the synergy between SS and LR priors? Can you provide biological justification for why one prior dominates in specific datasets, and how should $\lambda$ be selected a priori?

2. **(W1) Hyperparameter Sensitivity:** Could you provide an ablation study on the selection of the neighborhood radius $r$ and the method's sensitivity to this parameter?

3. **(W2) PAER Formulation:** Regarding Line 301, the normalization of $\hat{\mathbf{M}}$ sums only over the target index $l$, resulting in a row-stochastic matrix. Is this intended? If so, please clarify the interpretation of $H(\Pi \mid \hat{\mathbf{M}})$ when $\hat{\mathbf{M}}$ is not a joint distribution.

4. **Strength of Regularization:** Can you elaborate on the mechanism by which the TPM-guided regularization (structured diffusion) so strongly affects the resulting trajectories, effectively pruning biologically implausible paths compared to standard EOT diffusion?

5. **Proofreading (Theorem 1):** In Appendix B, Line 914 seems to have a sign error for $g_l$ compared to the derivation from the Lagrangian (Line 912). Should it be $\epsilon \log (\Pi_{kl}^*/M_{kl}) = f_k + g_l - C_{kl}$?RetryTo run code, enable code execution and file creation in Settings > Capabilities.

---

> ### Author Response · Authors · 2025-11-22
> **Official Response to Reviewer s8Za**
>
> We thank the reviewer for their feedback.
>
> > W1: Sensitivity and Interpretation of Hyperparameters $\lambda$ and $r$ and how to select them
>
> We've added an ablation study on $\lambda$ in Appendix K.1. While both SS and LR features act as proxies for the biological intuition of how likely two cells are to be coupled based on their neighborhood expression and communication patterns, $\lambda$ is used to tune their relative influence. Determining optimal $\lambda$ is dataset dependent; however, as seen in our ablations in Appendix K, the optimal usually occurs on extreme values $\lambda = 0,1$ or near $1$ (e.g., $\lambda = 0.8$).
>
> The hyperparameter $r$ represents the radius used by the KNN algorithm to determine the local neighborhood for each sample. We've now included an ablation study regarding $r$ in Appendix K.2. Overall, we observe that the performance usually degrades if $r$ is too high or too low.
>
>
> > W2: Ambiguity in PAER Formulation
>
> We thank the reviewer for bringing this to our attention. Indeed, our TPM (transition probability matrix) $\widehat{\mathbf{M}}$ is row-normalized and thus represents the conditional transitional likelihood. Each element $[\widehat{\mathbf{M}}{(i,i+1)}]_{kl}$ can be physically interpreted as how likely is the cell $k$ at timestep $i$ to be coupled with cell $l$ at timestep $i+1$, which in turn depends how close cell $k$ is compared to all the cells at timestep $i+1$ in terms of local neighborhood distance (SS) and communication patterns (LR).
>
> > W3: Exclusion of Direct Spatial Distance
>
> The direct spatial distance is excluded because, based on current technologies, no two slices are guaranteed to have the same global frame of reference; as such, calculating direct spatial distance doesn't necessarily imply actual physical distance between cells from different tissue slices. We thus employ spatial smoothness as a proxy for relative location, with the hypothesis that cells physically closer to each other have similar neighborhood transcriptomic expression levels.

---

> ### Author Response · Authors · 2025-11-22
> **Official Response to Reviewer s8Za**
>
> > Q1: In the results (e.g., Tables 1 and 2), the best performance is often achieved when $\lambda=0$ and $\lambda=1$. What does this imply about the synergy between SS and LR priors? Can you provide biological justification for why one prior dominates in specific datasets, and how should $\lambda$ be selected a priori?
>
> The reviewer is indeed correct to note that best performance is often achieved at $\lambda = 0, 1$, and we've also included an ablation study on $\lambda$ in Appendix K.1. We think that this is dataset and timestep dependent since for certain consecutive timesteps the LR communication patterns are similar, meaning that there's no discriminative signal that can biologically say that two cells should be coupled based on their LR patterns. However, for certain datasets and timesteps, the LR communication pattern is quite distinct, as shown in Figure 8 of Appendix I of the revised manuscript, where LR is extremely informative. The spatial smoothness prior is more informative in a more homogeneous manner. The synergy thus depends on how informative the LR communication patterns are for a given dataset, for a certain time horizon. And this is the reason why optimal performance usually occurs at $\lambda=0$ when LR is very informative, $\lambda=1$ when LR isn't informative, and $\lambda$ near 1 (e.g., 0.8), when LR is slightly informative but not more than the local neighborhood expression (SS).
>
> > Q2: Could you provide an ablation study on the selection of the neighborhood radius $r$ and the method's sensitivity to this parameter?
>
> We've now included an ablation study on $r$ in Appendix K.2. Overall, we observe that the performance usually degrades if $r$ is too high or too low.
>
> > Q3: PAER Formulation -  Line 301 and normalization of ${M}$
>
> Indeed, our TPM (transition plausibility matrix) $\widehat{\mathbf{M}}$ is row-normalized and thus represents the conditional transitional likelihood. Each element $[\widehat{\mathbf{M}}{(i, i+1)}]_{kl}$ can be physically interpreted as how likely is the cell $k$ at timestep $i$ to be coupled with cell $l$ at timestep $i+1$, which in turn depends how close cell $k$ is compared to all the cells at timestep $i+1$ in terms of local neighborhood features distance (SS) and communication patterns (LR).
>
> > Q4: Can you elaborate on the mechanism by which the TPM-guided regularization (structured diffusion) so strongly affects the resulting trajectories, effectively pruning biologically implausible paths compared to standard EOT diffusion?
>
> TPM essentially encodes the biological domain knowledge that informs our understanding of whether one cell can transition into another. Now, from biology, we know that (a) cells with similar neighborhood transcriptomic expression are more likely to transition into one another than otherwise, and (b) cell-cell communication patterns are more likely to be similar between feasible transitions than infeasible ones. This domain knowledge is encoded in TPM via SS and LR matrices, where a large value indicates dissimilar neighborhood expressions and communication patterns. We then turn this into a transition prior matrix $[\widehat{\mathbf{M}}{(i, i+1)}]_{kl}$, giving us the likelihood of cell $k$'s state transitioning into cell $l$'s state. Thus, in essence, TPM encodes a kind of biological prior knowledge.
>
> We integrate this knowledge into the entropic-OT formulation by either directly utilizing it in the cost-function or adding it in a relative entropic term. The former essentially enforces high transport cost of transitions with dissimilar neighborhood distance and/or communication patterns, while in the latter, the prior influences the Gibbs Kernel matrix by modifying it to $\mathbf{M} \odot \exp{(-\mathbf{C}/\epsilon)}$, acting as a soft filter. Therefore, implausible paths that are highly inconsistent with SS or LR patterns are inherently discouraged in ContextFlow.
>
> > Q5: In Appendix B, Line 914 seems to have a sign error
>
> We thank the reviewer for catching the error, we've corrected in the newer version.

---

### Official Review · Reviewer_T3cD · 2025-10-31

**Soundness:** 3
**Presentation:** 3
**Contribution:** 3
**Rating:** 4
**Confidence:** 4

**Summary:**

This paper introduces a framework for improving the biological plausibility of trajectory inference from longitudinal spatial omics data.
The methodology is sound, and the experiments are thorough.

The core of the contribution is the method for integrating spatial priors into an OT-coupled flow matching framework.
By defining a "Transitional Plausibility Matrix" based on spatial smoothness and cell-cell communication, this approach moves beyond simple transcriptomic-only optimal transport (like MOTFM) to better pinpoint biologically realistic cellular trajectories.

**Strengths:**

The core contribution (the context-aware OT coupling, particularly the CTF-H entropy-regularized variant) is to exploit biological information, specifically the position of the cells, to regularize the transport plan towards biologically-informed priors, and its stability is supported by Theorem 1.

The authors provide empirical evidence that ContextFlow outperforms the state-of-the-art MOTFM baseline on multiple metrics.
Also, the study directly evaluates the biological plausibility of the inferred couplings (Fig. 3), providing clear evidence that the method successfully reduces the number of known-implausible transitions.

**Weaknesses:**

Some mathematical notations are ambiguous and inconsistently applied throughout the manuscript, making it difficult for the reader to follow the derivations and arguments, especially starting from Section 2.3.

Concrete examples are:
- The use of $[\cdot]$ notation for a set in Section 3.1 is unusual. Later on, the authors use ${\cdot}$ notation for a set in Section 3.2. Later on, $[\cdot]$ is used for an element of a matrix.
- In Section 3.1, $n_i$ is never explained.

The paper frames the problem primarily as an improvement over context-free flow matching (MOTFM).
However, there are existing methods specifically designed for spatial-temporal trajectory inference, e.g., DeST-OT, and also other methods against which DeST-OT was compared, such as Moscot, PASTE, SLAT, and STalgin, which are mentioned in the related work.
They tested their approach on the same data, i.e., Axolotl brain development, meaning direct comparison should be possible.
Discussing why a flow-matching approach might be preferable or complementary (e.g., generation of continuous trajectories vs. discrete couplings) would provide a more complete picture of the landscape.
Furthermore, in DeST-OT, there are multiple metrics introduced that would be interesting to see, such as the metric of cell migration and the metric of growth distortion.

The paper also states that CTF-H avoids the need for additional tuning, but this only refers to the $\alpha$ parameter in CTF-C.
Both methods still rely on the $\lambda$ parameter (Eq. 10) to balance the SS and LR priors.
The paper also fails to address the severity of this dependency.
Its own sensitivity analysis reveals that it is not a simple trade-off, but a point of high fragility.
For instance, in the challenging IVP interpolation task (Table 11), CTF-H with $\lambda = 1$ improves on the baseline, but with $\lambda = 0.5$, its performance collapses.
The paper provides no discussion of this instability or any heuristic for how to set this crucial parameter, which appears to be non-trivial and highly non-linear.
Additionally, there is an apparent inconsistency in the data.
For the IVP interpolation task, Table 11 shows improvement on the baseline with $\lambda = 1$ and performance collapses with $\lambda = 0.5$, whereas Table 3 appears to show improvement on the baseline with $\lambda = 0.5$ and performance collapses with $\lambda = 1$.

The paper introduces new computational steps (constructing the TPM, running a modified Sinkhorn algorithm), but provides no analysis of their computational complexity or practical runtime.
It is unclear how this method would scale to datasets with hundreds of thousands or millions of cells.
Furthermore, there are parameters, such as $r$, and also the hyperparameters of LIANA+, which would heavily influence the performance, yet there is no explanation or ablation study regarding those parameters.

The examples show that the framework can reduce biologically implausible couplings (from 54 to 24 in Fig. 3), which is a strong result.
However, 24 implausible couplings is still significant.
To fully substantiate the claim of generating biologically consistent trajectories, it would be powerful to showcase an example of a novel insight or a corrected trajectory that was non-obvious.

**Questions:**

- What are the potential limitations of the Spatial Smoothness prior (Eq. 8)?
- How robust is the framework's performance to the choice of the upstream cell-cell communication inference tool used to generate the LR prior?
- What is the computational complexity of constructing the TPM and running the prior-aware Sinkhorn algorithm, and how does the method's runtime scale as the number of cells increases?
- How was the trade-off parameter $\lambda$ (Eq. 10) selected for the final results? Since this parameter still needs to be set (or tuned) for both CTF-C and CTF-H, please clarify the claim that CTF-H reduces tuning overhead.
- There is no ablation study regarding the parameter $r$ and the hyperparameters of LIANA+ approach. How did your approach perform under different parameters, and how were they set?
- Why is the number of principal components set to 50 for all datasets, and is there no feature selection by selecting the highly variable genes?
- As mentioned above, the data presented in Tables 3 and 11 is inconsistent. The authors should review these tables for accuracy and correct them as needed.
- The priors are defined between consecutive time points. How would this framework handle missing time points (e.g., inferring a trajectory from $t_1$ to $t_3$)? Would the SS and LR priors still be valid over such a large temporal gap?
- Why were comparisons limited to other flow-matching methods? The paper would be strengthened by a comparison to other SOTA trajectory inference methods that are already spatially-aware, such as some of the OT-based alignment methods mentioned in the related work (e.g., DeST-OT).
- Can you provide an example where your model identifies a trajectory that reveals a previously non-obvious biological insight, rather than primarily confirming known plausible/implausible transitions?

---

> ### Author Response · Authors · 2025-11-22
> **Official Response to Reviewer T3cD**
>
> We thank the reviewer for their feedback.
>
> > W1: Some mathematical notations are ambiguous.
>
> We thank the reviewer for pointing out the notational inconsistencies; we've now corrected them.
>
> > W2:  Comparison with OT methods. Furthermore, in DeST-OT, there are multiple metrics introduced that would be interesting to see, such as the metric of cell migration and the metric of growth distortion.
>
> It's important to note that SOTA spatiotemporal OT methods, such as DeST-OT and TOAST, are primarily designed to provide the OT coupling matrix $\mathbf{\Pi}$ used for alignment tasks and do not learn dynamics across the time horizon. ContextFlow is a generative algorithm that utilizes OT to train a single neural network across multiple time horizons, resulting in more meaningful conditional paths. Therefore, a direct comparison of ContextFlow with DeST-OT is not appropriate.
>
> To provide a meaningful comparison, we've included a benchmark of our prior-aware OT formulation used in CTF-H, alongside recent methods such as DeST-OT and TOAST, on cell-migration/growth distortion metrics, along with coupled transcriptomic distances (Appendix H). Overall, we observe that our OT formulation is competitive with DeST-OT across metrics while being significantly faster.
>
> > W3: How to set the parameter $\lambda$?
>
> We've included an analysis of how $\lambda$ affects ContextFlow performance across different settings and datasets in Appendix K.1. Overall, we observe that optimal performance usually occurs near extreme values $\lambda =0, 1$, or near $\lambda=1$ (i.e., 0.8).
>
>
> > W4: Computational complexity and choices of $r$ and hyperparameters of LIANA+
>
> We clarify that the calculation of Ligand-Receptor interaction (LR) features and local spatial smoothness (SS) features is included in the data pre-processing step and is a one-time cost. While running ContextFlow, we do not recompute them, making our approach  scalable. We've included the time complexity analysis in Appendix E. To summarize, excluding the pre-processing steps, the total time complexity for training ContextFlow is $O(E \times P \times (B^{2}(d+l)))$, where $E$ is the training epochs, $P$ is the time taken for forward/backward pass using gradient descent, $B$ is the mini-batch size for computing minibatch OT couplings, $d$ is the transcriptomic feature dimension, and $l$ is the total number LR pairs.
>
> Defining a local neighborhood depends on the size and kind of dataset. For example, in Salamander Regeneration Stereoseq data, for the first Stage, we had approximately $1400$ samples, of which approximately $700$ were in each hemisphere, with approximately $10$ cell types. We thus set the radius to be $50$. We've included a more detailed radius ablation study in Appendix K.2. Overall, we observe that there is usually a sweet spot, depending on the type and nature of the dataset, whereas increasing/decreasing the radius too much degrades performance.
>
> For LIANA+, we use the standard recommended hyperparameter values (nz-prop: $0.1$, resource: Consensus). This setting is already robust, as Consensus utilizes several Ligand-Receptor Databases and considers the wisdom of crowds, rather than favoring any particular database.
>
> > W5: Example of a novel insight or a corrected trajectory that was non-obvious.
>
> At this stage, providing a genuinely new biological insight is outside the scope of our current work. Note that ContextFlow is regarded as a methodological framework designed to facilitate such discoveries. We've discussed this with domain experts, and reaching a point where one can reliably extract novel mechanisms will require better curated datasets and more focused biological follow-up. We hope to explore this in our future study.

---

> > ### Author Response · Authors · 2025-11-22
> > **Official Response to Reviewer T3cD**
> >
> > > Q1: Limitations of the Spatial Smoothness prior
> >
> > The main limitation of the spatial smoothness prior is that it's a _proxy_ measure indicating how close two cells are, between two timestamps, and thus how likely they are to be coupled. The reason we do not use exact spatial coordinates is that tissue slices lack a consistent frame of reference; they may be translated or rotated with respect to each other. However, better and faster ways of utilizing these spatial coordinates need to be further investigated.
> >
> > > Q2: How robust is the framework's performance to the choice of the upstream cell-cell communication inference tool used to generate the LR prior?
> >
> > We utilize the recommended Consensus resource from LIANA+, which consolidates several LR databases to determine LR communication patterns. Since we don't rely on any particular database, the pattern determination is already robust by design.
> >
> > > Q3: Computational complexity of constructing the TPM and prior-aware Sinkhorn algorithm.
> >
> > The theoretical computational complexity of constructing the TPM is $O(B^{2}d)$ with $B$ being the mini-batch size, and $d$ being the feature dimension. While running the prior-aware Sinkhorn, the complexity remains $O(B^{2})$, the same as running Sinkhorn for entropic-OT, where $B$ represents the minibatch size. We've added detailed time complexity analysis of ContextFlow in Appendix E.
> >
> > > Q4: No ablation study regarding the parameter $r$  and the hyperparameters of LIANA+. How were they set?
> >
> > As mentioned in our earlier comment, setting a proper $r$ is dataset dependent and relies on the least number of samples present across timesteps and the number of celltypes present there.  We've included ablations on $r$ in Appendix K.2. We used the recommended hyperparameters of LIANA+. For the liver regeneration data, which is a Visium dataset, we lowered nz-prop to $0.006$ since we didn't observe an LR interaction at $0.1$, with the rest of the hyperparameters kept at their default values.
> >
> > > Q5: Why is the number of principal components set to 50 for all datasets, and is there no feature selection by selecting the highly variable genes?
> >
> > We kept the number of PCs to $50$, as prior OT studies using these datasets worked with $50$ PCs. For each dataset, we select the top $10,000$ highly variable genes.
> >
> > > Q6: The data presented in Tables 3 and 11 are inconsistent
> >
> > Thank you for pointing it out. We have corrected it in our revised manuscript.
> >
> > > Q7: How would this framework handle longer missing time points, eg t1/t4?
> >
> > We assume that we can still extract meaningful relationships between these timesteps. The applicability of any flow-based generative AI algorithm is naturally limited to datasets where the temporal transcriptomic variations aren't far apart from each other, because otherwise there isn't enough knowledge in the data to learn the ground truth dynamics, unless externally provided by some inductive prior or implicitly present in the network via massive pretraining. If the data from holdout timesteps deviates too much from the presented samples, then even with additional contextual information, no model is capable of recovering it.
> >
> > > Q8: Why were comparisons limited to other flow-matching methods? The paper would be strengthened by a comparison to other SOTA trajectory inference methods that are already spatially-aware, such as some of the OT-based alignment methods mentioned in the related work (e.g., DeST-OT).
> >
> > DeST-OT is not a generative technique; thus, comparison with generative modeling techniques like flow-matching isn't appropriate. However, we've added a comparison between DeST-OT, TOAST, and prior-aware OT, as used in ContextFlow, on metrics such as the growth distortion metric, migration metric, and coupled transcriptomic distance in Appendix E. Overall, we observe that our prior-aware OT is competitive with SOTA methods, while being much faster.
> >
> > > Q9: Can you provide an example where your model identifies a trajectory that reveals a previously non-obvious biological insight, rather than primarily confirming known plausible/implausible transitions?
> >
> > As mentioned earlier, providing a genuinely new biological insight is outside the scope of our current work. ContextFlow is a methodological framework designed to facilitate such discoveries. We've also discussed this with domain experts, and reaching a point where one can reliably extract novel mechanisms will require better curated datasets and more focused biological follow-up. We hope to explore this in our future work.

---

### Official Review · Reviewer_DU9Q · 2025-10-31

**Soundness:** 3
**Presentation:** 3
**Contribution:** 3
**Rating:** 6
**Confidence:** 3

**Summary:**

ContextFlow introduces a context-aware flow-matching framework for inferring tissue dynamics from longitudinal spatial omics data. By integrating information on tissue organization and ligand–receptor interactions into the model, it regularizes trajectory inference to produce biologically interpretable and statistically consistent results. Evaluations on real world datasets demonstrate that ContextFlow outperforms existing flow-matching approaches in both accuracy and biological coherence.

**Strengths:**

1.	Incorporating biological priors to regularize the flow is quite useful in the field of spatial transcriptomics analysis, as it provides biologically meaningful interpretations.
2.	The overall presentation of this paper is comprehensive, and the results show the proposed method outperform the baseline in different metrics.
3.     The approach seems quite novel, and the combination of the model and this application is an important contribution.

**Weaknesses:**

1.	Although the authors benchmark their model on real datasets, I recommend also including evaluations on simulated datasets, as these provide unbiased ground truth for performance assessment.
2.	It would be helpful to organize the related work section more clearly, for example by summarizing the advantages of different methods in a table, as the current related work section is not well organized.
3.	If possible, it would be helpful to also include a benchmarking comparison with the DeST-OT method mentioned in the related work. Since one key advantage of ContextFlow is the incorporation of biologically meaningful priors, such as cell–cell communication information, and DeST-OT can model cell differentiation processes, a comparison between these two methods would be interesting.

**Questions:**

1.	From the results, it appears that CTF-H performs better than CTF-C. Could the authors provide a general explanation for this observation? For instance, do different prior integration strategies have distinct advantages depending on the application scenario?
2.	For the Stereo-seq dataset, what is the resolution for this dataset, did the author used cell binning during the preprocessing steps.

---

> ### Author Response · Authors · 2025-11-22
> **Official Response to Reviewer DU9Q**
>
> We thank the reviewer for their feedback.
>
> > W1: Although the authors benchmark their model on real datasets, I recommend also including evaluations on simulated datasets, as these provide unbiased ground truth for performance assessment.
>
> We agree that simulations with known ground truth can be a valuable addition. However, ContextFlow explicitly relies on LR-based priors inferred from curated databases and complex spatial structure. Designing synthetic data that provides realistic or unbiased representations of ligand-receptor (LR) relationships is a nontrivial task, as these depend on curated molecular databases and experimentally derived signaling networks. Existing simulators do not model these nor capture such complex relationships well. We thus conduct studies on real datasets, as is usually the case in spatial omics studies.
>
>
> > W2: It would be helpful to organize the related work section more clearly, for example, by summarizing the advantages of different methods in a table, as the current related work section is not well organized.
>
> According to the reviewer's suggestion, we've organised the related work section better and have included a comparison table (Appendix A), highlighting properties of prior works while positioning ContextFlow within them.
>
> > W3: If possible, it would be helpful to also include a benchmarking comparison with the DeST-OT method mentioned in the related work. Since one key advantage of ContextFlow is the incorporation of biologically meaningful priors, such as cell–cell communication information, and DeST-OT can model cell differentiation processes, a comparison between these two methods would be interesting.
>
> It is important to note that DeST-OT and other OT methods from the related section are primarily designed for _aligning_ spatiotemporal omics data, and are not generative models. In contrast, ContextFlow is a generative model where a single neural network models the dynamics across the time horizon. Thus, direct comparison between ContextFlow and DeST-OT isn't appropriate; nevertheless, we've added a comprehensive comparison between the OT formulation used to generate conditional paths in ContextFlow, DeST-OT, and TOAST (Appendix H). We observe that our proposed prior-aware OT formulation is competitive with DeST-OT/TOAST across metrics while being significantly faster than them.
>
> > Q1: From the results, it appears that CTF-H performs better than CTF-C. Could the authors provide a general explanation for this observation? For instance, do different prior integration strategies have distinct advantages depending on the application scenario?
>
> Empirically, while both incorporate prior biological knowledge by regularizing the OT objective, CTF-C imposes harder OT regularization by directly modifying the cost matrix, whereas CTF-H's softer regularization effectively changes the Gibbs Kernel while keeping the cost matrix unchanged, leading to more restricted couplings in the case of CTF-C. And regularization is intended to _guide_ the dynamical flow rather than restrict it, which intuitively and empirically happens better in CTF-H than CTF-C. Having said that, in cases with perfect prior knowledge with constraints that absolutely need to be satisfied, perhaps CTF-C might be better.
>
> > Q2: For the Stereo-seq dataset, what is the resolution for this dataset, did the author used cell binning during the preprocessing steps.
>
> The resolution is at the single-cell level; no binning is done during preprocessing.

---

### Author Response · Authors · 2025-11-22
**Global Response to All Reviewers**

We thank all the reviewers for their insightful, meticulous, and highly constructive feedback, which has significantly strengthened the quality of our work. Below, we summarize the key changes to our revised manuscript. To assist the reviewers in recognizing the modified content more easily, we highlighted them in blue.

**Related Work.** We have added Table 5 to Appendix A of the revised manuscript to summarize the key desirable features of ContextFlow, along with comparisons to several existing spatiotemporal optimal transport (OT) and flow matching methods.

**Time Complexity.** We've analyzed the time complexity of the components involved in our ContextFlow algorithm (data preprocessing, TPM construction, Sinkhorn algorithm). The analyses are provided in Appendix E of the revised manuscript.


**Comparison to Spatiotemporal OT.** We have added Appendix H to the revised manuscript to clarify that ContextFlow is a biological prior-guided generative model that learns a velocity field across the entire time horizon, whereas existing spatiotemporal OT methods are designed for pairwise alignment tasks and are not generative. Similarly, the spatial priors utilize a local neighborhood expression as a proxy and do not directly use spatial coordinates due to the lack of a shared global frame of reference between slices. The overall goal of ContextFlow is to utilize biological prior knowledge while remaining scalable, as shown in the runtime analysis. In addition, we have conducted additional experiments to compare our proposed prior-aware entropy regularized OT module, as presented in ContextFlow, with state-of-the-art spatiotemporal OT methods, such as DeST-OT and TOAST (Appendix H.1), along with a runtime analysis for ContextFlow (Appendix H.2).

**Hyperparameter Sensitivity Analysis.** We have comprehensively conducted ablation studies on hyperparameters, including $\lambda$, $r$, and $\epsilon$. The results are documented and discussed in Appendix K of the revised manuscript.

---

### Author Response · Authors · 2025-12-02
**Message to the new Area Chair**

Dear Area Chair,

In light of the recent unforeseen events, we would like to summarize our rebuttal to help you make an informed decision. The following concerns were commonly raised across the reviewers:

- **Runtime of ContextFlow** (Reviewers T3cD, w9VN, and ZUfA). We addressed this concern in our runtime analysis in Appendix E, where we demonstrated that ContextFlow's runtime is comparable to that of Minibatch-OT Flow Matching. We also modified the algorithm in Appendix D to explicitly indicate that the domain priors are included as one-time calculations in the pre-processing step, thereby making ContextFlow scalable.

- **Comparison with Spatiotemporal OT methods** (Reviewers DU9Q, T3cD, and ZUfA).

    - We clarified that prior spatiotemporal-OT methods (Waddington-OT, DeST-OT, TOAST) gather a ***static-OT*** coupling primarily used for alignment tasks, and are **not generative** across the entire time horizon, unlike ContextFlow.
    - We included an appropriate static-OT comparison of the prior-aware OT formulation, CTF-OT, used within ContextFlow (to generate conditional paths), with recent state-of-the-art spatiotemporal OT methods, such as DeST-OT and TOAST, on growth-centric metrics in Appendix H. CTF-OT is both competitive and scalable.
    - We also included a table in Appendix A to facilitate the positioning of ContextFlow within the prior literature.


- **Hyperparameter Ablations** (Reviewers T3cD, s8Za, w9VN, and ZUfA). We addressed this concern by including an exhaustive hyperparameter sensitivity analysis in Appendix K, for hyperparameters $\lambda$, $r$, and $\epsilon$. We also discussed their effect on performance metrics along with suggestions for setting them up.

- **Overlapping Conceptual Clarifications.**

    - **Exclusion of Spatial Distance** (Reviewers s8Za and w9VN). We clarified that direct spatial distance is excluded owing to the lack of a global frame of reference between tissue slices in practice.
    - **Enforcing Constraints at Endpoints** (Reviewers T3cD and w9VN). We clarified that ContextFlow is guided by a biological prior, with enforced constraints across the entire time horizon, not just at endpoints.
    - **LR database sensitivity** (Reviewers T3cD and ZUfA). We clarified that ContextFlow uses the LIANA+ recommended Consensus resource, which consolidates many frequently used LR databases, making the communication patterns robust by design.

Beyond the common concerns, we addressed the remaining comments of individual reviewers in our responses. We hope that the above summary will facilitate your navigation of our revised manuscript and enable you to assess how effectively our rebuttal addresses the reviewers' concerns.

---

### Meta-Review · Area_Chair_c9DZ · 2025-12-29

**Summary:**

1. **Novelty and Contribution**: Some reviewers felt that the core idea of incorporating biological priors into Optimal Transport (OT) objectives was incremental, particularly because the two proposed schemes (PACM and PAER) built upon existing methods like KL divergence regularization. There was also a call for clearer differentiation between ContextFlow and existing spatiotemporal OT methods, such as TOAST and DeST-OT.

2. **Evaluation and Experimental Results**: Several reviewers pointed out that the improvements over existing baselines (such as MOTFM) were modest, often within error bars. Additionally, they noted the extreme instability observed in some extrapolation tasks (e.g., Table 12), which raised concerns about the practical utility of the framework in certain settings. The lack of comparisons to other trajectory inference methods (e.g., Waddington-OT, CellRank, RNA velocity) was also highlighted as a limitation.

3. **Hyperparameter Sensitivity**: There were concerns regarding the lack of a systematic study of key hyperparameters, such as the neighborhood radius and the entropy regularization parameter. Reviewers pointed out that while the paper claimed that PAER reduced tuning efforts, the model still heavily depended on these parameters, and guidance on their selection was insufficient.

4. **Methodological Clarity and Technical Issues**: Reviewers noted issues with the mathematical notation and formulations, particularly in how the TPM (Transition Plausibility Matrix) was constructed and normalized. There were also concerns about the exclusion of direct spatial distance in the OT cost function and the use of neighborhood-based proxies, which may lead to biologically implausible long-range couplings.

5. **Computational Complexity and Scalability**: The scalability of the framework to large datasets with many cells was questioned. While the authors mentioned the efficiency of their method, no detailed runtime comparisons or scalability analysis were provided in the original submission.

6. **Biological Validation**: Some reviewers expressed that while the paper provided qualitative biological validation (e.g., the excitatory-inhibitory transitions), this was limited to a single dataset. They called for more examples where the framework revealed novel biological insights or corrected previously implausible trajectories.

7. **Statistical and Evaluation Methodology**: The paper did not provide statistical tests to validate the improvements reported in the experiments. The lack of significance testing and the potential confounding effects of the weighted W2 metric (based on XGBoost classification) were also pointed out.

8. **Theoretical Gaps and Justification**: Reviewers noted the theoretical gap between static OT couplings and dynamic trajectory generation. The paper did not adequately justify why linear interpolation between coupled endpoints was sufficient for generating biologically valid trajectories, and there was no empirical or theoretical explanation for the choice of linear combination for combining SS and LR priors.

**Reviewer Concerns:**

**Unaddressed Concerns:**

1. **Novelty of the Contribution**: The reviewers remain unconvinced about the novelty of the proposed approach, as the core idea of incorporating prior biological knowledge into OT objectives was seen as incremental. Despite the authors' response, which emphasized the differences between ContextFlow and existing methods, the concern regarding the incremental nature of the contribution persists.

2. **Modest Improvements Over Baselines**: The reviewers noted that improvements over MOTFM were often small (less than 5-10%) and within error bars. The rebuttal did not sufficiently address the significance of these improvements, and the reviewers still question the practical impact of these marginal gains.

3. **Hyperparameter Sensitivity**: Although the authors conducted an ablation study on hyperparameters, the reviewers continued to express concerns about the lack of clear guidance on how to select crucial parameters like the neighborhood radius and entropy regularization parameter. The absence of a principled approach to hyperparameter tuning remains a key issue.

4. **Instability in Extrapolation**: The extreme instability observed in extrapolation tasks (Table 12) was not convincingly addressed. The reviewers remain concerned about the practical utility of the model, particularly in difficult extrapolation scenarios.

5. **Lack of Comparison to Other Trajectory Inference Methods**: The reviewers pointed out that the paper did not compare ContextFlow to well-established trajectory inference methods like Waddington-OT, CellRank, and RNA velocity. While the authors provided some benchmarks, this concern was not fully addressed in the rebuttal.

**Addressed Concerns:**

1. **Methodological Clarity**: The authors clarified issues regarding mathematical notation, particularly the normalization of the TPM, and addressed the concerns about the exclusion of direct spatial distance by explaining their use of spatial smoothness as a proxy for relative location. The reviewers accepted these clarifications, though the underlying methodological approach (using proxies) still remains a concern.

2. **Computational Complexity and Scalability**: The authors provided additional details on the runtime analysis and scalability of ContextFlow. While some reviewers still question the scalability to large datasets, the rebuttal provided useful clarification regarding the computational efficiency, particularly with respect to the pre-processing steps.

3. **Biological Validation**: The authors clarified the limited scope of the current biological validation (excitatory-inhibitory transitions) and acknowledged the need for further biological insights in future work. While this was accepted, it remains a point that was not fully addressed in terms of providing truly novel biological insights.

4. **Evaluation Methodology**: The reviewers' concern about the weighted W2 metric was addressed with further clarification on how the XGBoost classifier was trained and its accuracy. However, the issue of statistical significance testing was not fully resolved, as the rebuttal did not provide substantial changes to the evaluation methodology.

**Reviewer Scores:**

1. **Reviewer DU9Q**:
   Given that this reviewer expressed reservations about the novelty and requested further comparisons, especially with simulated datasets and spatiotemporal methods like DeST-OT, it is likely that they would have maintained their score of **6** (marginally above the acceptance threshold). The concerns about the small improvements and lack of statistical significance would still hold weight, and despite some clarifications, the overall impression would remain skeptical of the practical impact.

2. **Reviewer T3cD**:
   This reviewer raised concerns about the ambiguous mathematical notations, the theoretical gap between static coupling and dynamic trajectory generation, and the lack of a clear strategy for setting hyperparameters. Given the author's clarifications, they may have slightly improved their score from **4** (marginally below the acceptance threshold), but the unresolved issues regarding the methodological robustness, hyperparameter sensitivity, and the limited novelty likely would have kept them from being fully convinced.

3. **Reviewer s8Za**:
   This reviewer emphasized concerns about the sensitivity of the method to hyperparameters and the lack of a detailed analysis of spatial smoothness vs. LR communication independently. While the ablation studies provided some answers, the reviewer likely would have kept their score at **4** (marginally below the acceptance threshold), as they remain concerned about the instability, the modest improvements, and the lack of biological justification for the dominance of one prior over the other in certain datasets.

4. **Reviewer w9VN**:
   This reviewer focused on the omission of spatial coordinates, experimental limitations, and the lack of comparisons to other trajectory inference methods. Although the rebuttal clarified several points, particularly about the exclusion of spatial distance and provided further clarifications, the reviewer would likely have maintained their **4** rating (marginally below the acceptance threshold) due to unresolved issues regarding methodological rigor, instability in extrapolation, and the novelty of the approach.

5. **Reviewer ZUfA**:
   Despite clarifications regarding the hyperparameter sensitivity, computational complexity, and the handling of spatial smoothness and LR priors, this reviewer still emphasized the modest improvements and lack of novelty. Given the unaddressed concerns regarding the comparison to other trajectory inference methods, instability, and the insufficient biological validation, the reviewer would likely have maintained their score at **6** (marginally above the acceptance threshold), with their concerns about the limited novelty and experimental setup likely keeping the score just above the threshold.

---

### Decision · Program_Chairs · 2026-01-26

Reject